# Refined Analysis of Entropy-Regularized Actor-Critic

**Safwan Labbi** [1]   **Paul Mangold** [1]   **Daniil Tiapkin** [1 2 ⋆]   **Eric Moulines** [3 4]

## Abstract

In this paper, we study the role of the critic in actor–critic for entropy-regularized, finite, discounted environments. We establish that, when the critic is exact, using the latter as a baseline is a variance-reduction method in a strong sense. In this case, actor–critic with stochastic gradients matches the sample complexity of deterministic policy gradient, reaching an $\epsilon$-optimal regularized value with $\tilde{O}(\log(1/\epsilon))$ samples. In practice, the critic is learned alongside the actor: the variance of the actor update is then influenced by the critic's variance and bias. Specifically, when the critic has a sufficiently small error, the variance reduction and rapid convergence are preserved. This suggests to learn the critic first, keeping it up to date after each actor update, underscoring the crucial role of accurate critic estimation in actor–critic methods.

## 1. Introduction

Policy gradient methods are among the most widely used reinforcement learning (RL) algorithms (Williams, 1992; Sutton et al., 1998). Due to their inherent flexibility and scalability, they have emerged as the dominant approach in modern RL (Agarwal et al., 2021). Their widespread adoption has led to the development of numerous techniques and tricks to stabilize training and accelerate convergence, enabling faster discovery of high-performing policies.

A central technique in policy gradient methods is to introduce a *baseline*, which serves as a control variate that aims to reduce the variance of the gradient estimates (Konda & Tsitsiklis, 1999). This comes from the observation that,

when expressing the gradient, any term that does not change the expected gradient direction can be used to stabilize optimization. In practice, there is a strong consensus on the fact that using the value function of the current policy strongly enhances performance (Grondman et al., 2012; Schulman et al., 2015b). This effectively shifts the gradient's focus from 'rewards' to the so-called 'advantage' of one action over another (Baird & Leemon, 1993), thereby providing a denser learning signal. This observation is further supported by the dominance of *actor-critic* methods (AC, Barto et al. 1983; Konda & Tsitsiklis 1999), and most specifically algorithms like A2C (Mnih et al., 2016), PPO (Schulman et al., 2017), TRPO (Schulman et al., 2015a), among many others. Still, although this shift feels like a natural progression, the specific role of the value function as a stabilizer is poorly understood theoretically.

Recently, the theory of RL has seen a surge of novel theoretical analyses, providing rigorous foundations for many fundamental methods. While earlier work relied on asymptotic guarantees (Williams, 1992; Greensmith et al., 2004), recent studies established non-asymptotic, global convergence rates for policy gradient methods (Mei et al., 2020b; Xiao, 2022; Labbi et al., 2026b). More recently, theory has been extended to AC, proving global convergence in finite-time (Kumar et al., 2024), following analyses of policy gradient by relying on a uniform bound on the gradients.

Parallel to these theoretical advances, it has recently been observed empirically that improving the critic considerably improves the convergence of AC (Wang et al., 2025). While this gives strong evidence that AC reduces gradient variance, it is not clear to what extent. Specifically, we can distinguish between two types of variance reduction:

- *weak variance reduction*: the variance of the updates is reduced by a multiplicative constant, akin to tail averaging (Polyak & Juditsky, 1992) or moving average methods (Morales-Brotons et al., 2024);
- *strong variance reduction*: the variance of the gradient estimator vanishes as iterates approach the optimum, giving linear convergence, like SVRG (Johnson & Zhang, 2013), SAGA (Defazio et al., 2014), and other methods.

Naturally, strong variance-reduction results in much faster convergence rates than its weak counterpart. To our knowledge, it remains unknown whether AC methods achieve

[1]CMAP, CNRS, École Polytechnique, Institut Polytechnique de Paris, 91120 Palaiseau, France [2]Université Paris-Saclay, CNRS, LMO, 91405, Orsay, France. ⋆Now at Google DeepMind. [3]Mohamed bin Zayed University of Artificial Intelligence, UAE [4]LRE EPITA , 94270 Le Kremlin-Bicêtre, France. . Correspondence to: Safwan Labbi <safwan.labbi@polytechnique.edu>.

*Proceedings of the 43ʳᵈ International Conference on Machine Learning*, Seoul, South Korea. PMLR 306, 2026. Copyright 2026 by the author(s).

*weak* or *strong* variance reduction.

In this paper, we answer positively: *yes, AC achieves strong variance reduction*, at least when the critic is known. Our theory follows from the observation that, when the value function (i.e., the critic) is perfectly known, the variance of AC's stochastic gradient can be bounded by the norm of the deterministic gradient up to a multiplicative factor. In this case, we show that AC achieves $\mathcal{O}(\log(1/\epsilon))$ sample complexity, matching the deterministic iteration complexity. Alternatively, if the critic is learned online, we establish that AC's behavior is governed by the critic's bias and variance. Specifically, we show that if sufficiently many critic iterations are performed, AC attains $\tilde{O}(1/\epsilon)$ sample complexity. Our contributions can be summarized as follows:

- We propose a novel theoretical analysis of actor-critic. To the best of our knowledge, we provide the first proof that AC with an exact critic *acts as a strong variance-reduction method*, not merely reducing the residual variance of the updates, but entirely eliminating it.

- When the critic is inexact, it must be learned alongside the actor. In that case, the critic's bias and variance induce similar bias and variance on the actor. Consequently, when properly setting up the algorithm, one can learn the actor at a rate similar to the rate of learning the critic: in some sense, the most important part of the training is the critic, and it pays off to spend time learning it. Crucially, one needs to perform multiple updates of the critic in between actor updates, confirming recent empirical findings (Wang et al., 2025).

- We empirically confirm our findings on two environments, showing that the performance of the learned policy monotonically increases as the number of updates of the critic between consecutive actor updates increases.

We give a comparison with the closest works in Table 1, and discuss related work in Section 2. Section 3 introduces the background. We analyze AC with exact critic in Section 4, and with inexact critic in Section 5. Empirical study is in Section 6, and we discuss perspectives in Section 7.

## 2. Related Works

**Entropy-Regularized RL.** Entropy-regularized RL promotes stochastic policies by rewarding higher entropy, encouraging exploration, and improving stability in learning (Williams & Peng, 1991; Mnih et al., 2016; Neu et al., 2017; Haarnoja et al., 2018). It underpins widely used deep RL methods such as Soft Actor–Critic and entropy-regularized AC (Haarnoja et al., 2018), which remain poorly understood in theory despite their impressive practical performance. A line of work studies this objective both algorithmically and theoretically (Nachum et al.,

2017; Geist et al., 2019), relating entropy regularization to soft policy iteration and soft Q-learning, making explicit the links between policy-gradient and value-based methods. Another line of work (Lan, 2023) studies Policy Mirror Descent for entropy-regularized reinforcement learning and establishes convergence guarantees for the regularized objective. However, Policy Mirror Descent operates directly in policy space, which makes extensions beyond the tabular setting less straightforward. In contrast, actor–critic methods update parametrized policies directly, providing a more natural foundation for scalable extensions and motivating a dedicated analysis of actor–critic methods.

**Policy Gradient Methods.** Policy gradient (PG) methods date back to Williams (1992); Sutton et al. (1999). Mei et al. (2020b); Zhang et al. (2020); Xiao (2022) established global convergence for softmax policies with *exact* gradients by showing that the RL objective satisfies Łojasiewicz-type inequalities, obtaining linear convergence rates in entropy-regularized RL. With stochastic gradients, Zhang et al. (2021b); Yuan et al. (2022) proved convergence to first-order stationary points under Monte-Carlo gradient estimates, and subsequent work showed that global guarantees can be recovered under additional structure, notably through regularization (Zhang et al., 2021a; Ding et al., 2025; Labbi et al., 2026a;b). However, vanilla stochastic PG remains brittle in practice due to high variance, motivating the use of more involved methods. Several recent works establish convergence guarantees for variants of PG, relying for instance on Hessian-based variance reduction (Fatkhullin et al., 2023), momentum and importance sampling (Barakat et al., 2023), or inverse-Fisher preconditioning (Mondal & Aggarwal, 2024). The approaches of Fatkhullin et al. (2023) and Mondal & Aggarwal (2024) typically require heavier computations, such as estimating second-order quantities or inverting Fisher-type matrices, while the variance-reduction method of Barakat et al. (2023) is less competitive in practice than actor–critic methods. This motivates our focus on actor–critic methods (Barto et al., 1983; Konda & Tsitsiklis, 1999), which provide a scalable parameter-space framework while enabling variance reduction through the critic.

**Convergence analysis of Actor–Critic.** Early analyses of AC were asymptotic, using two-timescale stochastic approximation (Konda & Tsitsiklis, 1999), or ODE-based arguments (Bhatnagar et al., 2009; Castro & Meir, 2010). More recently, non-asymptotic rates have been obtained in special control settings: for LQR, Yang et al. (2019) proves global linear convergence, but requires $\tilde{\mathcal{O}}(\epsilon^{-5})$ critic updates in order to maintain sufficient value-estimation accuracy. In the general RL setting, several works establish non-asymptotic convergence to stationary points with finite-sample complexity bounds (Xu et al., 2020; Qiu et al.,

*Table 1.* Comparison with related actor-critic methods with unknown critic. Our method is the first to achieve $\tilde{\mathcal{O}}(1/\epsilon)$ sample complexity.

| Type | Algorithm | Strong Variance Reduction | Global Convergence | Sample complexity |
|---|---|:---:|:---:|:---:|
| | Olshevsky & Gharesifard 2023 | ✗ | ✗ | $\tilde{\mathcal{O}}(1/\epsilon^2)$ |
| Unregularized | Kumar et al. 2024 | ✗ | ✓ | $\tilde{\mathcal{O}}(1/\epsilon^3)$ |
| | Gaur et al. 2024 | ✗ | ✓ | $\tilde{\mathcal{O}}(1/\epsilon^3)$ |
| Entropy-Regularized[(1)] | Cayci et al. 2024 | ✗ | ✓ | $\tilde{\mathcal{O}}(1/\epsilon^5)$ |
| | `Ent-AC` (our work) | ✓ | ✓ | $\tilde{\mathcal{O}}(1/\epsilon)$ |

(1) The results here are provided for the entropy-regularized problem.

2021; Kumar et al., 2023; Olshevsky & Gharesifard, 2023; Chen & Zhao, 2023). In the unregularized setting, global convergence guarantees are obtained by combining stability arguments with uniform gradient/exploration controls (Kumar et al., 2024; Gaur et al., 2024), achieving $\tilde{O}(\epsilon^{-3})$ sample complexity. For the tabular entropy-regularized objective, Cayci et al. (2024) also derives finite-sample guarantees, albeit with even larger complexity $\tilde{O}(\epsilon^{-5})$. In continuous-action settings, recent analyses (Zorba et al., 2026; Kerimkulov et al., 2025) establish global convergence guarantees for entropy-regularized actor–critic methods, but only in deterministic regimes. In contrast, we analyze entropy-regularized AC in the stochastic setting and show that it is a *strong variance reduction method* when the critic is exact, reaching $\tilde{O}(\log(1/\epsilon))$ sample complexity; furthermore, we show that with inexact critic, entropy-regularized AC still achieves $\tilde{O}(\epsilon^{-1})$ sample complexity.

## 3. Background

**Markov Decision Process.** We consider a discounted MDP $\mathcal{M} = (\mathcal{S}, \mathcal{A}, \gamma, \mathsf{P}, \mathsf{r}, \rho)$ with finite state and action spaces $\mathcal{S}, \mathcal{A}$, discount factor $\gamma \in (0,1)$, transition kernel $\mathsf{P}(s'|s,a)$, reward $\mathsf{r}(s,a) \in [0,1]$, and initial distribution $\rho$. A stationary policy $\pi \colon \mathcal{S} \to \mathcal{P}(\mathcal{A})$ induces $\mathsf{P}_\pi(s'|s) \triangleq \sum_a \mathsf{P}(s'|s,a)\pi(a|s)$. The value function is

$$\mathsf{v}_\pi(s) \triangleq \mathbb{E}_s^\pi \left[ \sum_{t=0}^\infty \gamma^t \mathsf{r}(S_t, A_t) \right], \quad (1)$$

with $S_0 = s$, $A_t \sim \pi(\cdot|S_t)$, $S_{t+1} \sim \mathsf{P}(\cdot|S_t, A_t)$. For $\rho \in \mathcal{P}(\mathcal{S})$, define $\mathsf{v}_\pi(\rho) \triangleq \sum_s \rho(s)\mathsf{v}_\pi(s)$. For a given policy $\pi$, we define the occupancy measure

$$d_\rho^\pi(s) \triangleq (1-\gamma) \sum_{t=0}^\infty \gamma^t \rho \mathsf{P}_\pi^t(s) \quad (2)$$

of $\pi$, measuring the discounted probability of visiting states along trajectories generated by $\pi$ starting from $\rho$.

**Entropy-regularized RL.** In entropy-regularized RL, near-deterministic policies are penalized by modifying the value of a policy $\pi$ to

$$\tilde{\mathsf{v}}_\pi^\lambda(s) \triangleq \mathbb{E}_s^\pi \left[ \sum_{t=0}^\infty \gamma^t \tilde{\mathsf{r}}_\pi(S_t, A_t) \right], \quad (3)$$

$$\tilde{\mathsf{r}}_\pi(s,a) \triangleq \mathsf{r}(s,a) - \lambda \log(\pi(a|s)), \quad (4)$$

where $\lambda > 0$ is the temperature, which determines the strength of the penalty. We also define the regularized Q-function and regularized advantage as

$$\tilde{\mathsf{q}}_\pi^\lambda(s,a) \triangleq \mathsf{r}(s,a) + \gamma \sum_{s' \in \mathcal{S}} \mathsf{P}(s'|s,a)\tilde{\mathsf{v}}_\pi^\lambda(s'), \quad (5)$$

$$\tilde{\mathsf{a}}_\pi^\lambda(s,a) \triangleq \tilde{\mathsf{q}}_\pi^\lambda(s,a) - \lambda \log(\pi(a|s)) - \tilde{\mathsf{v}}_\pi^\lambda(s). \quad (6)$$

Importantly, this regularized value satisfies the following regularized Bellman equation

$$\tilde{\mathsf{v}}_\pi^\lambda(s) = \sum_{a \in \mathcal{A}} \pi(a|s)\left(\tilde{\mathsf{q}}_\pi^\lambda(s,a) - \lambda \log \pi(a|s)\right). \quad (7)$$

The optimal regularized value function, defined by $\tilde{\mathsf{v}}_\star^\lambda(s) \triangleq \max_{\pi \in \Pi} \tilde{\mathsf{v}}_\pi^\lambda(s)$, satisfies the following consistency equations (Nachum et al., 2017; Geist et al., 2019):

$$\tilde{\mathsf{v}}_\star^\lambda(s) = \lambda \log\left(\sum_{a \in \mathcal{A}} \exp(\tilde{\mathsf{q}}_\star^\lambda(s,a)/\lambda)\right) \quad (8)$$

$$\pi_\star^\lambda(a|s) = \exp\left((\tilde{\mathsf{q}}_\star^\lambda(s,a) - \tilde{\mathsf{v}}_\star^\lambda(s))/\lambda\right), \quad (9)$$

which link the optimal values and policies together. In this work, we consider *softmax policies*; that is, given $\theta \in \mathbb{R}^{|\mathcal{S}||\mathcal{A}|}$, we consider the policy defined as $\pi_\theta(a|s) \propto \exp(\theta(s,a))$ together with the proper normalization. Given these definitions, we aim to optimize the regularized value function with softmax parametrization

$$\max_{\theta \in \mathbb{R}^{|\mathcal{S}||\mathcal{A}|}} \left\{ \tilde{J}_\lambda(\theta) \triangleq \tilde{\mathsf{v}}_{\pi_\theta}^\lambda(\rho) \right\}, \quad (10)$$

where we define $\tilde{J}_\lambda^\star \triangleq \max_{\theta \in \mathbb{R}^{|\mathcal{S}||\mathcal{A}|}} \tilde{J}_\lambda(\theta)$. With a slight abuse of notation, we also define $d_\rho^\theta \triangleq d_\rho^{\pi_\theta}$, $\tilde{\mathsf{q}}_\theta^\lambda \triangleq \tilde{\mathsf{q}}_{\pi_\theta}^\lambda$, and $\tilde{\mathsf{v}}_\theta^\lambda \triangleq \tilde{\mathsf{v}}_{\pi_\theta}^\lambda$.

**Properties of the regularized value.** The gradient of the regularized value can be expressed as follows.

**Lemma 1** (Lemma 10 of Mei et al. 2020b). *For any $\theta \in \mathbb{R}^{|\mathcal{S}||\mathcal{A}|}$, and $(s,a) \in \mathcal{S} \times \mathcal{A}$, it holds that*

$$\frac{\partial \tilde{\mathsf{v}}_{\pi_\theta}^\lambda(\rho)}{\partial \theta(s,a)} = \frac{1}{1-\gamma} d_\rho^{\pi_\theta}(s)\pi_\theta(a|s)\tilde{\mathsf{a}}_{\pi_\theta}^\lambda(s,a).$$

The regularized value function is also smooth with respect to $\theta$. Specifically, it satisfies the following lemma

**Lemma 2** (Lemma 7 and 14 of Mei et al. 2020b). *The regularized value $\tilde{\mathsf{v}}_{\pi_\theta}^\lambda(\rho)$ is L-smooth with*

$$L \triangleq (8 + \lambda(4 + 8\log(|\mathcal{A}|)))/(1-\gamma)^3.$$

We now introduce the classical state exploration assumption (Mei et al., 2020a;b; Agarwal et al., 2021).

**Assumption $A_\rho$.** *The smallest coefficient $\rho_{\min} \triangleq \min_{s \in \mathcal{S}} \rho(s)$ of the initial distribution $\rho$ satisfies $\rho_{\min} > 0$.*

Under the previous assumption, the regularized value satisfies a Non-Uniform Łojasiewicz property.

**Lemma 3** (Lemma 15 of Mei et al. 2020b). *For any $\theta \in \mathbb{R}^{|\mathcal{S}||\mathcal{A}|}$, we have*

$$\|\nabla_\theta \tilde{v}^\lambda_{\pi_\theta}(\rho)\|^2 \geq \tilde{\mu}_\lambda(\theta) \left( \tilde{v}^\lambda_\star(\rho) - \tilde{v}^\lambda_{\pi_\theta}(\rho) \right) \ ,$$

$$\tilde{\mu}_\lambda(\theta) \triangleq \lambda(1-\gamma)\rho^2_{\min} \min_{(s,a) \in \mathcal{S} \times \mathcal{A}} \pi_\theta(a|s)^2/|\mathcal{S}| \ ,$$

*and where $\tilde{v}^\lambda_\star = \max_{\theta \in \mathbb{R}^{|\mathcal{S}||\mathcal{A}|}} \tilde{v}^\lambda_{\pi_\theta}(\rho)$.*

**Entropy-regularized AC.** Actor critic with entropy regularization (`Ent-AC`) alternates between actor and critic updates. In this paper, we study the variant where at iteration $k$ the critic $\hat{q}_k$ is updated $H$ times, using the TD update (11). The regularized value and advantages are then updated as

$$\hat{v}_k(s) = \sum_{a \in \mathcal{A}} \pi_{\theta_k}(a|s) \left[ \hat{q}_k(s,a) - \lambda \log\left(\pi_{\theta_k}(a|s)\right) \right],$$
$$\hat{a}_k(s,a) = \hat{q}_k(s,a) - \lambda \log\left(\pi_{\theta_k}(a|s)\right) - \hat{v}_k(s) \tag{13}$$

where $\pi_{\theta_k}$ is the actor at iteration $k$, using softmax parameterization with parameter $\theta_k$. At the end of the iteration, the actor is updated using the gradient update (12).

We give the pseudo-code of the procedure in Algorithm 1. Given an initial distribution $\rho$ over states, we define the two sampling distributions $\nu^c(\theta; \cdot)$ and $\nu^a(\theta; \cdot)$ as, for $x = (s, a, \tilde{s}, \tilde{a}) \in \mathcal{S} \times \mathcal{A} \times \mathcal{S} \times \mathcal{A}$, and $y = (s, a) \in \mathcal{S} \times \mathcal{A}$,

$$\nu^c(\theta; x) \triangleq d^\theta_\rho(s)\pi_\theta(a|s)P(\tilde{s}|s,a)\pi_\theta(\tilde{a}|\tilde{s}) \ , \tag{14}$$

$$\nu^a(\theta; y) \triangleq d^\theta_\rho(s)\pi_\theta(a|s) \ . \tag{15}$$

These are respectively used to compute the TD update of the critic and stochastic actor gradients.

# 4. Analysis of `Ent-AC`: Exact Critic Case

In this section, we show that, when used with a perfect critic, entropy-regularized AC achieves *strong variance reduction*. Formally, this consists in setting $\hat{q}_k \equiv \tilde{q}^\lambda_{\theta_k}$ for all $k \geq 0$. To conduct our analysis, we first derive a bound on $\pi_{\min}$, the minimal value of the policy through the optimization. We then use $\pi_{\min}$ to link the variance of the stochastic actor gradient to the norm of its deterministic counterpart.

**Controlling $\pi_{\min}$.** To control $\pi_{\min}$, we project the current policy onto a smaller subspace, following Zhang et al.

(2021a); Labbi et al. (2026b). For any threshold $\tau > 0$ and policy $\pi$, we introduce $\mathcal{A}^\pi_\tau(s) \triangleq \{a \in \mathcal{A}, \pi(a|s) \leq \tau\}$, as well as the operator $\mathcal{U}_\tau$ which acts on $(s,a) \in \mathcal{S} \times \mathcal{A}$ as

$$\mathcal{U}_\tau(\pi)(a|s) \triangleq \begin{cases} \tau, & \text{if } \pi(a|s) \leq \tau, \\ \pi(a|s) - b_\pi(s), & \text{if } a = a^\pi_{\max}(s), \\ \pi(a|s), & \text{otherwise,} \end{cases}$$

where $b_\pi(s) \triangleq \sum_{a \in \mathcal{A}^\pi_\tau(s)}(\tau - \pi(a|s))$, and $a^\pi_{\max}(s) \triangleq \arg\max_{a \in \mathcal{A}}\{\pi(a|s)\}$, choosing at random in the $\arg\max$ in case of ties. We denote $\mathcal{T}_\tau$ the corresponding operator in the logit space, i.e. for all $\theta$, $\pi_{\mathcal{T}_\tau(\theta)} \triangleq \mathcal{U}_\tau(\pi_\theta)$.

This operator prevents policies from reaching policies with low entropy, i.e., from becoming too deterministic: for any $s, a \in \mathcal{S} \times \mathcal{A}$, if $\pi(a|s)$ approaches zero, $\mathcal{U}_\tau$ raises it above a $\tau$-dependent threshold. With a suitable choice of $\tau$, $\mathcal{T}_\tau$ yields logits with a higher regularized value.

**Lemma 4.** *Assume that $\rho$ satisfies $A_\rho$. Let $\tau_\lambda \triangleq \min\left(\frac{1}{3}\exp\left(-\frac{16+8\gamma\lambda\log(|\mathcal{A}|)}{\lambda(1-\gamma)^2\rho_{\min}}\right), \frac{1}{3^8|\mathcal{A}|^4}\right)$. Then, for any $\theta \in \mathbb{R}^{|\mathcal{S}||\mathcal{A}|}$ and for $\tilde{\theta} = \mathcal{T}_{\tau_\lambda}(\theta)$, it holds that $\tilde{v}^\lambda_{\tilde{\theta}}(\rho) \geq \tilde{v}^\lambda_\theta(\rho)$ and that for any $(s,a) \in \mathcal{S} \times \mathcal{A}, \pi_{\tilde{\theta}}(a|s) \geq \tau_\lambda$.*

Based on this lemma, we observe that for any parameter obtained by applying the improvement operator $\mathcal{T}_{\tau_\lambda}$, then $\pi_{\min}$ and the corresponding Łojasiewicz constant defined in Lemma 3 are bounded from below by

$$\pi_{\min} \geq \tau_\lambda \ , \quad \underline{\tilde{\mu}}_\lambda \triangleq \lambda(1-\gamma)\rho^2_{\min}\tau^2_\lambda/|\mathcal{S}|, \tag{16}$$

where $\tau_\lambda$ is defined in Lemma 4.

**Convergence of AC with exact critic.** To establish the convergence of AC with exact critic, we show that its gradient is unbiased, but most importantly that its variance can be linked to the norm of the *deterministic gradient*. This property is formalized in the following lemma.

**Lemma 5.** *Assume $A_\rho$ and assume that for all $(s,a) \in \mathcal{S} \times \mathcal{A}$, we have $\pi_\theta(a|s) \geq \pi_{\min} > 0$. For any $\theta \in \mathbb{R}^{|\mathcal{S}||\mathcal{A}|}$, it holds that*

$$\mathbb{E}_{Y \sim \nu^a(\theta)}\left[g^Y_a(\tilde{a}^\lambda_\theta)\right] = \frac{\partial \tilde{J}_\lambda(\theta)}{\partial \theta} \ ,$$

$$\mathbb{E}_{Y \sim \nu^a(\theta)}\left[\left\|g^Y_a(\tilde{a}^\lambda_\theta) - \frac{\partial \tilde{J}_\lambda(\theta)}{\partial \theta}\right\|^2_2\right] \leq \frac{(1-\gamma)^{-1}}{\pi_{\min}\rho_{\min}}\left\|\frac{\partial \tilde{J}_\lambda(\theta)}{\partial \theta}\right\|^2_2 \ .$$

*Proof sketch.* The key observation is that the variance of the gradient can be expressed as

$$\text{Var}(g^Y_a(\tilde{a}^\lambda_\theta)) = \frac{1}{(1-\gamma)^2}\sum_{(s',a') \in \mathcal{S} \times \mathcal{A}} d^\theta_\rho(s')\pi_\theta(a'|s')\delta_{s',a'} \ ,$$

where $\delta_{s',a'} = \tilde{a}^\lambda_\theta(s',a')^2 - d^\theta_\rho(s')\pi_\theta(a'|s')\tilde{a}^\lambda_\theta(s',a')^2$. Using $\min_{(s,a) \in \mathcal{S} \times \mathcal{A}}\pi_\theta(a|s) \geq \pi_{\min}$ to bound the terms

---

**Algorithm 1** `Ent-AC`: Entropy-regularized Actor-Critic

---

1: **Input:** stepsizes $\eta_a, \eta_c$; initial parameters $\hat{q}_{-1}$, and $\theta_0$; projection operator $\mathcal{T}$.
2: **for** $k = 0$ **to** $K - 1$ **do**
3:   Set $\hat{q}_k^0 = \hat{q}_{k-1}$ and sample $X_k = (X_k^h)_{h=1}^H$ where $X_k^h = (S_k^h, A_k^h, \tilde{S}_k^h, \tilde{A}_k^h)$ and $X_k^h \sim \nu^c(\theta_k, \cdot)$ (defined in (14)).
4:   **for** $h = 0$ **to** $H - 1$ **do**
5:     Compute the stochastic gradient for the critic's update $g_c^{X_k^{h+1}}(\theta_k, \hat{q}_k^h) \in \mathbb{R}^{|\mathcal{S}||\mathcal{A}|}$ defined with:

$$[g_c^{X_k^{h+1}}(\theta_k, \hat{q}_k^h)]_{s,a} = 1_{(s,a)}(S_k^{h+1}, A_k^{h+1})\delta(X_k^{h+1}, \theta_k, \hat{q}_k^h) \ , \quad \text{where} \tag{11}$$

$$\delta(X_k^{h+1}, \theta_k, \hat{q}_k^h) = \left[ r(S_k^{h+1}, A_k^{h+1}) + \gamma\left[\hat{q}_k^h(\tilde{S}_k^{h+1}, \tilde{A}_k^{h+1}) - \lambda\log(\pi_\theta(\tilde{A}_k^{h+1}|\tilde{S}_k^{h+1}))\right] - \hat{q}_k^h(S_k^{h+1}, A_k^{h+1})\right]$$

6:     Update estimate: $\hat{q}_k^{h+1} = \hat{q}_k^h + \eta_c\, g_c^{X_k^{h+1}}(\theta_k, \hat{q}_k^h)$ {# Update critic using Temporal Difference error}
7:   **end for**
8:   Set $\hat{q}_k = \hat{q}_k^H$. For all $(s,a) \in \mathcal{S} \times \mathcal{A}$, compute the regularized values and advantages using (13).
9:   Sample $Y_{k+1} = (S_{k+1}, A_{k+1}) \sim \nu^a(\theta_k, \cdot)$ (see (15)) and compute the stochastic gradient $g_a^{Y_{k+1}}(\hat{a}_k)$ defined with:

$$[g_a^{Y_{k+1}}(\hat{a}_k)]_{s,a} = 1_{(s,a)}(S_{k+1}, A_{k+1})(1-\gamma)^{-1}\hat{a}_k(S_{k+1}, A_{k+1}) \tag{12}$$

10:   Update the actor: $\theta_{k+1} = \mathcal{T}(\theta_k + \eta_a\, g_a^{Y_{k+1}}(\hat{a}_k))$.
11: **end for**

*(Left margin labels: **I: Critic** for lines 5–8, **II: Actor** for lines 9–10)*

---

$d_\rho^\theta(s')\pi_\theta(a'|s') \le (d_\rho^\theta(s')\pi_\theta(a'|s'))^2/((1-\gamma)\pi_{\min}\rho_{\min})$ and recognizing the expression of the gradient given in Lemma 1 gives the result. Full proof in Section B. □

This result proves that, when using the critic as a baseline, which is the actual value function of the current policy, the AC stochastic gradient's variance can be upper-bounded by the corresponding deterministic gradient's norm up to a multiplicative constant. Remarkably, this means that when the algorithm has converged, and the deterministic gradient is zero, AC does not have any remaining variance at all! This property can be leveraged to derive the following convergence result, which shows that when the critic is exact, AC converges to the optimal policy.

**Theorem 1.** *Assume that the initial distribution $\rho$ satisfies $A_\rho$. Fix $\eta_a \le (1-\gamma)\rho_{\min}\pi_{\min}/L$ and consider the iterates of* `Ent-AC` *with projection operator $\mathcal{T} = \mathcal{T}_{\tau_\lambda}$. It holds that $\min_{k\ge0}\min_{(s,a)\in\mathcal{S}\times\mathcal{A}} \pi_{\theta_k}(a|s) \ge \tau_\lambda$ almost surely. Additionally, for any $k \ge 0$ we have that*

$$\mathbb{E}\left[\tilde{J}_\lambda^\star - \tilde{J}_\lambda(\theta_k)\right] \le \left(1 - \eta_a\underline{\tilde{\mu}}_\lambda\right)^k \left(\tilde{J}_\lambda^\star - \tilde{J}_\lambda(\theta_0)\right) \ .$$

This theorem is a consequence of Lemma 5. To our knowledge, this result is the first to show that the Actor-Critic method *is a strong variance reduction method*. When using a perfect baseline, it converges linearly towards the optimal policy, matching the rates of deterministic methods.

**Corollary 1** (Sample complexity). *Under the same assumptions as Theorem 1, for any $\epsilon > 0$, it suffices to take*

$$K \ge \frac{L}{\underline{\tilde{\mu}}_\lambda}\frac{1}{(1-\gamma)\rho_{\min}\tau_\lambda}\log\left(\frac{\tilde{J}_\lambda^\star - \tilde{J}_\lambda(\theta_0)}{\epsilon}\right)$$

*iterations to guarantee $\tilde{J}_\lambda^\star - \mathbb{E}\left[\tilde{J}_\lambda(\theta_K)\right] \le \epsilon$.*

This is a direct consequence of Theorem 1. This shows that, akin to deterministic gradient methods, the sample complexity of AC is of order $\log(1/\epsilon)$. Crucially, this shows that using the critic as a baseline does not diminish the scale of the stochastic gradient's variance, but in fact completely removes its importance.

## 5. Analysis of `Ent-AC`: General Case

In practice, no oracles for the critic are available, and one has to learn it alongside the actor. In such a case, we show that *the variance of the actor does not matter*, and all the residual variance in the actor-critic algorithm comes from *estimating the critic*. Given a policy $\pi$ and an estimated regularized q-value q, we can construct an estimate of regularized advantage for any state action pair $(s,a)$ as

$$a(s,a) \triangleq q(s,a) - \lambda\log(\pi(a\mid s)) - v(s) \ ,$$
$$v(s) \triangleq \sum_{a\in\mathcal{A}}\pi(a|s)(q(s,a) - \lambda\log(\pi(a|s))) \ .$$

Using the estimated advantage a gives a gradient estimator $g_a^Y(a)$. Inexactness of the critic estimates has an impact on the bias and variance of this estimator, which we define as

$$b(\theta, a) \triangleq \|\bar{g}_a(\theta, a) - \frac{\partial\tilde{J}_\lambda(\theta)}{\partial\theta}\|_2^2 \ , \tag{17}$$

$$\mathrm{Var}(\theta, a) \triangleq \mathbb{E}_{Y\sim\nu^a(\theta)}\left[\|g_a^Y(a) - \bar{g}_a(\theta, a)\|_2^2\right] \ , \tag{18}$$

where $\bar{g}_a(\theta, a) \triangleq \mathbb{E}_{Y\sim\nu^a(\theta)}[g_a^Y(a)]$. Next, we give a counterpart of Lemma 5 with an inexact critic, bounding the bias and variance of the current estimator.

**Lemma 6.** *Fix $\theta \in \mathbb{R}^{|\mathcal{S}||\mathcal{A}|}$ and $\mathrm{a} \in \mathbb{R}^{|\mathcal{S}||\mathcal{A}|}$. It holds that*

$$\mathrm{b}(\theta,\mathrm{a}) = \frac{1}{(1-\gamma)^2} \sum_{(s,a)\in\mathcal{S}\times\mathcal{A}} d_\rho^\theta(s)^2 \pi_\theta(a|s)^2 (\mathrm{a}(s,a) - \tilde{\mathrm{a}}_\theta^\lambda(s,a))^2,$$

$$\mathrm{Var}(\theta,\mathrm{a}) \leq \frac{1}{(1-\gamma)^2} \sum_{(s,a)\in\mathcal{S}\times\mathcal{A}} d_\rho^\theta(s) \pi_\theta(a|s) \mathrm{a}(s,a)^2 \ .$$

We provide a proof in Section C. While this lemma is very similar to its exact critic counterpart, it differs in a fundamental way: the advantages are replaced by the estimated advantages. Consequently, having a biased critic ends up creating bias in the gradient estimator itself, directly depending on the difference between the estimated advantage and its true value. However, the variance can still be bounded using the norm of the deterministic gradient and the bias directly when the minimal probability of the policy is uniformly lower-bounded. In order to guarantee that this coefficient remains uniformly lower bounded, we restrict the optimization to a smaller subspace, eliminating policies for which the regularization is too strong. By proceeding in this way, we guarantee that the minimal probability stays uniformly lower-bounded.

Next, we derive the recursions for the actor's and critic's updates separately, then combine them to obtain the final convergence rate. Before that, we define the filtration adapted to the iterates of the actor:

$$\mathcal{F}_k \triangleq \sigma\Big(X_\ell : \ell \in \{0, \ldots, k-1\}, Y_\ell : \ell \in \{1, \ldots, k\}\Big) \ .$$

**Updating the Actor.** Below, we derive a recursion on the actor updates with a full proof in Section C.

**Lemma 7** (Actor recursion). *Assume $A_\rho$ and consider the iterates of* `Ent-AC` *with projection operator $\mathcal{T} = \mathcal{T}_{\tau_\lambda}$. It holds that $\min_{k\geq 0}\min_{(s,a)\in\mathcal{S}\times\mathcal{A}} \pi_{\theta_k}(a|s) > \tau_\lambda$ almost surely. For any $k \geq 0$, we also have that*

$$\mathbb{E}\left[\tilde{J}_\lambda^\star - \tilde{J}_\lambda(\theta_{k+1}) \big| \mathcal{F}_k\right]$$

$$\leq \tilde{J}_\lambda^\star - \tilde{J}_\lambda(\theta_k) - \left(\frac{\eta_\mathrm{a}}{2} - \frac{2L\eta_\mathrm{a}^2}{(1-\gamma)\rho_{\min}\tau_\lambda}\right)\|\nabla\tilde{J}_\lambda(\theta_k)\|_2^2$$

$$+ \eta_\mathrm{a}\frac{2\mathrm{b}(\theta_k\mathbb{E}[\hat{\mathrm{a}}_k(s,a)|\mathcal{F}_k])}{(1-\gamma)^2} + L\eta_\mathrm{a}^2\frac{2\mathbb{E}[\mathrm{Var}(\theta_k,\hat{\mathrm{a}}_k)|\mathcal{F}_k]}{(1-\gamma)^2} \ ,$$

*where $\mathrm{b}(\theta,\mathrm{a})$ and $\mathrm{Var}(\theta,\mathrm{a})$ are defined in (17) and (18) respectively.*

This recursion captures that the policy improvement at each actor step is controlled entirely by the critic's *bias* (17) and *variance* (18). In particular, the stochasticity of the actor update is fully absorbed into the descent term $\|\nabla\tilde{J}_\lambda(\theta_k)\|_2^2$. Moreover, when the critic is learned exactly, both the bias and variance terms vanish, and we recover the linear convergence regime of Theorem 1. This stands in sharp contrast to recent actor–critic analyses, e.g. (Kumar et al.,

2024), which do not establish variance reduction and do not transfer the critic's estimation error to the actor's variance. Thus, the central challenge is to obtain a sufficiently accurate critic. Next, we bound the critic's mean-squared error.

**Updating the Critic.** Before deriving a bound on the Mean Squared error of the critic, we need to:(1) establish sufficient state-action exploration; (2) track how the switch of policy affects the switch of the critic target, which is what we do subsequently. The state-action exploration is often used as an assumption in prior analysis of actor-critic, see e.g. (Kumar et al., 2024; Chen & Zhao, 2023). Additionally, we emphasize that assuming such a property in the unregularized setting is contradictory, as it requires maintaining exploratory policies during the learning process while the goal is to learn a deterministic policy. Thanks to the regularization and the projection operator, we relax this assumption. *Using only the sufficient state-exploration, we establish the sufficient state-action exploration condition*. The proof of the following lemma and all subsequent results are provided in Section D.

**Lemma 8.** *Assume $A_\rho$. For $k \geq 0$, $v \in \mathbb{R}^{|\mathcal{S}||\mathcal{A}|}$, it holds*

$$\langle \mathrm{D}_{\theta_k}(\mathrm{Id} - \gamma\widetilde{\mathrm{P}}_{\theta_k})v, v\rangle \geq \frac{1}{2}(1-\gamma)^2\rho_{\min}\tau_\lambda \|v\|_2^2 \ ,$$

*where $\mathrm{D}_\theta \triangleq \mathrm{diag}((d_\rho^\theta(s)\pi_\theta(a|s))_{s,a})$ and $\widetilde{\mathrm{P}}_{\theta_k}$ is a matrix of size $|\mathcal{S}||\mathcal{A}| \times |\mathcal{S}||\mathcal{A}|$ defined for $(s,a,\tilde{s},\tilde{a}) \in \mathcal{S} \times \mathcal{A} \times \mathcal{S} \times \mathcal{A}$, by $\widetilde{\mathrm{P}}_\theta(\tilde{s},\tilde{a}|s,a) \triangleq \mathrm{P}(\tilde{s}|s,a)\pi_\theta(\tilde{a}|\tilde{s})$.*

Next, we bound the distance between the regularized q-functions of two successive policies.

**Lemma 9.** *Assume $A_\rho$. It holds that*

$$\left\|\tilde{\mathrm{q}}_{\theta_{k+1}}^\lambda - \tilde{\mathrm{q}}_{\theta_k}^\lambda\right\|_2 \leq \tilde{C}_\lambda\eta_\mathrm{a}\left|\hat{\mathrm{a}}_k(S_{k+1}, A_{k+1})\right| \ ,$$

*where $\tilde{C}_\lambda$ is a coefficient that depends only on the problem parameters (see Corollary 4 for the exact expression).*

Finally, we derive a bound on the MSE of the critic.

**Lemma 10.** *Assume $A_\rho$ and assume that $\eta_\mathrm{c} \leq (1-\gamma)^2\rho_{\min}\tau_\lambda/40$ and $H \geq \frac{2}{\eta_\mathrm{c}\tilde{\mu}_\mathrm{c}}\log(2 + 4\tilde{C}_\lambda^2\eta_\mathrm{a}^2)$. For any $k \geq 0$, it holds that*

$$\mathbb{E}\left[\|\hat{\mathrm{q}}_k - \tilde{\mathrm{q}}_{\theta_k}^\lambda\|_2^2\right] \lesssim (1 - \eta_\mathrm{c}\tilde{\mu}_\mathrm{c})^{H(k+1)/2}\|\hat{\mathrm{q}}_{-1} - \tilde{\mathrm{q}}_{\theta_0}^\lambda\|_2^2$$

$$+ \frac{\tilde{C}_\lambda^2\eta_\mathrm{a}^2(1+\lambda^2\log(|\mathcal{A}|)^2)}{(1-\gamma)^2}\frac{1}{1-(1-\eta_\mathrm{c}\tilde{\mu}_\mathrm{c})^{H/2}} + \frac{\eta_\mathrm{c}\sigma_\mathrm{c}^2}{\tilde{\mu}_\mathrm{c}} \ ,$$

*where $\tilde{\mu}_\mathrm{c} \triangleq (1-\gamma)^2\rho_{\min}\tau_\lambda/2$, and the variance term is defined as $\sigma_\mathrm{c}^2 \triangleq \frac{36 + 4\lambda^2 + 36\lambda^2\log(|\mathcal{A}|)^2}{(1-\gamma)^2}$.*

The preceding lemma shows that, with $H$ TD steps per actor update, the critic tracks the moving target $\tilde{q}_{\theta_k}^\lambda$: the MSE

$\mathbb{E}\|\hat{q}_k - \tilde{q}_{\theta_k}^\lambda\|_2^2$ contracts geometrically from initialization up to a steady state. The residual error decomposes into a drift term $O(\eta_a^2)$ (due to the actor moving the target) and a stochastic TD floor $O(\eta_c)$; taking $H$ sufficiently large preserves accurate critic estimation along the actor trajectory.

**Remark 1** (Choice of the projection operator.). *Note that the choice of the projection operator in Algorithm 1 is crucial for our theoretical analysis. Specifically, we stress that despite similarities with the projection operator defined in (Zhang et al., 2021a; Labbi et al., 2026b), our operator is different and better suited for actor-critic. Indeed, this projection results in less aggressive policy shifts: defining the set $\Pi_\tau \triangleq \{\pi, \text{ such that for all } (s,a) \in \mathcal{S} \times \mathcal{A}, \pi(a|s) \geq \tau\}$, we remark that for any $\pi_1 \notin \Pi_\tau$ and $\pi_2 \in \Pi_\tau$,*

$$\|\pi_1 - \mathcal{U}_\tau(\pi_1)\|_1 \leq \|\pi_1 - \pi_2\|_1 \quad .$$

*As such, $\mathcal{U}_\tau$ is indeed a projection on $\Pi_\tau$. In contrast, the operator defined in Zhang et al. (2021a) would yield a policy switch in $L_1$ norm of at least $\tau_\lambda/2$, inducing strong bias in the critic. With our operator, the switch $\|\pi_{\theta_{k+1}} - \pi_{\tilde{\theta}_{k+1}}\|_1$ is bounded by $\|\pi_{\tilde{\theta}_{k+1}} - \pi_{\theta_k}\|_1$. This allows the critic at a given time step to benefit from the warm start provided by the critic from the previous step.*

**Convergence of Actor-Critic.** Combining the two previous recursions allows us to get the following convergence rate for `Ent-AC`. The proof is provided in Section E.

**Theorem 2.** *Assume $A_\rho$ and assume that $\eta_c \leq (1-\gamma)^2\rho_{\min}\tau_\lambda/40$, $H \geq \frac{2}{\eta_c\tilde{\mu}_c}\log(2 + 4\tilde{C}_\lambda^2\eta_a^2)$, and that $\eta_a \leq \frac{(1-\gamma)\rho_{\min}\tau_\lambda}{8L}$. For any $K \geq 0$, it holds that*

$$\mathbb{E}\left[\tilde{J}_\lambda^\star - \tilde{J}_\lambda(\theta_K)\right] \lesssim \left(1 - \frac{\eta_a\tilde{\mu}_\lambda}{8}\right)^K \left[\tilde{J}_\lambda^\star - \tilde{J}_\lambda(\theta_0)\right]$$

$$+ \frac{L\eta_a^2 K}{(1-\gamma)^2}\max\left(1 - \frac{\eta_a\tilde{\mu}_\lambda}{8}, (1-\eta_c\tilde{\mu}_c)^{H/2}\right)^K \|\hat{q}_{-1} - \tilde{q}_{\theta_0}^\lambda\|_2^2$$

$$+ \frac{(1-\eta_c\tilde{\mu}_c)^H}{\tilde{\mu}_\lambda(1-\gamma)^2}B + \eta_a^3\frac{\tilde{C}_\lambda^2 L(1+\lambda^2\log(|\mathcal{A}|)^2)}{\tilde{\mu}_\lambda(1-\gamma)^4} + \frac{L\eta_a\eta_c\sigma_c^2}{(1-\gamma)^2\tilde{\mu}_c\tilde{\mu}_\lambda} \quad ,$$

*where $B$ is a constant that depends only on the problem parameters, whose complete expression is provided in (31).*

The previous bound clearly separates the contribution of four effects. First, the first two terms quantify the *geometric forgetting* of the initialization error: the suboptimality contracts at a linear rate, and the influence of the initial critic mismatch is washed out at the slower of the actor and critic contraction factors. Second, the term $\frac{(1-\eta_c\tilde{\mu}_c)^H}{\tilde{\mu}_\lambda(1-\gamma)^2}B$ captures the *bias of the critic*, which arises when the inner TD loop is not run long enough to accurately track the current policy. Third, the bound contains a *policy-switch* contribution of order $\tilde{O}(\eta_a^3)$, which accounts for the higher-order cost induced by changing the target that the critic

must estimate. Finally, the remaining term corresponds to the *variance of the critic* (scaling like $\eta_a\eta_c$), i.e., the stochastic TD noise floor propagated to the actor.

A key takeaway is that the actor's own sampling variance does not appear explicitly: it is fully absorbed by the descent term and is effectively replaced by the critic's *bias* and *variance*. Consequently, if $H$ is too small, the multiplicative factor $(1 - \eta_c\tilde{\mu}_c)^H$ remains sizable and the resulting bias term can prevent convergence to the optimal solution. In contrast, when $H$ is sufficiently large, the critic bias becomes negligible, and the limiting behavior is dominated by the variance floor. This suggests choosing $H$ large enough to remove the bias, after which additional critic steps mainly improve the transient but do not change the asymptotic noise-dominated regime. Next, we derive the sample complexity of the entropy-regularized problem

**Corollary 2.** *Assume $A_\rho$. Let $\epsilon > 0$, and set*

$$\eta_a \lesssim \min\left(\frac{(1-\gamma)\rho_{\min}\tau_\lambda}{L}, \frac{\tilde{\mu}_\lambda^{1/3}(1-\gamma)^{4/3}\epsilon^{1/3}\tilde{C}_\lambda^{-2/3}}{L^{1/3}(1+\lambda\log(|\mathcal{A}|))^{2/3}}, \frac{\tilde{\mu}_c\tilde{\mu}_\lambda\epsilon}{L\sigma_c^2\rho_{\min}\tau_\lambda}\right),$$

*as well as $\eta_c \lesssim (1-\gamma)^2\rho_{\min}\tau_\lambda$. In this case, `Ent-AC`, achieves $\mathbb{E}\left[\tilde{J}_\lambda^\star - \tilde{J}_\lambda(\theta_K)\right] \leq \epsilon$, with a number of critic updates per actor update of*

$$H \gtrsim \frac{\max\left\{\log\left(1 + \frac{\tilde{C}_\lambda^2(1-\gamma)^2\rho_{\min}^2\tau_\lambda^2}{L^2}\right), \log\left(\frac{B}{\tilde{\mu}_\lambda(1-\gamma)^2\epsilon}\right)\right\}}{(1-\gamma)^2\rho_{\min}\tau_\lambda\tilde{\mu}_c},$$

*and a total number of actor updates of*

$$K \gtrsim \max\left(\frac{L\tilde{\mu}_\lambda^{-1}\tau_\lambda^{-1}}{(1-\gamma)\rho_{\min}}, \frac{\tilde{C}_\lambda^{2/3}L\left(1+\lambda\log(|\mathcal{A}|)\right)^{2/3}}{\tilde{\mu}_\lambda^{5/3}(1-\gamma)^{4/3}\epsilon^{1/3}}, \frac{L\sigma_c^2\rho_{\min}\tau_\lambda}{\tilde{\mu}_c\tilde{\mu}_\lambda^2\epsilon}\right)$$

$$\times \max\left\{\log\left(\frac{(\tilde{J}_\lambda^\star - \tilde{J}_\lambda(\theta_0))}{\epsilon}\right), \log\left(\frac{\rho_{\min}\tau_\lambda\|\hat{q}_{-1} - \tilde{q}_{\theta_0}^\lambda\|_2^2}{(1-\gamma)\tilde{\mu}_\lambda\epsilon}\right)\right\},$$

*where $B$ is a constant that depends only on the problem parameters, whose complete expression is provided in (31).*

Corollary 2 shows that the *actor* (outer) loop enjoys the standard "linear-with-noise" complexity i.e., geometric contraction up to a variance-limited $1/\epsilon$ regime driven by the *critic* noise $\sigma_c^2$. Moreover, the *inner* critic loop needs only $H = \tilde{O}(\log(1/\epsilon))$ TD updates per actor step to reduce the critic bias below $\epsilon$; beyond this, extra critic steps mainly improve transients without affecting the asymptotic rate.

## 6. Experiments

In this section, we evaluate the empirical performance of `Ent-AC` across multiple tasks. We focus on how the approximation of the regularized value function, controlled by the number of critic steps $H$, impacts the convergence and stability of the actor's policy. We compare our learned critic configurations against an "Exact Critic"

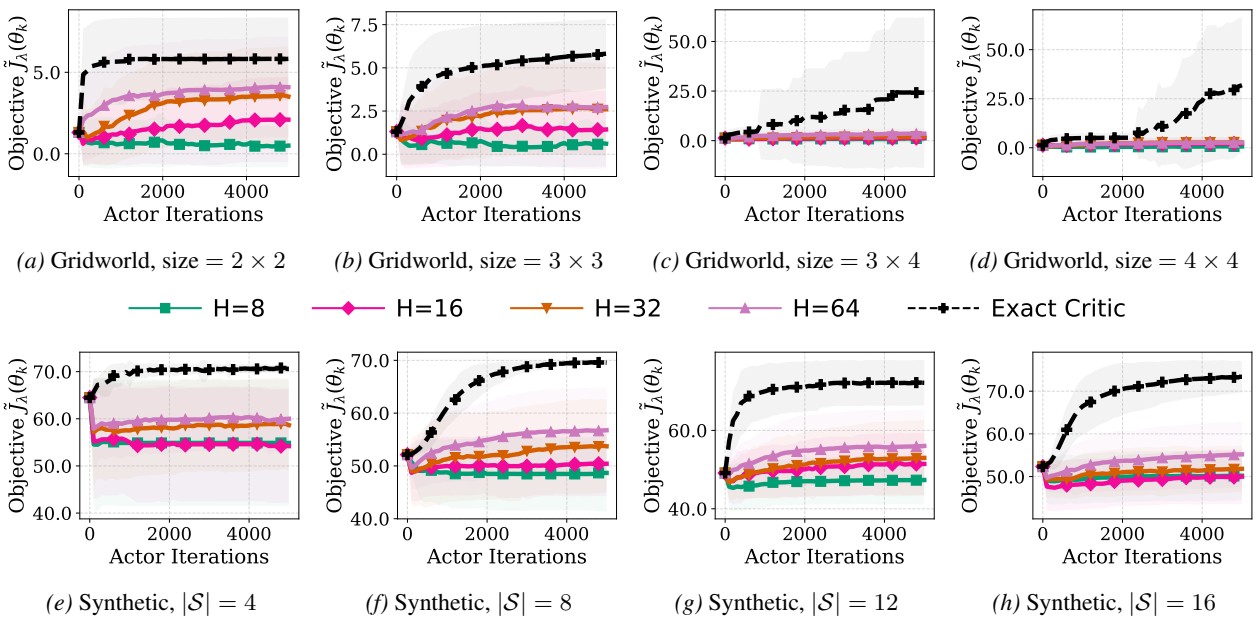

*Figure 1.* **Performance of `Ent-AC` across Gridworld and Synthetic environments.** We report the mean objective value $\tilde{J}_\lambda(\theta_k)$ as a function of actor iterations for varying critic update frequencies $H \in \{8, 16, 32, 64\}$. Panels (a)–(d) show results for tabular Gridworld layouts of increasing scale, while panels (e)–(h) illustrate performance on synthetic MDPs with varying state space sizes $|\mathcal{S}|$ and fixed action space of size $|\mathcal{A}| = 4$. The "Exact Critic" baseline (dashed black line) represents an oracle reference obtained by solving the critic to optimality. Shaded regions denote one standard deviation across 50 independent random seeds. Increasing $H$ consistently reduces the approximation gap relative to the exact critic, which greatly enhances the performance of the algorithm.

oracle to establish a performance upper bound across varying environment complexities. The code is available online at https://github.com/Labbi-Safwan/Actor-Critic. We describe below the two environments that we will use in the experiments.

**Experimental Setup.** We start by describing the two environments we use, as well as the algorithmic setup.

*(Synthetic ([Zheng et al., 2023](#)).)* The synthetic environment is generated by sampling a dense tabular model $(\mathsf{P}, \mathsf{r}, \rho)$. For each $(s, a) \in \mathcal{S} \times \mathcal{A}$, the transition kernel $\mathsf{P}(\cdot|s, a)$ is drawn uniformly at random from the $|\mathcal{S}|$-dimensional simplex, so that each action induces a distribution over all next states. Rewards are sampled independently as $R(s, a) \sim \mathrm{Unif}[0, 1]$ and returned deterministically given $(s, a)$, and the initial distribution is set to $\rho(s) = 1/|\mathcal{S}|$. We evaluate this environment in the discounted setting with $\gamma = 0.99$ for four state-space sizes $|\mathcal{S}| \in \{4, 8, 12, 16\}$, with a fixed action set of size $|\mathcal{A}| = 4$.

*(Gridworld ([Domingues et al., 2021](#)).)* We evaluate our method on a suite of tabular Gridworld layouts ([Domingues et al.](#), 2021) with dimensions $M \times N \in \{2 \times 2, 3 \times 3, 3 \times 4, 4 \times 4\}$. Each environment is modeled as a finite MDP where the state space $\mathcal{S}$ consists of discrete grid coordinates and the action space $\mathcal{A}$ comprises the four cardinal directions. Transitions are deterministic; an action $a \in \mathcal{A}$ moves the agent to the adjacent cell in the specified direction, or

leaves the agent's position unchanged if the move targets a boundary. The agent is initialized at the bottom-left coordinate and must navigate to a goal state at the top-right of the grid. The environment employs a sparse reward signal where the agent receives $R = 1$ only upon goal reaching, and receives $R = 0$ in any other case.

*(Algorithmic Setup.)* To ensure a rigorous comparison across different values of $H \in \{8, 16, 32, 64\}$, we performed a grid search over actor and critic learning rates, $\eta_{\mathsf{a}}, \eta_{\mathsf{c}} \in \{0.003, 0.01, 0.03, 0.1\}$. The regularization parameter was fixed at $\lambda = 0.05$ for all experiments. In addition to evaluating `Ent-AC`, we include an "ideal critic" baseline, where the critic is set to the exact regularized value, to characterize the performance upper bound and highlight the impact of the critic's bias and variance. We report the mean objective value $\tilde{J}_\lambda(\theta_k)$ over 5000 actor iterations, with shaded regions representing one standard deviation over 50 independent runs.

**AC with exact critic converges fast.** When the critic is exactly known, AC can leverage this strong baseline to consistently learn fast across all eight configurations of Gridworld (Figures [1a](#) to [1d](#)) and Synthetic MDP (Figures [1e](#) to [1h](#)). This is in line with our theory, which shows that AC enjoys performance comparable to using deterministic gradients when the critic is perfectly known.

**It pays off to learn the critic.** Our experiments reveal a consistent trend: AC's performance is strictly monotonic with respect to the number of critic steps $H$. In all scenarios in Figure 1, increasing $H$ from $8$ to $64$ leads to faster convergence and higher objective values. This phenomenon is particularly pronounced in the MDPs of smaller sizes, where lower $H$ values (e.g., $H = 8$) often result in a significant performance gap compared to the oracle. This can be explained through the lens of critic bias and variance; when $H$ is small, the critic's estimate of the regularized q-value $\tilde{q}^\lambda_{\theta_k}$ remains "cold" and fails to converge to the fixed point of the regularized Bellman operator. In contrast, when $H$ is larger, we obtain a more precise q-value estimate, enabling more accurate policy updates. Overall, while increasing $H$ incurs a higher computational cost per actor iteration, it consistently leads to superior policy performance: *learning the critic pays off*.

## 7. Conclusion

We established novel global convergence rates for actor-critic in entropy-regularized reinforcement learning, with a specific focus on the variance-reduction phenomenon. First, we proved that, when using a perfect critic as a baseline, AC achieves $O(\log(1/\epsilon))$ sample complexity. To our knowledge, this is the first result proving that AC enjoys a *strong variance-reduction* property, akin to methods like SVRG (Johnson & Zhang, 2013) or SAGA (Defazio et al., 2014) in stochastic optimization. When no perfect critic is available, we show that most of the complexity of the algorithm amounts to learning the critic, and obtain the first $O(1/\epsilon)$ rates for AC. Our results shed new light on AC, confirming previous empirical evidence that estimation of the critic is crucial for good performance. These results open new perspectives for AC, where it remains unknown whether its properties remain beyond the tabular case. A promising research direction is to extend our results to the unregularized case and try to achieve faster rates by combining our approach with acceleration techniques for faster estimation of the critic and actor simultaneously.

## Impact Statement

This paper presents work whose goal is to advance the field of Machine Learning. There are many potential societal consequences of our work, none which we feel must be specifically highlighted here.

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

## A. Notations

**Distribution of the state-action sequence.** The state–action sequence $(S_t, A_t)_{t \geq 0}$ defines a stochastic process on the canonical space $(\mathcal{S} \times \mathcal{A})^{\mathbb{N}}$. For any initial state $s_0 \in \mathcal{S}$, we denote by $\mathbb{P}^{\pi}_{s_0}$ the law of this process. That is, for any $n \in \mathbb{N}$ and any subset $B \subset (\mathcal{S} \times \mathcal{A})^n$,

$$\mathbb{P}^{\pi}_{s_0}(B) = \sum_{(a_0,\ldots,a_{n-1}) \in \mathcal{A}^n} \sum_{(s_1,\ldots,s_{n-1}) \in \mathcal{S}^{n-1}} \mathbb{1}_B\big((s_0, a_0), \ldots, (s_{n-1}, a_{n-1})\big) \prod_{i=0}^{n-1} \pi(a_i \mid s_i) \, \mathsf{P}(s_{i+1} \mid s_i, a_i),$$

with the convention $s_0$ is the given initial state. We denote by $\mathbb{E}^{\pi}_{s_0}$ the corresponding expectation operator. In particular, the state sequence $(s_t)_{t \geq 0}$ defines a Markov reward process (Section 2.1.6 in (Puterman, 1994)) with transition kernel

$$\mathsf{P}_{\pi}(s' \mid s) = \sum_{a \in \mathcal{A}} \mathsf{P}(s' \mid s, a) \, \pi(a \mid s) \ .$$

**Norms.** For $x \in \mathbb{R}^d$, we define the norms

$$\|x\|_{\infty} = \max_{i \in \{1,\ldots,d\}} |x_i| \ , \quad \|x\|_1 = \sum_{i=1}^d |x_i| \ , \quad \|x\|_2 = \left( \sum_{i=1}^d |x_i|^2 \right)^{1/2} \ .$$

For a $d \times d$ matrix $M$, we denote by $\|M\|_{\infty}, \|M\|_1$, and $\|M\|_2$ respectively the max *row* sum, the max *column* sum, and the spectral norm:

$$\|M\|_{\infty} = \sup_{x \neq 0}\{\|Mx\|_{\infty}/\|x\|_{\infty}\} = \sup_{i \in \{1,\ldots,d\}} \sum_{j=1}^d |M_{i,j}| \ , \quad \|M\|_1 = \sup_{x \neq 0}\{\|Mx\|_1/\|x\|_1\} = \sup_{j \in \{1,\ldots,d\}} \sum_{i=1}^d |M_{i,j}|, \quad (19)$$

$$\|M\|_2 = \sup_{x \neq 0}\{\|Mx\|_2/\|x\|_2\} \ . \quad (20)$$

Recall that, for any $x \in \mathbb{R}^d$, $\|Mx\|_{\infty} \leq \|M\|_{\infty}\|x\|_{\infty}$ and $\|Mx\|_2 \leq \|M\|_2\|x\|_2$.

**KL divergence.** For two discrete probability distributions $p = (p_i)_{i=1}^n$ and $q = (q_i)_{i=1}^n$ on a finite set $\{1, \ldots, n\}$, the Kullback–Leibler (KL) divergence from $q$ to $p$ is defined as

$$\mathrm{KL}(p\|q) := \sum_{i=1}^n p_i \log\left(\frac{p_i}{q_i}\right),$$

with the convention that $0\log(0/q_i) = 0$ and $\mathrm{KL}(p\|q) = +\infty$ if there exists $i$ such that $p_i > 0$ and $q_i = 0$.

**Functional and matrix forms.** For notational convenience, we also view $\mathsf{P}$ as a $(|\mathcal{S}| \cdot |\mathcal{A}|) \times |\mathcal{S}|$ matrix with entries $\mathsf{P}_{(s,a),s'} = \mathsf{P}(s' \mid s, a)$. Similarly, $\tilde{\mathsf{v}}^{\lambda}_{\pi}$ is a vector of size $|\mathcal{S}|$ and $\tilde{\mathsf{q}}^{\lambda}_{\pi}$ a vector of size $|\mathcal{S}| \times |\mathcal{A}|$. Finally, we identify the parameter $\theta \in \mathbb{R}^{\mathcal{S} \times \mathcal{A}}$ with its vector representation $\theta \in \mathbb{R}^{|\mathcal{S}||\mathcal{A}|}$, indexed by $(s, a) \in \mathcal{S} \times \mathcal{A}$. This slight abuse of notation allows us to conveniently switch between functional and matrix views.

## B. Convergence with exact critic

For any $\mathsf{a} \in \mathbb{R}^{|\mathcal{S}||\mathcal{A}|}$ and $y = (s, a) \in \mathcal{S} \times \mathcal{A}$, we define

$$[\mathsf{g}^y_{\mathsf{a}}(a)]_{(x,b)} \overset{\Delta}{=} \mathbb{1}_{(x,b)}(s, a) \, (1 - \gamma)^{-1} \, \mathsf{a}(s, a)$$

In the exact critic setting, the update of `Ent-AC` simplifies to:

$$\theta_{k+1} = \mathcal{T}\big(\theta_k + \eta_{\mathsf{a}} \, \mathsf{g}^{Y_{k+1}}_{\mathsf{a}}(\tilde{\mathsf{a}}^{\lambda}_{\theta_k})\big) \ ,$$

where $Y_{k+1} \sim \nu^{\mathsf{a}}(\theta k + 1)$. We preface the proof of the convergence of `Ent-AC` in this setting, with the following key lemma:

**Lemma 11.** *Assume* $\mathbf{A}_\rho$ *and that for all* $(s,a) \in \mathcal{S} \times \mathcal{A}$, *we have* $\pi_\theta(a|s) \geq \pi_{\min} > 0$. *For any* $\theta \in \mathbb{R}^{|\mathcal{S}||\mathcal{A}|}$, *it holds that*

$$\mathbb{E}_{Y \sim \nu^{\mathsf{a}}(\theta)} \left[ \mathrm{g}_{\mathsf{a}}^Y(\tilde{\mathrm{a}}_\theta^\lambda) \right] = \frac{\partial \tilde{J}_\lambda(\theta)}{\partial \theta} \quad , \quad \mathbb{E}_{Y \sim \nu^{\mathsf{a}}(\theta)} \left[ \left\| \mathrm{g}_{\mathsf{a}}^Y(\tilde{\mathrm{a}}_\theta^\lambda) - \frac{\partial \tilde{J}_\lambda(\theta)}{\partial \theta} \right\|_2^2 \right] \leq \frac{1}{(1-\gamma)\pi_{\min}\rho_{\min}} \left\| \frac{\partial \tilde{J}_\lambda(\theta)}{\partial \theta} \right\|_2^2 \;,$$

*where* $\nu^{\mathsf{a}}(\theta)$ *is defined in* (15)

*Proof.* Firstly, note that for any $(s,a) \in \mathcal{S} \times \mathcal{A}$, we have

$$\left[ \mathbb{E}_{Y \sim \nu^{\mathsf{a}}(\theta)} \left[ \mathrm{g}_{\mathsf{a}}^Y(\tilde{\mathrm{a}}_\theta^\lambda) \right] \right]_{s,a} = \sum_{s',a'} d_\rho^\theta(s')\pi_\theta(a'|s')\mathbf{1}_{(s,a)}(s',a')(1-\gamma)^{-1}\tilde{\mathrm{a}}_\theta^\lambda(s',a') = \frac{\partial \tilde{J}_\lambda(\theta)}{\partial \theta(s,a)} \;,$$

where the last equality follows from Lemma 1. Next, we have

$$\mathbb{E}_{Y \sim \nu^{\mathsf{a}}(\theta)} \left[ \left\| \mathrm{g}_{\mathsf{a}}^Y(\tilde{\mathrm{a}}_\theta^\lambda) - \frac{\partial \tilde{J}_\lambda(\theta)}{\partial \theta} \right\|_2^2 \right] = \mathbb{E}_{Y \sim \nu^{\mathsf{a}}(\theta)} \left[ \left\| \mathrm{g}_{\mathsf{a}}^Y(\tilde{\mathrm{a}}_\theta^\lambda) \right\|_2^2 \right] - \left\| \frac{\partial \tilde{J}_\lambda(\theta)}{\partial \theta} \right\|_2^2$$

$$= \frac{1}{(1-\gamma)^2} \sum_{(s,a) \in \mathcal{S} \times \mathcal{A}} d_\rho^\theta(s)\pi_\theta(a|s) \left[ \tilde{\mathrm{a}}_\theta^\lambda(s,a)^2 - d_\rho^\theta(s)\pi_\theta(a|s)\tilde{\mathrm{a}}_\theta^\lambda(s,a)^2 \right] \;.$$

Using $\mathbf{A}_\rho$ combined with the assumption $\min_{(s,a) \in \mathcal{S} \times \mathcal{A}} \pi_\theta(a|s) \geq \pi_{\min}$, and Lemma 1 concludes the proof. $\qquad\square$

In the following, we define the filtration adapted to the iterates of `Ent-AC` as

$$\mathcal{F}_k \overset{\Delta}{=} \sigma\Big(Y_k : k \in \{1, \ldots, K\}\Big) \;.$$

**Theorem 3.** *Assume* $\mathbf{A}_\rho$. *Fix* $\eta_{\mathsf{a}} \leq (1-\gamma)\rho_{\min}\pi_{\min}/L$ *and consider the iterates of* `Ent-AC`. *It holds that* $\min_{(s,a) \in \mathcal{S} \times \mathcal{A}} \pi_{\theta_k}(a|s) > \tau_\lambda$ *almost surely. Addtionnally, for any* $k \geq 0$ *we have that*

$$\tilde{J}_\lambda^\star - \mathbb{E}\left[ \tilde{J}_\lambda(\theta_k) \right] \leq \left[ 1 - \eta_{\mathsf{a}}\underline{\tilde{\mu}}_\lambda \right]^k \left( \tilde{J}_\lambda^\star - \tilde{J}_\lambda(\theta_0) \right) \;.$$

*Proof.* Define $\tilde{\theta}_{k+1} = \theta_k + \eta_{\mathsf{a}} \, \mathrm{g}_{\mathsf{a}}^{Y_{k+1}}(\tilde{\mathrm{a}}_{\theta_k}^\lambda)$. Using the monotone improvement property of $\mathcal{T}$, provided in Lemma 21, we have that

$$\tilde{J}_\lambda(\theta_{k+1}) = \tilde{J}_\lambda(\mathcal{T}(\tilde{\theta}_{k+1})) \geq \tilde{J}_\lambda(\tilde{\theta}_{k+1}) \;.$$

Next, applying Lemma 2, combined with Lemma 23 yields

$$\tilde{J}_\lambda(\tilde{\theta}_{k+1}) \geq \tilde{J}_\lambda(\theta_k) + 2\eta_{\mathsf{a}}\langle \nabla \tilde{J}_\lambda(\theta_k), \mathrm{g}_{\mathsf{a}}^{Y_{k+1}}(\tilde{\mathrm{a}}_{\theta_k}^\lambda)\rangle - \frac{\eta_{\mathsf{a}}^2 L}{2} \left\| \mathrm{g}_{\mathsf{a}}^{Y_{k+1}}(\tilde{\mathrm{a}}_{\theta_k}^\lambda) \right\|_2^2 \;.$$

Combining the two previous bounds, and taking the conditional expectation with respect to $\mathcal{F}_k$ gives

$$\mathbb{E}\left[ \tilde{J}_\lambda(\theta_{k+1})|\mathcal{F}_k \right] \geq \tilde{J}_\lambda(\theta_k) + 2\eta_{\mathsf{a}} \left\| \nabla \tilde{J}_\lambda(\theta_k) \right\|_2^2 - \frac{\eta_{\mathsf{a}}^2 L}{2} \mathbb{E}\left[ \left\| \mathrm{g}_{\mathsf{a}}^{Y_{k+1}}(\tilde{\mathrm{a}}_{\theta_k}^\lambda) \right\|_2^2 \Big|\mathcal{F}_k \right]$$

$$= \tilde{J}_\lambda(\theta_k) + \left( 2\eta_{\mathsf{a}} - \frac{\eta_{\mathsf{a}}^2 L}{2} \right) \left\| \nabla \tilde{J}_\lambda(\theta_k) \right\|_2^2 - \frac{\eta_{\mathsf{a}}^2 L}{2} \mathbb{E}\left[ \left\| \mathrm{g}_{\mathsf{a}}^{Y_{k+1}}(\tilde{\mathrm{a}}_{\theta_k}^\lambda) - \nabla \tilde{J}_\lambda(\theta_k) \right\|_2^2 \Big|\mathcal{F}_k \right] \;,$$

where the last identity is obtained via the bias–variance decomposition of the estimator $\mathrm{g}_{\mathsf{a}}^{Y_{k+1}}(\tilde{\mathrm{a}}_{\theta_k}^\lambda)$. We then apply Lemma 11 to obtain

$$\mathbb{E}\left[ \tilde{J}_\lambda(\theta_{k+1})|\mathcal{F}_k \right] \geq \tilde{J}_\lambda(\theta_k) + \left( 2\eta_{\mathsf{a}} - \frac{\eta_{\mathsf{a}}^2 L}{2} - \frac{\eta_{\mathsf{a}}^2 L}{2(1-\gamma)\pi_{\min}\rho_{\min}} \right) \left\| \nabla \tilde{J}_\lambda(\theta_k) \right\|_2^2 \;.$$

After subtracting $\tilde{J}_\lambda^\star$ from both sides, we apply the non-uniform PL inequality for $\tilde{J}_\lambda$ recalled in Lemma 3, along with the uniform lower bound on the PL coefficient in (16), to obtain

$$\tilde{J}_\lambda^\star - \mathbb{E}\left[\tilde{J}_\lambda(\theta_{k+1})|\mathcal{F}_k\right] \leq \left[1 - \underline{\tilde{\mu}}_\lambda\left(2\eta_{\mathsf{a}} - \frac{\eta_{\mathsf{a}}^2 L}{2} - \frac{\eta_{\mathsf{a}}^2 L}{2(1-\gamma)\pi_{\min}\rho_{\min}}\right)\right](\tilde{J}_\lambda^\star - \tilde{J}_\lambda(\theta_k))$$

$$\leq \left[1 - \eta_{\mathsf{a}}\underline{\tilde{\mu}}_\lambda\right](\tilde{J}_\lambda^\star - \tilde{J}_\lambda(\theta_k)) \; ,$$

where the last inequality follows from the step-size condition $\eta_{\mathsf{a}} \leq \frac{(1-\gamma)\rho_{\min}\pi_{\min}}{L}$. Finally, taking expectation over all the randomness and unrolling the recursion gives the desired result. $\qquad\square$

Next, we derive the sample complexity of `Ent-AC`, in case of this exact critic.

**Corollary 3** (Sample complexity). *Under the same assumptions as Theorem 3, for any $\epsilon > 0$, setting $\eta_{\mathsf{a}} = (1 - \gamma)\rho_{\min}\pi_{\min}/L$, and taking*

$$k \geq \frac{L}{\underline{\tilde{\mu}}_\lambda}\frac{1}{(1-\gamma)\rho_{\min}\tau_\lambda}\log\left(\frac{\tilde{J}_\lambda^\star - \tilde{J}_\lambda(\theta_0)}{\epsilon}\right) \; ,$$

*we get $\tilde{J}_\lambda^\star - \mathbb{E}\left[\tilde{J}_\lambda(\theta_k)\right] \leq \epsilon$.*

## C. General Analysis of `Ent-AC`: Actor Recursion

We define

$$\bar{g}_{\mathsf{a}}(\theta, \mathsf{a}) \stackrel{\Delta}{=} \mathbb{E}_{Y\sim\nu^{\mathsf{a}}(\theta)}\left[g_{\mathsf{a}}^Y(\mathsf{a})\right] \; .$$

We start with the following lemma that bounds the variance and the bias of the estimator in the presence of an inexact advantage used in the computation of the critic.

**Lemma 12.** *Fix $\theta \in \mathbb{R}^{|\mathcal{S}||\mathcal{A}|}$ and $\mathsf{a} \in \mathbb{R}^{|\mathcal{S}||\mathcal{A}|}$. It holds that*

$$\left\|\bar{g}_{\mathsf{a}}(\theta, \mathsf{a}) - \frac{\partial\tilde{v}_\theta^\lambda}{\partial\theta}\right\|_2^2 = \frac{1}{(1-\gamma)^2}\sum_{(s,a)\in\mathcal{S}\times\mathcal{A}} d_\rho^\theta(s)^2\pi_\theta(a|s)^2\left(\mathsf{a}(s,a) - \tilde{\mathsf{a}}_\theta^\lambda(s,a)\right)^2 \; ,$$

$$\mathbb{E}_{Y\sim\nu^{\mathsf{a}}(\theta)}\left[\left\|g_{\mathsf{a}}^Y(\mathsf{a}) - \bar{g}_{\mathsf{a}}(\theta, \mathsf{a})\right\|_2^2\right] \leq \frac{1}{(1-\gamma)^2}\sum_{(s,a)\in\mathcal{S}\times\mathcal{A}} d_\rho^\theta(s)\pi_\theta(a|s)\mathsf{a}(s,a)^2 \; .$$

*Proof.* Firstly, note that for any $(s,a)\in\mathcal{S}\times\mathcal{A}$, we have

$$[\bar{g}_{\mathsf{a}}(\theta, \mathsf{a})]_{s,a} = \left[\mathbb{E}_{Y\sim\nu^{\mathsf{a}}(\theta)}\left[g_{\mathsf{a}}^Y(\mathsf{a})\right]\right]_{s,a} = \sum_{s',a'} d_\rho^\theta(s')\pi_\theta(a'|s')1_{(s,a)}(s',a')(1-\gamma)^{-1}\mathsf{a}(s',a')$$

$$= (1-\gamma)^{-1}d_\rho^\theta(s)\pi_\theta(a|s)\mathsf{a}(s,a) \; .$$

Combining the previous identity with Lemma 1, yields the result. Next, we have

$$\mathbb{E}_{Y\sim\nu^{\mathsf{a}}(\theta)}\left[\left\|g_{\mathsf{a}}^Y(\mathsf{a}) - \bar{g}_{\mathsf{a}}(\theta, \mathsf{a})\right\|_2^2\right] = \mathbb{E}_{Y\sim\nu^{\mathsf{a}}(\theta)}\left[\left\|g_{\mathsf{a}}^Y(\mathsf{a})\right\|_2^2\right] - \left\|\bar{g}_{\mathsf{a}}(\theta, \mathsf{a})\right\|_2^2$$

$$= \frac{1}{(1-\gamma)^2}\sum_{(s,a)\in\mathcal{S}\times\mathcal{A}} d_\rho^\theta(s')\pi_\theta(a'|s')\left[\mathsf{a}(s,a)^2 - d_\rho^\theta(s)\pi_\theta(a|s)\mathsf{a}(s,a)^2\right] \; ,$$

which concludes the proof. $\qquad\square$

Next, we define the two following filtrations

$$\mathcal{F}_k \stackrel{\Delta}{=} \sigma\left(X_\ell : \ell \in \{0,\ldots,k-1\}, Y_\ell : \ell \in \{1,\ldots,k\}\right) \; , \quad \mathcal{G}_k \stackrel{\Delta}{=} \sigma\left(X_\ell : \ell \in \{1,\ldots,k\}, Y_\ell : \ell \in \{1,\ldots,k\}\right) \; .$$

In the latter, we derive a recursion on the error of the actor.

**Lemma 13** (Actor recursion). *Assume $A_\rho$. For any $k \geq 0$ and $(s, a) \in \mathcal{S} \times \mathcal{A}$, it holds that*

$$\pi_{\theta_k}(a|s) \geq \tau_\lambda .$$

*Additionally, it holds that*

$$\mathbb{E}\left[\tilde{J}_\lambda^\star - \tilde{J}_\lambda(\theta_{k+1})\big|\mathcal{F}_k\right] \leq \tilde{J}_\lambda^\star - \tilde{J}_\lambda(\theta_k) - \left(\frac{\eta_\mathsf{a}}{2} - \frac{2L\eta_\mathsf{a}^2}{(1-\gamma)\rho_{\min}\tau_\lambda}\right)\left\|\nabla\tilde{J}_\lambda(\theta_k)\right\|_2^2$$

$$+ \frac{2\eta_\mathsf{a}}{(1-\gamma)^2}\sum_{(s,a)\in\mathcal{S}\times\mathcal{A}} d_\rho^{\theta_k}(s)^2\pi_{\theta_k}(a|s)^2\left(\mathbb{E}[\hat{\mathsf{a}}_k(s,a)|\mathcal{F}_k] - \tilde{\mathsf{a}}_{\theta_k}^\lambda(s,a)\right)^2$$

$$+ \frac{2L\eta_\mathsf{a}^2}{(1-\gamma)^2}\sum_{(s,a)\in\mathcal{S}\times\mathcal{A}} d_\rho^{\theta_k}(s)\pi_{\theta_k}(a|s)\mathbb{E}\left[(\hat{\mathsf{a}}_k(s,a) - \tilde{\mathsf{a}}_{\theta_k}^\lambda(s,a))^2|\mathcal{F}_k\right] .$$

*Proof.* First, the claimed lower bound on

$$\min_{k\geq 0}\min_{(s,a)}\pi_{\theta_k}(a \mid s)$$

is an immediate consequence of the update rule and Lemma 21. In addition, Lemma 4 gives

$$\tilde{J}_\lambda(\theta_{k+1}) \geq \tilde{J}_\lambda\left(\theta_k + \eta_\mathsf{a}\mathsf{g}_\mathsf{a}^{Y_{k+1}}(\hat{\mathsf{a}}_k)\right).$$

Since $\tilde{J}_\lambda$ is $L$-smooth by Lemma 2, we may then apply Lemma 23 to obtain

$$\tilde{J}_\lambda(\theta_{k+1}) \geq \tilde{J}_\lambda(\theta_k) + \eta_\mathsf{a}\langle\nabla\tilde{J}_\lambda(\theta_k), \mathsf{g}_\mathsf{a}^{Y_{k+1}}(\hat{\mathsf{a}}_k)\rangle - \frac{L\eta_\mathsf{a}^2}{2}\left\|\mathsf{g}_\mathsf{a}^{Y_{k+1}}(\hat{\mathsf{a}}_k)\right\|_2^2 .$$

Subtracting $\tilde{J}_\lambda^\star$ and multiplying both sides by $-1$, yields

$$\tilde{J}_\lambda^\star - \tilde{J}_\lambda(\theta_{k+1}) \leq \tilde{J}_\lambda^\star - \tilde{J}_\lambda(\theta_k) - \eta_\mathsf{a}\langle\nabla\tilde{J}_\lambda(\theta_k), \mathsf{g}_\mathsf{a}^{Y_{k+1}}(\hat{\mathsf{a}}_k)\rangle + \frac{L\eta_\mathsf{a}^2}{2}\left\|\mathsf{g}_\mathsf{a}^{Y_{k+1}}(\hat{\mathsf{a}}_k)\right\|_2^2 .$$

Taking the conditionnal expectation with respect to $\mathcal{G}_k$, yields

$$\mathbb{E}\left[\tilde{J}_\lambda^\star - \tilde{J}_\lambda(\theta_{k+1})\big|\mathcal{G}_k\right] \leq \tilde{J}_\lambda^\star - \tilde{J}_\lambda(\theta_k) - \eta_\mathsf{a}\langle\nabla\tilde{J}_\lambda(\theta_k), \bar{\mathsf{g}}_\mathsf{a}(\theta_k, \hat{\mathsf{a}}_k)\rangle + \frac{L\eta_\mathsf{a}^2}{2}\mathbb{E}\left[\left\|\mathsf{g}_\mathsf{a}^{Y_{k+1}}(\hat{\mathsf{a}}_k)\right\|_2^2\Big|\mathcal{G}_k\right]$$

$$= \mathbb{E}\left[\tilde{J}_\lambda^\star - \tilde{J}_\lambda(\theta_k)\big|\mathcal{G}_k\right] - \eta_\mathsf{a}\langle\nabla\tilde{J}_\lambda(\theta_k), \bar{\mathsf{g}}_\mathsf{a}(\theta_k, \hat{\mathsf{a}}_k)\rangle$$

$$+ \frac{L\eta_\mathsf{a}^2}{2}\mathbb{E}\left[\left\|\mathsf{g}_\mathsf{a}^{Y_{k+1}}(\hat{\mathsf{a}}_k) - \bar{\mathsf{g}}_\mathsf{a}(\theta_k, \hat{\mathsf{a}}_k)\right\|_2^2\Big|\mathcal{G}_k\right] + \frac{L\eta_\mathsf{a}^2}{2}\left\|\bar{\mathsf{g}}_\mathsf{a}(\theta_k, \hat{\mathsf{a}}_k)\right\|_2^2 . \tag{21}$$

Applying Young's inequality gives

$$\frac{L\eta_\mathsf{a}^2}{2}\left\|\bar{\mathsf{g}}_\mathsf{a}(\theta_k, \hat{\mathsf{a}}_k)\right\|_2^2 \leq L\eta_\mathsf{a}^2\left\|\bar{\mathsf{g}}_\mathsf{a}(\theta_k, \hat{\mathsf{a}}_k) - \nabla\tilde{J}_\lambda(\theta_k)\right\|_2^2 + L\eta_\mathsf{a}^2\left\|\nabla\tilde{J}_\lambda(\theta_k)\right\|_2^2 .$$

Combining this with (21) and the variance formula of Lemma 12 yields

$$\mathbb{E}\left[\tilde{J}_\lambda^\star - \tilde{J}_\lambda(\theta_{k+1})\big|\mathcal{G}_k\right] \leq \tilde{J}_\lambda^\star - \tilde{J}_\lambda(\theta_k) - \eta_\mathsf{a}\langle\nabla\tilde{J}_\lambda(\theta_k), \bar{\mathsf{g}}_\mathsf{a}(\theta_k, \hat{\mathsf{a}}_k)\rangle$$

$$+ \frac{L\eta_\mathsf{a}^2}{2(1-\gamma)^2}\sum_{(s,a)\in\mathcal{S}\times\mathcal{A}} d_\rho^{\theta_k}(s)\pi_{\theta_k}(a|s)\hat{\mathsf{a}}_k(s,a)^2$$

$$+ L\eta_\mathsf{a}^2\left\|\bar{\mathsf{g}}_\mathsf{a}(\theta_k, \hat{\mathsf{a}}_k) - \nabla\tilde{J}_\lambda(\theta_k)\right\|_2^2 + L\eta_\mathsf{a}^2\left\|\nabla\tilde{J}_\lambda(\theta_k)\right\|_2^2 .$$

Adding and subtracting $\nabla\tilde{J}_\lambda(\theta_k)$ in the scalar product term, combined with Lemma 12 applied to the fourth term, gives

$$\mathbb{E}\left[\tilde{J}_\lambda^\star - \tilde{J}_\lambda(\theta_{k+1})\big|\mathcal{G}_k\right] \leq \tilde{J}_\lambda^\star - \tilde{J}_\lambda(\theta_k) - \eta_\mathsf{a}\left\|\nabla\tilde{J}_\lambda(\theta_k)\right\|_2^2 - \eta_\mathsf{a}\langle\nabla\tilde{J}_\lambda(\theta_k), \bar{\mathsf{g}}_\mathsf{a}(\theta_k, \hat{\mathsf{a}}_k) - \nabla\tilde{J}_\lambda(\theta_k)\rangle$$

$$+ \frac{L\eta_{\mathsf{a}}^2}{2(1-\gamma)^2} \sum_{(s,a)\in\mathcal{S}\times\mathcal{A}} d_\rho^{\theta_k}(s)\pi_{\theta_k}(a|s)\hat{\mathsf{a}}_k(s,a)^2$$

$$+ \frac{L\eta_{\mathsf{a}}^2}{(1-\gamma)^2} \sum_{(s,a)\in\mathcal{S}\times\mathcal{A}} d_\rho^{\theta_k}(s)^2\pi_{\theta_k}(a|s)^2 \left(\hat{\mathsf{a}}_k(s,a) - \tilde{\mathsf{a}}_{\theta_k}^\lambda(s,a)\right)^2 + L\eta_{\mathsf{a}}^2 \left\|\nabla\tilde{J}_\lambda(\theta_k)\right\|_2^2 .$$

Next, taking the conditional expectation, with respect to $\mathcal{F}_k$, yields

$$\mathbb{E}\left[\tilde{J}_\lambda^\star - \tilde{J}_\lambda(\theta_{k+1})\big|\mathcal{F}_k\right] \leq \tilde{J}_\lambda^\star - \tilde{J}_\lambda(\theta_k) - (\eta_{\mathsf{a}} - L\eta_{\mathsf{a}}^2)\left\|\nabla\tilde{J}_\lambda(\theta_k)\right\|_2^2 - \eta_{\mathsf{a}}\underbrace{\langle\nabla\tilde{J}_\lambda(\theta_k), \mathbb{E}\left[\bar{\mathsf{g}}_{\mathsf{a}}(\theta_k,\hat{\mathsf{a}}_k)|\mathcal{F}_k\right] - \nabla\tilde{J}_\lambda(\theta_k)\rangle}_{(\mathbf{P})}$$

$$+ \frac{L\eta_{\mathsf{a}}^2}{2(1-\gamma)^2} \sum_{(s,a)\in\mathcal{S}\times\mathcal{A}} d_\rho^{\theta_k}(s)\pi_{\theta_k}(a|s)\mathbb{E}[\hat{\mathsf{a}}_k(s,a)^2|\mathcal{F}_k]$$

$$+ \frac{L\eta_{\mathsf{a}}^2}{(1-\gamma)^2} \sum_{(s,a)\in\mathcal{S}\times\mathcal{A}} d_\rho^{\theta_k}(s)^2\pi_{\theta_k}(a|s)^2\mathbb{E}[\left(\hat{\mathsf{a}}_k(s,a) - \tilde{\mathsf{a}}_{\theta_k}^\lambda(s,a)\right)^2|\mathcal{F}_k] .$$

To bound $(\mathbf{P})$, we apply Cauchy-Schwarz inequality, followed by Young's inequality, which gives

$$|\mathbf{P}| \leq \frac{\eta_{\mathsf{a}}}{2}\left\|\nabla\tilde{J}_\lambda(\theta_k)\right\|_2^2 + 2\eta_{\mathsf{a}}\left\|\mathbb{E}[\bar{\mathsf{g}}_{\mathsf{a}}(\theta_k,\hat{\mathsf{a}}_k)|\mathcal{F}_k] - \nabla\tilde{J}_\lambda(\theta_k)\right\|_2^2 .$$

Plugging in the previous bound on $(\mathbf{P})$ in the preceding inequality gives

$$\mathbb{E}\left[\tilde{J}_\lambda^\star - \tilde{J}_\lambda(\theta_{k+1})\big|\mathcal{F}_k\right] \leq \tilde{J}_\lambda^\star - \tilde{J}_\lambda(\theta_k) - \left(\frac{\eta_{\mathsf{a}}}{2} - L\eta_{\mathsf{a}}^2\right)\left\|\nabla\tilde{J}_\lambda(\theta_k)\right\|_2^2 + 2\eta_{\mathsf{a}}\left\|\mathbb{E}[\bar{\mathsf{g}}_{\mathsf{a}}(\theta_k,\hat{\mathsf{a}}_k)|\mathcal{F}_k] - \nabla\tilde{J}_\lambda(\theta_k)\right\|_2^2$$

$$+ \frac{L\eta_{\mathsf{a}}^2}{2(1-\gamma)^2} \sum_{(s,a)\in\mathcal{S}\times\mathcal{A}} d_\rho^{\theta_k}(s)\pi_{\theta_k}(a|s)\mathbb{E}\left[\hat{\mathsf{a}}_k(s,a)^2|\mathcal{F}_k\right]$$

$$+ \frac{L\eta_{\mathsf{a}}^2}{(1-\gamma)^2} \sum_{(s,a)\in\mathcal{S}\times\mathcal{A}} d_\rho^{\theta_k}(s)^2\pi_{\theta_k}(a|s)^2\mathbb{E}\left[\left(\hat{\mathsf{a}}_k(s,a) - \tilde{\mathsf{a}}_{\theta_k}^\lambda(s,a)\right)^2|\mathcal{F}_k\right] . \tag{22}$$

Observe that applying Young's inequality, we have that

$$\sum_{(s,a)\in\mathcal{S}\times\mathcal{A}} d_\rho^{\theta_k}(s)\pi_{\theta_k}(a|s)\mathbb{E}\left[\hat{\mathsf{a}}_k(s,a)^2|\mathcal{F}_k\right] \leq 2\sum_{(s,a)\in\mathcal{S}\times\mathcal{A}} d_\rho^{\theta_k}(s)\pi_{\theta_k}(a|s)\mathbb{E}\left[(\hat{\mathsf{a}}_k(s,a) - \tilde{\mathsf{a}}_{\theta_k}^\lambda(s,a))^2|\mathcal{F}_k\right]$$

$$+ 2\sum_{(s,a)\in\mathcal{S}\times\mathcal{A}} d_\rho^{\theta_k}(s)\pi_{\theta_k}(a|s)\mathbb{E}\left[\tilde{\mathsf{a}}_{\theta_k}^\lambda(s,a)^2|\mathcal{F}_k\right] .$$

Plugging in the previous inequality in (22) yields

$$\mathbb{E}\left[\tilde{J}_\lambda^\star - \tilde{J}_\lambda(\theta_{k+1})\big|\mathcal{F}_k\right] \leq \tilde{J}_\lambda^\star - \tilde{J}_\lambda(\theta_k) - \left(\frac{\eta_{\mathsf{a}}}{2} - L\eta_{\mathsf{a}}^2\right)\left\|\nabla\tilde{J}_\lambda(\theta_k)\right\|_2^2 + 2\eta_{\mathsf{a}}\left\|\mathbb{E}[\bar{\mathsf{g}}_{\mathsf{a}}(\theta_k,\hat{\mathsf{a}}_k)|\mathcal{F}_k] - \nabla\tilde{J}_\lambda(\theta_k)\right\|_2^2$$

$$+ \underbrace{\frac{L\eta_{\mathsf{a}}^2}{(1-\gamma)^2} \sum_{(s,a)\in\mathcal{S}\times\mathcal{A}} d_\rho^{\theta_k}(s)\pi_{\theta_k}(a|s)\tilde{\mathsf{a}}_{\theta_k}^\lambda(s,a)^2}_{(\mathbf{A})}$$

$$+ \underbrace{\frac{L\eta_{\mathsf{a}}^2}{(1-\gamma)^2} \sum_{(s,a)\in\mathcal{S}\times\mathcal{A}} d_\rho^{\theta_k}(s)\pi_{\theta_k}(a|s)\mathbb{E}\left[\left(\hat{\mathsf{a}}_k(s,a) - \tilde{\mathsf{a}}_{\theta_k}^\lambda(s,a)\right)^2|\mathcal{F}_k\right]}_{(\mathbf{B})}$$

$$+ \underbrace{\frac{L\eta_{\mathsf{a}}^2}{(1-\gamma)^2} \sum_{(s,a)\in\mathcal{S}\times\mathcal{A}} d_\rho^{\theta_k}(s)^2\pi_{\theta_k}(a|s)^2\mathbb{E}\left[\left(\hat{\mathsf{a}}_k(s,a) - \tilde{\mathsf{a}}_{\theta_k}^\lambda(s,a)\right)^2|\mathcal{F}_k\right]}_{(\mathbf{C})} .$$

**Bounding (A).** Using $\mathbf{A}_\rho$ combined with the fact that for any $(s,a) \in \mathcal{S} \times \mathcal{A}$, we have that $\pi_{\theta_k}(a|s) \geq \tau_\lambda$, we have that

$$(\mathbf{A}) \leq \frac{1}{(1-\gamma)\rho_{\min}\tau_\lambda} \frac{L\eta_{\mathsf{a}}^2}{(1-\gamma)^2} \sum_{(s,a)\in\mathcal{S}\times\mathcal{A}} d_\rho^{\theta_k}(s)^2 \pi_{\theta_k}(a|s)^2 \tilde{\mathsf{a}}_{\theta_k}^\lambda(s,a)^2 \leq \frac{L\eta_{\mathsf{a}}^2}{(1-\gamma)\rho_{\min}\tau_\lambda} \left\| \nabla \tilde{J}_\lambda(\theta_k) \right\|_2^2 .$$

**Bounding (B).** Using that $d_\rho^{\theta_k}(s)\pi_{\theta_k}(a\mid s) \leq 1$, we have $(\mathbf{B}) \leq (\mathbf{C})$.

Combining the previous bounds, we get

$$\mathbb{E}\left[\tilde{J}_\lambda^\star - \tilde{J}_\lambda(\theta_{k+1})\big|\mathcal{F}_k\right] \leq \tilde{J}_\lambda^\star - \tilde{J}_\lambda(\theta_k) - \left(\frac{\eta_{\mathsf{a}}}{2} - L\eta_{\mathsf{a}}^2\right) \left\| \nabla \tilde{J}_\lambda(\theta_k) \right\|_2^2 + 2\eta_{\mathsf{a}} \left\| \mathbb{E}[\bar{\mathsf{g}}_{\mathsf{a}}(\theta_k,\hat{\mathsf{a}}_k)|\mathcal{F}_k] - \nabla \tilde{J}_\lambda(\theta_k) \right\|_2^2$$
$$+ \frac{L\eta_{\mathsf{a}}^2}{(1-\gamma)\rho_{\min}\tau_\lambda} \left\| \nabla \tilde{J}_\lambda(\theta_k) \right\|_2^2$$
$$+ \frac{2L\eta_{\mathsf{a}}^2}{(1-\gamma)^2} \sum_{(s,a)\in\mathcal{S}\times\mathcal{A}} d_\rho^{\theta_k}(s)\pi_{\theta_k}(a|s)\mathbb{E}\left[\left(\hat{\mathsf{a}}_k(s,a) - \tilde{\mathsf{a}}_{\theta_k}^\lambda(s,a)\right)^2\big|\mathcal{F}_k\right] .$$

Next, using that

$$L\eta_{\mathsf{a}}^2 \leq \frac{L\eta_{\mathsf{a}}^2}{(1-\gamma)\rho_{\min}\tau_\lambda} ,$$

we get

$$\mathbb{E}\left[\tilde{J}_\lambda^\star - \tilde{J}_\lambda(\theta_{k+1})\big|\mathcal{F}_k\right] \leq \tilde{J}_\lambda^\star - \tilde{J}_\lambda(\theta_k) - \left(\frac{\eta_{\mathsf{a}}}{2} - \frac{2L\eta_{\mathsf{a}}^2}{(1-\gamma)\rho_{\min}\tau_\lambda}\right) \left\| \nabla \tilde{J}_\lambda(\theta_k) \right\|_2^2 + 2\eta_{\mathsf{a}} \left\| \mathbb{E}[\bar{\mathsf{g}}_{\mathsf{a}}(\theta_k,\hat{\mathsf{a}}_k)|\mathcal{F}_k] - \nabla \tilde{J}_\lambda(\theta_k) \right\|_2^2$$
$$+ \frac{2L\eta_{\mathsf{a}}^2}{(1-\gamma)^2} \sum_{(s,a)\in\mathcal{S}\times\mathcal{A}} d_\rho^{\theta_k}(s)\pi_{\theta_k}(a|s)\mathbb{E}\left[(\hat{\mathsf{a}}_k(s,a) - \tilde{\mathsf{a}}_{\theta_k}^\lambda(s,a))^2|\mathcal{F}_k\right] .$$

Finally, using that $\mathbb{E}[\bar{\mathsf{g}}_{\mathsf{a}}(\theta_k,\hat{\mathsf{a}}_k)|\mathcal{F}_k] = \bar{\mathsf{g}}_{\mathsf{a}}(\theta_k, \mathbb{E}[\hat{\mathsf{a}}_k|\mathcal{F}_k])$, combined with Lemma 17 concludes the proof. $\qquad \square$

Importantly, in the preceding lemma, we observe that if the critic is perfectly learned, we recover the linear convergence regime for constant step-size observed in Theorem 3. The next lemma bounds the distance between two consecutive policies computed by `Ent-AC`.

**Lemma 14.** *Assume $\mathbf{A}_\rho$. It holds that*

$$\left\| \tilde{\mathsf{q}}_{\theta_{k+1}}^\lambda - \tilde{\mathsf{q}}_{\theta_k}^\lambda \right\|_\infty \leq C_\lambda \eta_{\mathsf{a}} |\hat{\mathsf{a}}_k(S_{k+1}, A_{k+1})| , \text{ where } C_\lambda = \frac{2\gamma}{1-\gamma} \left(\frac{1 + \lambda\log(|\mathcal{A}|)}{1-\gamma} + \lambda\log(1/\tau_\lambda) + \frac{\lambda}{2\tau_\lambda}\right) .$$

*Proof.* Denote by $\tilde{\theta}_{k+1} = \theta_k + \eta_{\mathsf{a}} \mathsf{g}_{\mathsf{a}}^{Y_{k+1}}(\hat{\mathsf{a}}_k)$. By Equation (5). For any $(s,a) \in \mathcal{S} \times \mathcal{A}$, we have

$$\left| \tilde{\mathsf{q}}_{\theta_{k+1}}^\lambda(s,a) - \tilde{\mathsf{q}}_{\theta_k}^\lambda(s,a) \right| \leq \left| \tilde{\mathsf{q}}_{\theta_{k+1}}^\lambda(s,a) - \tilde{\mathsf{q}}_{\tilde{\theta}_{k+1}}^\lambda(s,a) \right| + \left| \tilde{\mathsf{q}}_{\tilde{\theta}_{k+1}}^\lambda(s,a) - \tilde{\mathsf{q}}_{\theta_k}^\lambda(s,a) \right|$$
$$= \gamma \underbrace{\left| \sum_{s'\in\mathcal{S}} \mathsf{P}(s'|s,a) \left[\tilde{\mathsf{v}}_{\theta_{k+1}}^\lambda(s') - \tilde{\mathsf{v}}_{\tilde{\theta}_{k+1}}^\lambda(s')\right] \right|}_{(\mathbf{A_1})} + \gamma \underbrace{\left| \sum_{s'\in\mathcal{S}} \mathsf{P}(s'|s,a) \left[\tilde{\mathsf{v}}_{\tilde{\theta}_{k+1}}^\lambda(s') - \tilde{\mathsf{v}}_{\theta_k}^\lambda(s')\right] \right|}_{(\mathbf{A_2})} ,$$

where in the last equality, we used (5). Next, we bound each of these terms similarly.

**Bounding $(\mathbf{A_1})$.** Applying the soft performance-difference lemma (Lemma 26), we obtain

$$(\mathbf{A_1}) \leq \left| \sum_{s'\in\mathcal{S}} \mathsf{P}(s'|s,a) \left[\frac{\gamma}{1-\gamma} \sum_{s''\in\mathcal{S}} d_{s'}^{\tilde{\theta}_{k+1}}(s'') \left[\sum_{a'\in\mathcal{A}} \left(\pi_{\theta_{k+1}}(a'|s'') - \pi_{\tilde{\theta}_{k+1}}(a'|s'')\right)\right.\right.\right.$$

$$\times \left[\tilde{q}^{\lambda}_{\theta_{k+1}}(s'', a') - \lambda \log\left(\pi_{\theta_{k+1}}(a'|s'')\right)\right] + \lambda \operatorname{KL}\left(\pi_{\tilde{\theta}_{k+1}}(\cdot|s'') \,\middle\|\, \pi_{\theta_{k+1}}(\cdot|s'')\right)\Big]\Big]\Big| \,.$$

Next, since for all $(s, a) \in \mathcal{S} \times \mathcal{A}$ we have

$$\pi_{\theta_k}(a|s) \geq \tau_\lambda \qquad \text{and} \qquad \pi_{\theta_{k+1}}(a|s) \geq \tau_\lambda,$$

Lemma 29 implies that, for every $s'' \in \mathcal{S}$,

$$\operatorname{KL}\left(\pi_{\tilde{\theta}_{k+1}}(\cdot|s'') \,\middle\|\, \pi_{\theta_{k+1}}(\cdot|s'')\right) \leq \frac{1}{2\tau_\lambda} \left\|\pi_{\tilde{\theta}_{k+1}}(\cdot|s'') - \pi_{\theta_{k+1}}(\cdot|s'')\right\|_1 \,.$$

Therefore, using the triangle inequality together with

$$\sum_{(s', s'') \in \mathcal{S} \times \mathcal{S}} \mathsf{P}(s'|s, a)\, d^{\tilde{\theta}_{k+1}}_{s'}(s'') = 1,$$

we arrive at

$$(\mathbf{A_1}) \leq \frac{\gamma}{1-\gamma} \max_{s'' \in \mathcal{S}} \left| \sum_{a' \in \mathcal{A}} \left(\pi_{\theta_{k+1}}(a'|s'') - \pi_{\tilde{\theta}_{k+1}}(a'|s'')\right)\left[\tilde{q}^{\lambda}_{\theta_{k+1}}(s'', a') - \lambda \log\left(\pi_{\theta_{k+1}}(a'|s'')\right)\right]\right|$$
$$+ \frac{\lambda}{2\tau_\lambda}\left\|\pi_{\tilde{\theta}_{k+1}}(\cdot|s'') - \pi_{\theta_{k+1}}(\cdot|s'')\right\|_1 \,.$$

Now, using Lemma 27, we deduce that

$$(\mathbf{A_1}) \leq \frac{\gamma}{1-\gamma} \max_{s' \in \mathcal{S}} \left\{ \left\|\pi_{\tilde{\theta}_{k+1}}(\cdot|s') - \pi_{\theta_{k+1}}(\cdot|s')\right\|_1 \left(\frac{1 + \lambda \log(|\mathcal{A}|)}{1-\gamma} - \lambda \log(\tau_\lambda) + \frac{\lambda}{\tau_\lambda}\right)\right\} \,.$$

Finally, using Lemma 22, the fact that $\pi_{\theta_{k+1}} = \mathcal{U}_{\tau_\lambda}(\pi_{\tilde{\theta}_{k+1}})$, and $\pi_{\theta_k} \in \Pi_{\tau_\lambda}$, ensures that

$$\left\|\pi_{\tilde{\theta}_{k+1}}(\cdot|s') - \pi_{\theta_{k+1}}(\cdot|s')\right\|_1 \leq \left\|\pi_{\tilde{\theta}_{k+1}}(\cdot|s') - \pi_{\theta_k}(\cdot|s')\right\|_1.$$

Substituting this estimate into the previous display gives

$$(\mathbf{A_1}) \leq \frac{\gamma}{1-\gamma} \max_{s' \in \mathcal{S}} \left\{ \left\|\pi_{\tilde{\theta}_{k+1}}(\cdot|s') - \pi_{\theta_k}(\cdot|s')\right\|_1 \left(\frac{1 + \lambda \log(|\mathcal{A}|)}{1-\gamma} - \lambda \log(\tau_\lambda) + \frac{\lambda}{2\tau_\lambda}\right)\right\} \,.$$

**Bounding $(\mathbf{A_2})$.** Applying again the soft performance-difference lemma (Lemma 26), we get

$$(\mathbf{A_2}) \leq \left| \sum_{s' \in \mathcal{S}} \mathsf{P}(s'|s, a)\left[\frac{\gamma}{1-\gamma} \sum_{s'' \in \mathcal{S}} d^{\tilde{\theta}_{k+1}}_{s'}(s'')\left[\sum_{a' \in \mathcal{A}} \left(\pi_{\theta_k}(a'|s'') - \pi_{\tilde{\theta}_{k+1}}(a'|s'')\right)\right.\right.\right.$$
$$\times \left[\tilde{q}^{\lambda}_{\theta_k}(s'', a') - \lambda \log\left(\pi_{\theta_k}(a'|s'')\right)\right] + \lambda \operatorname{KL}\left(\pi_{\tilde{\theta}_{k+1}}(\cdot|s'') \,\middle\|\, \pi_{\theta_k}(\cdot|s'')\right)\Big]\Big]\Big| \,.$$

Using again the lower bound on the policy probabilities together with Lemma 29, we obtain

$$\operatorname{KL}\left(\pi_{\tilde{\theta}_{k+1}}(\cdot|s'') \,\middle\|\, \pi_{\theta_k}(\cdot|s'')\right) \leq \frac{1}{2\tau_\lambda} \left\|\pi_{\tilde{\theta}_{k+1}}(\cdot|s'') - \pi_{\theta_k}(\cdot|s'')\right\|_1 \,.$$

Proceeding exactly as in the bound on $(\mathbf{A_1})$, and using both

$$\sum_{(s', s'') \in \mathcal{S} \times \mathcal{S}} \mathsf{P}(s'|s, a)\, d^{\tilde{\theta}_{k+1}}_{s'}(s'') = 1$$

and the bound on the regularized $Q$-function, we obtain

$$(\mathbf{A_2}) \leq \frac{\gamma}{1-\gamma} \max_{s' \in \mathcal{S}} \left\{ \left\| \pi_{\tilde{\theta}_{k+1}}(\cdot|s') - \pi_{\theta_k}(\cdot|s') \right\|_1 \left( \frac{1 + \lambda \log(|\mathcal{A}|)}{1-\gamma} - \lambda \log(\tau_\lambda) + \frac{\lambda}{2\tau_\lambda} \right) \right\} .$$

Combining the bounds on $(\mathbf{A_1})$ and $(\mathbf{A_2})$ yields

$$\left| \tilde{q}^\lambda_{\theta_{k+1}}(s, a) - \tilde{q}^\lambda_{\theta_k}(s, a) \right| \leq \frac{2\gamma}{1-\gamma} \max_{s' \in \mathcal{S}} \left\{ \left\| \pi_{\tilde{\theta}_{k+1}}(\cdot|s') - \pi_{\theta_k}(\cdot|s') \right\|_1 \right. \tag{23}$$

$$\left. \times \left( \frac{1 + \lambda \log(|\mathcal{A}|)}{1-\gamma} - \lambda \log(\tau_\lambda) + \frac{\lambda}{2\tau_\lambda} \right) \right\} . \tag{24}$$

It remains to bound the distance $\left\| \pi_{\tilde{\theta}_{k+1}}(\cdot|s') - \pi_{\theta_k}(\cdot|s') \right\|_1$. To this end, we combine Pinsker's inequality (Lemma 28) with the KL-logit inequality (Lemma 30) to obtain

$$\left\| \pi_{\tilde{\theta}_{k+1}}(\cdot|s') - \pi_{\theta_k}(\cdot|s') \right\|_1 \leq \sqrt{2} \sqrt{\mathrm{KL}\left( \pi_{\tilde{\theta}_{k+1}}(\cdot|s') \, \middle\| \, \pi_{\theta_k}(\cdot|s') \right)}$$

$$\leq \left\| \tilde{\theta}_{k+1}(s', \cdot) - \theta_k(s', \cdot) \right\|_\infty$$

$$\leq \eta_{\mathsf{a}} \left| \hat{\mathsf{a}}_k(S_{k+1}, A_{k+1}) \right| .$$

Plugging this estimate into Equation (23) concludes the proof. $\qquad \square$

**Corollary 4.** *Assume $\mathbf{A}_\rho$. It holds that*

$$\left\| \tilde{q}^\lambda_{\theta_{k+1}} - \tilde{q}^\lambda_{\theta_k} \right\|_2 \leq \tilde{C}_\lambda \eta_{\mathsf{a}} \left| \hat{\mathsf{a}}_k(S_{k+1}, A_{k+1}) \right| , \quad \textit{where } \tilde{C}_\lambda = \frac{2\gamma \sqrt{|\mathcal{S}||\mathcal{A}|}}{1-\gamma} \left( \frac{1 + \lambda \log(|\mathcal{A}|)}{1-\gamma} + \lambda \log(1/\tau_\lambda) + \frac{\lambda}{2\tau_\lambda} \right) .$$

## D. General Analysis of `Ent-AC`: Critic Recursion

The critic step in `Ent-AC` consists of iterating $H$ times a *stochastic* Bellman operator with a step-size $\eta_{\mathsf{c}}$. For completeness, we start by establishing the contractive property of this operator (in contrast, (Kumar et al., 2024) directly assumes its contractivity and does not prove it). Afterwards, we give a bound on the variance of the gradient estimator used to build this stochastic *stochastic* Bellman operator. Next, we give control over the estimated q-function obtained after iterating the $H$-critic steps.

### D.1. Contractivity of the Bellman operator

Firstly, define the expected gradient of the critic and the regularized TD operator as

$$\bar{\mathsf{g}}_{\mathsf{c}}(\theta, \mathsf{q}) \triangleq \mathbb{E}_{X \sim \nu^c(\theta; \cdot)} \left[ \mathsf{g}^X_{\mathsf{c}}(\theta, \mathsf{q}) \right], \qquad \mathsf{T}^{\eta_{\mathsf{c}}}_\theta \mathsf{q} \triangleq \mathsf{q} + \eta_{\mathsf{c}} \mathsf{D}_\theta \left[ \mathsf{r} + \gamma \widetilde{\mathsf{P}}_\theta \left[ \mathsf{q} - \lambda \log(\pi_\theta) \right] - \mathsf{q} \right] , \tag{25}$$

where the diagonal weight matrix $\mathsf{D}_\theta$ is given by

$$\mathsf{D}_\theta = \mathrm{diag}\left( \left( d^\theta_\rho(s) \pi_\theta(a|s) \right)_{(s,a) \in \mathcal{S} \times \mathcal{A}} \right),$$

the extended transition kernel $\widetilde{\mathsf{P}}_\theta \in \mathbb{R}^{|\mathcal{S}||\mathcal{A}| \times |\mathcal{S}||\mathcal{A}|}$ is defined by

$$\widetilde{\mathsf{P}}_\theta(\tilde{s}, \tilde{a} \mid s, a) = \mathsf{P}(\tilde{s} \mid s, a) \pi_\theta(\tilde{a} \mid \tilde{s}), \qquad \forall (s, a, \tilde{s}, \tilde{a}) \in \mathcal{S} \times \mathcal{A} \times \mathcal{S} \times \mathcal{A},$$

and $\log(\pi_\theta) \in \mathbb{R}^{|\mathcal{S}||\mathcal{A}|}$ denotes the vector whose coordinates satisfy

$$[\log(\pi_\theta)]_{(s,a)} = \log\left( \pi_\theta(a \mid s) \right), \qquad \forall (s, a) \in \mathcal{S} \times \mathcal{A}.$$

By construction, $\tilde{q}^\lambda_\theta$ is a fixed point of the operator $\mathsf{T}^{\eta_{\mathsf{c}}}_\theta$.

**Lemma 15.** *Assume $A_\rho$. For any $\theta \in \mathbb{R}^{|\mathcal{S}||\mathcal{A}|}$ such that for all $(s,a) \in \mathcal{S} \times \mathcal{A}$, we have $\pi_\theta(a|s) \geq \pi_{\min}$ for some $\pi_{\min} > 0$, and $v \in \mathbb{R}^{|\mathcal{S}||\mathcal{A}|}$, it holds that*

$$\langle \mathsf{D}_\theta (\mathrm{Id} - \gamma \widetilde{\mathsf{P}}_\theta) v, v \rangle \geq \frac{1}{2}(1-\gamma)^2 \rho_{\min} \pi_{\min} \|v\|_2^2 \ .$$

*Proof.* It holds that

$$
\begin{aligned}
&v^\top \mathsf{D}_\theta (\mathrm{Id} - \gamma \widetilde{\mathsf{P}}_\theta) v \\
&= \sum_{s,a} d_\rho^\theta(s) \pi_\theta(a|s) v(s,a)^2 - \gamma \sum_{(s,a,\tilde{s},\tilde{a})} d_\rho^\theta(s) \pi_\theta(a|s) \mathsf{P}(\tilde{s}|s,a) \pi_\theta(\tilde{a}|\tilde{s}) v(s,a) v(\tilde{s},\tilde{a}) \\
&\geq (1 - \frac{\gamma}{2}) \sum_{s,a} d_\rho^\theta(s) \pi_\theta(a|s) v(s,a)^2 - \frac{\gamma}{2} \sum_{(s,a,\tilde{s},\tilde{a})} d_\rho^\theta(s) \pi_\theta(a|s) \mathsf{P}(\tilde{s}|s,a) \pi_\theta(\tilde{a}|\tilde{s}) v(\tilde{s},\tilde{a})^2 \ , \quad (26)
\end{aligned}
$$

where in the last inequality, we used Young's inequality. Next, using the flow conservation constraints for occupancy measures, see Lemma 24, for any $(\tilde{s}, \tilde{a})$, we have

$$
\begin{aligned}
d_\rho^\theta(\tilde{s}) \pi_\theta(\tilde{a}|\tilde{s}) &= (1-\gamma)\rho(\tilde{s})\pi_\theta(\tilde{a}|\tilde{s}) + \gamma \sum_{(s,a)} \pi_\theta(\tilde{a}|\tilde{s}) \mathsf{P}(\tilde{s}|s,a) \pi_\theta(a|s) d_\rho^\theta(s) \\
&\geq \gamma \sum_{(s,a)} \pi_\theta(\tilde{a}|\tilde{s}) \mathsf{P}(\tilde{s}|s,a) \pi_\theta(a|s) d_\rho^\theta(s) \ .
\end{aligned}
$$

Multiplying both sides by $v(\tilde{s}, \tilde{a})^2$ and summing over $(\tilde{s}, \tilde{a}) \in \mathcal{S} \times \mathcal{A}$, gives

$$\sum_{(s,a,\tilde{s},\tilde{a})} d_\rho^\theta(s) \pi_\theta(a|s) \mathsf{P}(\tilde{s}|s,a) \pi_\theta(\tilde{a}|\tilde{s}) v(\tilde{s},\tilde{a})^2 \leq \frac{1}{\gamma} \sum_{\tilde{s},\tilde{a}} d_\rho^\theta(\tilde{s}) \pi_\theta(\tilde{a}|\tilde{s}) v(\tilde{s},\tilde{a})^2 \ .$$

Plugging in the previous identity in (26), gives

$$v^\top \mathsf{D}_\theta (\mathrm{Id} - \gamma \widetilde{\mathsf{P}}_\theta) v \geq \frac{1}{2}(1-\gamma) \sum_{s,a} d_\rho^\theta(s) \pi_\theta(a|s) v(s,a)^2 \geq \frac{1}{2}(1-\gamma)^2 \rho_{\min} \pi_{\min} \|v\|_2^2 \ ,$$

where the last inequality holds by $A_\rho$ combined with the fact that each entry $\pi_\theta$ is larger than $\pi_{\min}$. $\qquad \square$

**Lemma 16** (Contractivity of $\mathsf{T}_\theta^{\eta_c}$ in $L_2$ norm). *Assume $A_\rho$. Fix $\theta \in \mathbb{R}^{|\mathcal{S}||\mathcal{A}|}$ and $\mathsf{q} \in \mathbb{R}^{|\mathcal{S}||\mathcal{A}|}$. Additionnally, assume that for all $(s,a) \in \mathcal{S} \times \mathcal{A}$, we have $\pi_\theta(a|s) \geq \tau_\lambda$. It holds that*

$$\left\| \tilde{\mathsf{q}}_\theta^\lambda - \mathsf{T}_\theta^{\eta_c} \mathsf{q} \right\|_2^2 \leq \left( 1 - \eta_c(1-\gamma)^2 \rho_{\min} \tau_\lambda + \eta_c^2(1+\gamma)^2 \right) \left\| \tilde{\mathsf{q}}_\theta^\lambda - \mathsf{q} \right\|_2^2 \ .$$

*Proof.* Fix $\theta \in \mathbb{R}^{|\mathcal{S}||\mathcal{A}|}$ and $\mathsf{q} \in \mathbb{R}^{|\mathcal{S}||\mathcal{A}|}$. Using that $\tilde{\mathsf{q}}_\theta^\lambda$ is a fixed point of $\mathsf{T}_\theta^{\eta_c}$ implies that

$$
\begin{aligned}
\left\| \tilde{\mathsf{q}}_\theta^\lambda - \mathsf{T}_\theta^{\eta_c} \mathsf{q} \right\|_2^2 &= \left\| \mathsf{T}_\theta^{\eta_c} \tilde{\mathsf{q}}_\theta^\lambda - \mathsf{T}_\theta^{\eta_c} \mathsf{q} \right\|_2^2 \\
&= \left\| \tilde{\mathsf{q}}_\theta^\lambda + \eta_c \mathsf{D}_\theta \left[ \mathsf{r} + \gamma \widetilde{\mathsf{P}}_\theta [\tilde{\mathsf{q}}_\theta^\lambda - \lambda \log(\pi_\theta)] - \tilde{\mathsf{q}}_\theta^\lambda \right] - \mathsf{q} - \eta_c \mathsf{D}_\theta \left[ \mathsf{r} + \gamma \widetilde{\mathsf{P}}_\theta [\mathsf{q} - \lambda \log(\pi_\theta)] - \mathsf{q} \right] \right\|_2^2 \\
&= \left\| \tilde{\mathsf{q}}_\theta^\lambda + \eta_c \mathsf{D}_\theta \left[ \gamma \widetilde{\mathsf{P}}_\theta \tilde{\mathsf{q}}_\theta^\lambda - \tilde{\mathsf{q}}_\theta^\lambda \right] - \mathsf{q} - \eta_c \mathsf{D}_\theta \left[ \gamma \widetilde{\mathsf{P}}_\theta \mathsf{q} - \mathsf{q} \right] \right\|_2^2 \\
&= \left\| \tilde{\mathsf{q}}_\theta^\lambda - \mathsf{q} \right\|_2^2 + \underbrace{(-2)\eta_c \langle \mathsf{D}_\theta (\mathrm{Id} - \gamma \widetilde{\mathsf{P}}_\theta)(\tilde{\mathsf{q}}_\theta^\lambda - \mathsf{q}), \tilde{\mathsf{q}}_\theta^\lambda - \mathsf{q} \rangle}_{(\mathbf{M})} + \underbrace{\eta_c^2 \left\| \mathsf{D}_\theta (\mathrm{Id} - \gamma \widetilde{\mathsf{P}}_\theta)(\tilde{\mathsf{q}}_\theta^\lambda - \mathsf{q}) \right\|_2^2}_{(\mathbf{N})} \ .
\end{aligned}
$$

**Bounding $(\mathbf{M})$:** To bound $(\mathbf{M})$, we apply Lemma 15 which yields

$$(\mathbf{M}) \leq -\eta_c(1-\gamma)^2 \rho_{\min} \tau_\lambda \left\| \tilde{\mathsf{q}}_\theta^\lambda - \mathsf{q} \right\|_2^2 \ .$$

**Bounding (N):** Define the matrix $A_\theta = \mathsf{D}_\theta(\mathrm{Id} - \gamma\widetilde{\mathsf{P}}_\theta)$. We claim $\|A_\theta\|_2 \leq 1 + \gamma$, where $\|A_\theta\|_2$ is the spectral norm norm of $A_\theta$ (see Section A) for the exact definition). Indeed, sing that $\mathsf{D}_\theta$ is diagonal with nonnegative entries $(d_\rho^\theta(s)\pi_\theta(a|s))_{(s,a)\in\mathcal{S}\times\mathcal{A}}$ that sum to 1, we have for every row $(s,a)$:

$$\|A_\theta\|_\infty = \sum_{(\tilde{s},\tilde{a})} |(A_\theta)_{(s,a),(\tilde{s},\tilde{a})}| = d_\rho^\theta(s)\pi_\theta(a|s) \sum_{(\tilde{s},\tilde{a})\in\mathcal{S}\times\mathcal{A}} |\delta_{(s,a)}(\tilde{s},\tilde{a}) - \gamma\widetilde{\mathsf{P}}_\theta(\tilde{s},\tilde{a} \mid s,a)|$$

$$\leq d_\rho^\theta(s)\pi_\theta(a|s) \sum_{(\tilde{s},\tilde{a})\in\mathcal{S}\times\mathcal{A}} \delta_{(s,a)}(\tilde{s},\tilde{a}) + \gamma\mathsf{P}(\tilde{s} \mid s,a)\,\pi_\theta(\tilde{a} \mid \tilde{s})) = d_\rho^\theta(s)\pi_\theta(a|s)(1+\gamma) \leq 1 + \gamma.$$

Hence $\|A_\theta\|_\infty \leq 1 + \gamma$. Similarly for every column $(\tilde{s},\tilde{a})$:

$$\sum_{(s,a)\in\mathcal{S}\times\mathcal{A}} |(A_\theta)_{(s,a),(\tilde{s},\tilde{a})}| \leq \sum_{(s,a)\in\mathcal{S}\times\mathcal{A}} d_\rho^\theta(s)\pi_\theta(a|s)(\delta_{(s,a)}(\tilde{s},\tilde{a}) + \gamma\mathsf{P}(\tilde{s} \mid s,a)\,\pi_\theta(\tilde{a} \mid \tilde{s}))$$

$$= d_\rho^\theta(\tilde{s})\pi_\theta(\tilde{a}|\tilde{s}) + \gamma \sum_{(s,a)\in\mathcal{S}\times\mathcal{A}} d_\rho^\theta(s)\pi_\theta(a|s)\mathsf{P}(\tilde{s} \mid s,a)\pi_\theta(\tilde{a} \mid \tilde{s})$$

$$\leq 1 + \gamma \ ,$$

so $\|A_\theta\|_1 \leq 1 + \gamma$. Using the standard inequality $\|A_\theta\|_2 \leq \sqrt{\|A_\theta\|_1\|A_\theta\|_\infty}$ gives

$$\|A_\theta\|_2 \leq 1 + \gamma.$$

Therefore

$$(\mathbf{N}) \leq \eta_{\mathsf{c}}^2\|\mathsf{D}_\theta(\mathrm{Id} - \gamma\widetilde{\mathsf{P}}_\theta)(\tilde{\mathsf{q}}_\theta^\lambda - \mathsf{q})\|_2^2 \leq (1+\gamma)^2\|(\tilde{\mathsf{q}}_\theta^\lambda - \mathsf{q})\|_2^2 \ .$$

Combining the two previous bounds concludes the proof. $\qquad\square$

## D.2. Properties of the stochastic gradient

Next, we provide properties on the gradient estimator used in the critic's update.

**Lemma 17** (Properties of the critic estimator). *Fix $\theta \in \mathbb{R}^{|\mathcal{S}||\mathcal{A}|}$ and $\mathsf{q} \in \mathbb{R}^{|\mathcal{S}||\mathcal{A}|}$. Then*

$$\bar{\mathsf{g}}_{\mathsf{c}}(\theta, \mathsf{q}) = \frac{\mathsf{T}_\theta^{\eta_{\mathsf{c}}}\mathsf{q} - \mathsf{q}}{\eta_{\mathsf{c}}},$$

*and*

$$\mathbb{E}_{X\sim\nu^{\mathsf{c}}(\theta)}\left[\left\|\mathsf{g}_{\mathsf{c}}^X(\theta, \mathsf{q}) - \bar{\mathsf{g}}_{\mathsf{c}}(\theta, \mathsf{q})\right\|_2^2\right] \leq 8\|\mathsf{q}\|_\infty^2 + 4 + 4\lambda^2 + 4\lambda^2\log(|\mathcal{A}|)^2.$$

*Proof.* Fix $\theta \in \mathbb{R}^{|\mathcal{S}||\mathcal{A}|}$ and $\mathsf{q} \in \mathbb{R}^{|\mathcal{S}||\mathcal{A}|}$. We first compute the mean of the critic estimator. For any $(s,a) \in \mathcal{S} \times \mathcal{A}$, we have

$$[\bar{\mathsf{g}}_{\mathsf{c}}(\theta, \mathsf{q})]_{s,a} = \left[\mathbb{E}_{X\sim\nu^{\mathsf{c}}(\theta;\cdot)}\left[\mathsf{g}_{\mathsf{c}}^X(\theta, \mathsf{q})\right]\right]_{s,a}$$

$$= \sum_{(s',a',\tilde{s}',\tilde{a}')} d_\rho^\theta(s')\,\pi_\theta(a'|s')\,\mathsf{P}(\tilde{s}'|s',a')\,\pi_\theta(\tilde{a}'|\tilde{s}')\,\mathbf{1}_{(s,a)}(s',a')$$

$$\times \left[\mathsf{r}(s',a') + \gamma\big(\mathsf{q}(\tilde{s}',\tilde{a}') - \lambda\log\pi_\theta(\tilde{a}'|\tilde{s}')\big) - \mathsf{q}(s',a')\right]$$

$$= d_\rho^\theta(s)\,\pi_\theta(a|s)\left[\mathsf{r}(s,a) + \gamma\sum_{(\tilde{s},\tilde{a})} \mathsf{P}(\tilde{s}|s,a)\,\pi_\theta(\tilde{a}|\tilde{s})\right.$$

$$\left. \times \big(\mathsf{q}(\tilde{s},\tilde{a}) - \lambda\log\pi_\theta(\tilde{a}|\tilde{s})\big) - \mathsf{q}(s,a)\right]$$

$$= \left[\mathsf{D}_\theta\Big(\mathsf{r} + \gamma\widetilde{\mathsf{P}}_\theta[\mathsf{q} - \lambda\log(\pi_\theta)] - \mathsf{q}\Big)\right]_{s,a}.$$

By the definition of $\mathsf{T}_\theta^{\eta_c}$, this proves that

$$\bar{\mathsf{g}}_c(\theta, \mathsf{q}) = \frac{\mathsf{T}_\theta^{\eta_c}\mathsf{q} - \mathsf{q}}{\eta_c}.$$

We now prove the second-moment bound. Using

$$\mathbb{E}\big[\|X - \mathbb{E}[X]\|_2^2\big] \leq \mathbb{E}[\|X\|_2^2],$$

we obtain

$$\mathbb{E}_{X \sim \nu^c(\theta)}\left[\left\|\mathsf{g}_c^X(\theta, \mathsf{q}) - \bar{\mathsf{g}}_c(\theta, \mathsf{q})\right\|_2^2\right] \leq \mathbb{E}_{X \sim \nu^c(\theta)}\left[\left\|\mathsf{g}_c^X(\theta, \mathsf{q})\right\|_2^2\right].$$

Expanding the last term gives

$$\mathbb{E}_{X \sim \nu^c(\theta)}\left[\left\|\mathsf{g}_c^X(\theta, \mathsf{q})\right\|_2^2\right]$$

$$= \mathbb{E}_{X \sim \nu^c(\theta)}\left[\sum_{(s,a) \in \mathcal{S} \times \mathcal{A}} \left(\mathbf{1}_{(s,a)}(S, A)\Big[\mathsf{r}(S, A) + \gamma\big(\mathsf{q}(\tilde{S}, \tilde{A}) - \lambda \log \pi_\theta(\tilde{A}|\tilde{S})\big) - \mathsf{q}(S, A)\Big]\right)^2\right]$$

$$= \sum_{(s,a,\tilde{s},\tilde{a})} d_\rho^\theta(s)\, \pi_\theta(a|s)\, \mathsf{P}(\tilde{s}|s, a)\, \pi_\theta(\tilde{a}|\tilde{s})$$

$$\times \big[\mathsf{r}(s,a) + \gamma\big(\mathsf{q}(\tilde{s}, \tilde{a}) - \lambda \log \pi_\theta(\tilde{a}|\tilde{s})\big) - \mathsf{q}(s, a)\big]^2.$$

Using

$$(a + b + c + d)^2 \leq 4(a^2 + b^2 + c^2 + d^2),$$

we deduce

$$\mathbb{E}_{X \sim \nu^c(\theta)}\left[\left\|\mathsf{g}_c^X(\theta, \mathsf{q})\right\|_2^2\right] \leq 4 \sum_{(s,a,\tilde{s},\tilde{a})} d_\rho^\theta(s)\, \pi_\theta(a|s)\, \mathsf{P}(\tilde{s}|s, a)\, \pi_\theta(\tilde{a}|\tilde{s})$$

$$\times \left(\mathsf{r}(s,a)^2 + \gamma^2\mathsf{q}(\tilde{s}, \tilde{a})^2 + \gamma^2\lambda^2 \log \pi_\theta(\tilde{a}|\tilde{s})^2 + \mathsf{q}(s, a)^2\right).$$

Since the reward is bounded by 1 and $\gamma \leq 1$, this yields

$$\mathbb{E}_{X \sim \nu^c(\theta)}\left[\left\|\mathsf{g}_c^X(\theta, \mathsf{q})\right\|_2^2\right]$$

$$\leq 4(\gamma^2 + 1)\|\mathsf{q}\|_\infty^2 + 4\sum_{s,a} d_\rho^\theta(s)\, \pi_\theta(a|s) + 4\lambda^2 \max_{\tilde{s} \in \mathcal{S}} \sum_{\tilde{a} \in \mathcal{A}} \pi_\theta(\tilde{a}|\tilde{s}) \log \pi_\theta(\tilde{a}|\tilde{s})^2$$

$$\leq 8\|\mathsf{q}\|_\infty^2 + 4 + 4\lambda^2 \max_{\tilde{s} \in \mathcal{S}} \sum_{\tilde{a} \in \mathcal{A}} \pi_\theta(\tilde{a}|\tilde{s}) \log \pi_\theta(\tilde{a}|\tilde{s})^2.$$

Finally, using that for any probability vector $p \in \mathcal{P}(\mathcal{A})$ (see Lemma 31), we have

$$\sum_{a \in \mathcal{A}} p(a) \log(p(a))^2 \leq 1 + \log(|\mathcal{A}|)^2 \ ,$$

concludes the proof. $\qquad\square$

### D.3. Critic recursion

Define the filtration adapted to the local iterates of the critic

$$\mathcal{F}_k^h \triangleq \sigma\Big(X_\ell, Y_\ell : \ell \in \{1, \ldots, k-1\}, X_k^p : p \in \{1, \ldots, h\}, Y_k\Big) \ ,$$

Additionally, define:

$$\tilde{\mu}_c \triangleq (1 - \gamma)^2 \rho_{\min} \tau_\lambda / 2 \ , \quad \text{and} \quad \sigma_c^2 \triangleq \frac{36 + 4\lambda^2 + 36\lambda^2 \log(|\mathcal{A}|)^2}{(1 - \gamma)^2} \ . \tag{27}$$

The following lemma bounds the variance and the bias of the critic after performing $H$ during iteration $k$.

**Lemma 18.** *Assume $A_\rho$ and assume that $\eta_c \le (1-\gamma)^2\rho_{\min}\tau_\lambda/40$. It holds that*

$$\mathbb{E}\left[\left\|\hat{q}_k^H - \tilde{q}_{\theta_k}^\lambda\right\|_2^2 \big| \mathcal{F}_k\right] \le (1-\eta_c\tilde{\mu}_c)^H \left\|\hat{q}_k^0 - \tilde{q}_{\theta_k}^\lambda\right\|_2^2 + \frac{\eta_c\sigma_c^2}{\tilde{\mu}_c}(1-(1-\eta_c\tilde{\mu}_c)^H) \ ,$$

*where $\tilde{\mu}_c$, and $\sigma_c^2$ are defined in* (27). *Additionally, we have that*

$$\left\|\mathbb{E}\left[\hat{q}_k^H|\mathcal{F}_k\right] - \tilde{q}_{\theta_k}^\lambda\right\|_2^2 \le (1-\eta_c\tilde{\mu}_c)^H \left\|\hat{q}_k^0 - \tilde{q}_{\theta_k}^\lambda\right\|_2^2 \ .$$

*Proof.* Fix $k \in \{0,\dots,K-1\}$ and $h \in \{0,\dots,H-1\}$. It holds that

$$\left\|\hat{q}_k^{h+1} - \tilde{q}_{\theta_k}^\lambda\right\|_2^2 = \left\|\hat{q}_k^h + \eta_c\, g_c^{X_k^{h+1}}(\theta_k,\hat{q}_k^h) - \tilde{q}_{\theta_k}^\lambda\right\|_2^2$$

$$= \left\|\hat{q}_k^h - \tilde{q}_{\theta_k}^\lambda\right\|_2^2 + 2\eta_c\langle g_c^{X_k^{h+1}}(\theta_k,\hat{q}_k^h), \hat{q}_k^h - \tilde{q}_{\theta_k}^\lambda\rangle + \eta_c^2\left\|g_c^{X_k^{h+1}}(\theta_k,\hat{q}_k^h)\right\|_2^2 \ .$$

Taking the conditional expectation with respect to $\mathcal{F}_k^h$, and using the fact that $\bar{g}_c(\theta_k,\hat{q}_k^h) = (T_{\theta_k}^{\eta_c}\hat{q}_k^h - \hat{q}_k^h)/\eta_c$ (Lemma 17) yields

$$\mathbb{E}\left[\left\|\hat{q}_k^{h+1} - \tilde{q}_{\theta_k}^\lambda\right\|_2^2 \big| \mathcal{F}_k^h\right] = \left\|\hat{q}_k^h - \tilde{q}_{\theta_k}^\lambda\right\|_2^2 + 2\eta_c\langle \bar{g}_c(\theta_k,\hat{q}_k^h), \hat{q}_k^h - \tilde{q}_{\theta_k}^\lambda\rangle + \eta_c^2\mathbb{E}\left[\left\|g_c^{X_k^{h+1}}(\theta_k,\hat{q}_k^h)\right\|_2^2 \big| \mathcal{F}_k^h\right]$$

$$= \left\|T_{\theta_k}^{\eta_c}\hat{q}_k^h - \tilde{q}_{\theta_k}^\lambda\right\|_2^2 + \eta_c^2\mathbb{E}\left[\left\|g_c^{X_k^{h+1}}(\theta_k,\hat{q}_k^h) - \bar{g}_c(\theta_k,\hat{q}_k^h)\right\|_2^2 \big| \mathcal{F}_k^h\right]$$

$$\le (1-(1-\gamma)^2\eta_c\rho_{\min}\tau_\lambda + \eta_c^2(1+\gamma)^2)\left\|\hat{q}_k^h - \tilde{q}_{\theta_k}^\lambda\right\|_2^2 + \eta_c^2\left(8\|\hat{q}_k^h\|_\infty^2 + 4 + 4\lambda^2 + 4\lambda^2\log(|\mathcal{A}|)^2\right) \ ,$$

where in the last identity, we used the contractivity of $T_{\theta_k}^{\eta_c}$ (Lemma 16) combined with the bound on the variance of the critic (Lemma 17). Next, by Young's inequality, we have $\|\hat{q}_k^h\|_\infty^2 \le 2\|\hat{q}_k^h - \tilde{q}_{\theta_k}^\lambda\|_\infty^2 + 2\|\tilde{q}_{\theta_k}^\lambda\|_\infty^2$, which implies that

$$\mathbb{E}\left[\left\|\hat{q}_k^{h+1} - \tilde{q}_{\theta_k}^\lambda\right\|_2^2 \big| \mathcal{F}_k^h\right] \le (1-(1-\gamma)^2\eta_c\rho_{\min}\tau_\lambda + 20\eta_c^2)\left\|\hat{q}_k^h - \tilde{q}_{\theta_k}^\lambda\right\|_2^2 + \eta_c^2\left(16\|\tilde{q}_{\theta_k}^\lambda\|_\infty^2 + 4 + 4\lambda^2 + 4\lambda^2\log(|\mathcal{A}|)^2\right) \ .$$

Taking the conditional expectation, with respect to $\mathcal{F}_k$, the fact that $\eta_c \le (1-\gamma)^2\rho_{\min}\tau_\lambda/40$, $\|\tilde{q}_{\theta_k}^\lambda\|_\infty^2 \le (2 + 2\lambda^2\log(|\mathcal{A}|)^2)/(1-\gamma)^2$ (which holds by a combination of Lemma 27 and then Young's inequality) gives

$$\mathbb{E}\left[\left\|\hat{q}_k^{h+1} - \tilde{q}_{\theta_k}^\lambda\right\|_2^2 \big| \mathcal{F}_k\right] \le (1-(1-\gamma)^2\eta_c\rho_{\min}\tau_\lambda/2)\mathbb{E}\left[\left\|\hat{q}_k^h - \tilde{q}_{\theta_k}^\lambda\right\|_2^2 \big| \mathcal{F}_k\right] + \eta_c^2\left(36 + 4\lambda^2 + 36\lambda^2\log(|\mathcal{A}|)^2\right)/(1-\gamma)^2 \ .$$

Finally, unrolling the recursion yields the result, and the second claim follows from the affinity of $T_{\theta_k}^{\eta_c}$. $\qquad\square$

The next lemma bounds the mean-squared error of the critic.

**Lemma 19.** *Assume $A_\rho$ and*

$$\eta_c \le (1-\gamma)^2\rho_{\min}\tau_\lambda/40 \ , \quad\text{and}\quad H \ge \frac{2}{\eta_c\tilde{\mu}_c}\log(2 + 4\tilde{C}_\lambda^2\eta_a^2) \ . \tag{28}$$

*For any $k \ge 0$, it holds that*

$$\mathbb{E}\left[\left\|\hat{q}_k - \tilde{q}_{\theta_k}^\lambda\right\|_2^2\right] \le (1-\eta_c\tilde{\mu}_c)^{H(k+1)/2}\left\|\hat{q}_{-1} - \tilde{q}_{\theta_0}^\lambda\right\|_2^2 + \frac{\tilde{C}_\lambda^2\eta_a^2(16 + 8\lambda^2 + 24\lambda^2\log(|\mathcal{A}|)^2)}{(1-\gamma)^2}\cdot\frac{1}{1-(1-\eta_c\tilde{\mu}_c)^{H/2}} + \frac{2\eta_c\sigma_c^2}{\tilde{\mu}_c} \ .$$

*Proof.* Firstly, applying Lemma 18 combined with the tower property of the conditional expectation, gives

$$\mathbb{E}\left[\left\|\hat{q}_k - \tilde{q}_{\theta_k}^\lambda\right\|_2^2\right] \le (1-\eta_c\tilde{\mu}_c)^H\mathbb{E}\left[\left\|\hat{q}_{k-1} - \tilde{q}_{\theta_k}^\lambda\right\|_2^2\right] + \frac{\eta_c\sigma_c^2}{\tilde{\mu}_c}(1-(1-\eta_c\tilde{\mu}_c)^H) \ .$$

Thus, the bound holds in the case where $k = 0$. Now consider the case where $k \geq 1$. Adding and subtracting $\tilde{q}^\lambda_{\theta_{k-1}}$ inside the squared norm of the right-hand side, and applying Jensen's inequality gives

$$\mathbb{E}\left[\left\|\hat{q}_k - \tilde{q}^\lambda_{\theta_k}\right\|_2^2\right] \leq (1 - \eta_c\tilde{\mu}_c)^H \left[2\mathbb{E}\left[\left\|\hat{q}_{k-1} - \tilde{q}^\lambda_{\theta_{k-1}}\right\|_2^2\right] + 2\mathbb{E}\left[\left\|\tilde{q}^\lambda_{\theta_{k-1}} - \tilde{q}^\lambda_{\theta_k}\right\|_2^2\right]\right] + \frac{\eta_c\sigma_c^2}{\tilde{\mu}_c}(1 - (1 - \eta_c\tilde{\mu}_c)^H) \ .$$

Applying Corollary 4 yields

$$\mathbb{E}\left[\left\|\hat{q}_k - \tilde{q}^\lambda_{\theta_k}\right\|_2^2\right] \leq (1 - \eta_c\tilde{\mu}_c)^H \left[2\mathbb{E}\left[\left\|\hat{q}_{k-1} - \tilde{q}^\lambda_{\theta_{k-1}}\right\|_2^2\right] + 2\tilde{C}_\lambda^2\eta_a^2\mathbb{E}\left[\hat{a}_{k-1}(S_k, A_k)^2\right]\right] + \frac{\eta_c\sigma_c^2}{\tilde{\mu}_c}(1 - (1 - \eta_c\tilde{\mu}_c)^H) \ . \quad (29)$$

Next, using Jensen's inequality combined with the definition of the true regularized advantage (6) and the estimated regularized advantage (13) gives

$$\mathbb{E}\left[\hat{a}_{k-1}(S_k, A_k)^2\right]$$
$$\leq 2\mathbb{E}\left[\left(\hat{a}_{k-1}(S_k, A_k) - \tilde{a}^\lambda_{\theta_{k-1}}(S_k, A_k)\right)^2\right] + 2\mathbb{E}\left[\tilde{a}^\lambda_{\theta_{k-1}}(S_k, A_k)^2\right]$$
$$\leq 2\mathbb{E}\left[\sum_{s\in\mathcal{S}}\sum_{a\in\mathcal{A}}d_\rho^{\theta_{k-1}}(s)\pi_{\theta_{k-1}}(a|s)\Big(\hat{q}_{k-1}(s,a) - \lambda\log(\pi_{\theta_{k-1}}(a|s)) - \hat{v}_{k-1}(s)\right.$$
$$\left. - (\tilde{q}^\lambda_{\theta_{k-1}}(s,a) - \lambda\log(\pi_{\theta_{k-1}}(a|s)) - \tilde{v}^\lambda_{\theta_{k-1}}(s))\Big)^2\right]$$
$$+ 2\mathbb{E}\left[\sum_{s\in\mathcal{S}}\sum_{a\in\mathcal{A}}d_\rho^{\theta_{k-1}}(s)\pi_{\theta_{k-1}}(a|s)(\tilde{q}^\lambda_{\theta_{k-1}}(s,a) - \lambda\log(\pi_{\theta_{k-1}}(a|s)) - \tilde{v}^\lambda_{\theta_{k-1}}(s))^2\right] \ .$$

Next, using that for any $p \in \mathcal{P}(\mathcal{A})$, and $x_1, \ldots, x_{|\mathcal{A}|} \in \mathbb{R}$, we have $\sum_{i=1}^{|\mathcal{A}|} p_i(x_i - \bar{x})^2 \leq \sum_{i=1}^{|\mathcal{A}|} p_i x_i^2$, where $\bar{x} = \sum_{i=1}^{|\mathcal{A}|} p_i x_i$, gives

$$\mathbb{E}\left[\hat{a}_{k-1}(S_k, A_k)^2\right] \leq 2\mathbb{E}\left[\sum_{s\in\mathcal{S}}\sum_{a\in\mathcal{A}}d_\rho^{\theta_{k-1}}(s)\pi_{\theta_{k-1}}(a|s)\left(\hat{q}_{k-1}(s,a) - \tilde{q}^\lambda_{\theta_{k-1}}(s,a)\right)^2\right]$$
$$+ 2\mathbb{E}\left[\sum_{s\in\mathcal{S}}\sum_{a\in\mathcal{A}}d_\rho^{\theta_{k-1}}(s)\pi_{\theta_{k-1}}(a|s)(\tilde{q}^\lambda_{\theta_{k-1}}(s,a) - \lambda\log(\pi_{\theta_{k-1}}(a|s)))^2\right]$$
$$\leq 2\mathbb{E}\left[\left\|\hat{q}_{k-1} - \tilde{q}^\lambda_{\theta_{k-1}}\right\|_2^2\right] + \frac{8(1 + \lambda^2\log(|\mathcal{A}|)^2)}{(1 - \gamma)^2} + 4\lambda^2 + 4\lambda^2\log(|\mathcal{A}|)^2 \ , \quad (30)$$

where in the last inequality, we used Young's inequality combined with Lemma 27 and Lemma 31. Plugging in the previous inequality in Equation (29) gives

$$\mathbb{E}\left[\left\|\hat{q}_k - \tilde{q}^\lambda_{\theta_k}\right\|_2^2\right] \leq (1 - \eta_c\tilde{\mu}_c)^H \left[(2 + 4\tilde{C}_\lambda^2\eta_a^2)\mathbb{E}\left[\left\|\hat{q}_{k-1} - \tilde{q}^\lambda_{\theta_{k-1}}\right\|_2^2\right] + \frac{\tilde{C}_\lambda^2\eta_a^2(16 + 8\lambda^2 + 24\lambda^2\log(|\mathcal{A}|)^2)}{(1 - \gamma)^2}\right]$$
$$+ \frac{\eta_c\sigma_c^2}{\tilde{\mu}_c}(1 - (1 - \eta_c\tilde{\mu}_c)^H) \ .$$

Next, unrolling the previous recursion yields

$$\mathbb{E}\left[\left\|\hat{q}_k - \tilde{q}^\lambda_{\theta_k}\right\|_2^2\right] \leq \left[(1 - \eta_c\tilde{\mu}_c)^H \cdot (2 + 4\tilde{C}_\lambda^2\eta_a^2)\right]^{k+1}\left\|\hat{q}_{-1} - \tilde{q}^\lambda_{\theta_0}\right\|_2^2$$
$$+ \frac{\tilde{C}_\lambda^2\eta_a^2(16 + 8\lambda^2 + 24\lambda^2\log(|\mathcal{A}|)^2)}{(1 - \gamma)^2}\sum_{\ell=0}^k \left[(1 - \eta_c\tilde{\mu}_c)^H \cdot (2 + 4\tilde{C}_\lambda^2\eta_a^2)\right]^\ell$$
$$+ \frac{\eta_c\sigma_c^2}{\tilde{\mu}_c}(1 - (1 - \eta_c\tilde{\mu}_c)^H)\sum_{\ell=0}^k \left[(1 - \eta_c\tilde{\mu}_c)^H \cdot (2 + 4\tilde{C}_\lambda^2\eta_a^2)\right]^\ell \ .$$

As $H \geq \frac{2}{\eta_{\mathsf{c}}\tilde{\mu}_{\mathsf{c}}} \log(2 + 4\tilde{C}_\lambda^2 \eta_{\mathsf{a}}^2)$, then $(1 - \eta_{\mathsf{c}}\tilde{\mu}_{\mathsf{c}})^H \cdot (2 + 4\tilde{C}_\lambda^2 \eta_{\mathsf{a}}^2) \leq (1 - \eta_{\mathsf{c}}\tilde{\mu}_{\mathsf{c}})^{H/2}$ which implies

$$\mathbb{E}\left[\left\|\hat{\mathsf{q}}_k - \tilde{\mathsf{q}}_{\theta_k}^\lambda\right\|_2^2\right] \leq (1 - \eta_{\mathsf{c}}\tilde{\mu}_{\mathsf{c}})^{H(k+1)/2} \left\|\hat{\mathsf{q}}_{-1} - \tilde{\mathsf{q}}_{\theta_0}^\lambda\right\|_2^2 + \frac{\tilde{C}_\lambda^2 \eta_{\mathsf{a}}^2 (16 + 8\lambda^2 + 24\lambda^2 \log(|\mathcal{A}|)^2)}{(1 - \gamma)^2} \sum_{\ell=0}^k (1 - \eta_{\mathsf{c}}\tilde{\mu}_{\mathsf{c}})^{H\ell/2}$$

$$+ \frac{\eta_{\mathsf{c}}\sigma_{\mathsf{c}}^2}{\tilde{\mu}_{\mathsf{c}}}(1 - (1 - \eta_{\mathsf{c}}\tilde{\mu}_{\mathsf{c}})^H) \sum_{\ell=0}^k (1 - \eta_{\mathsf{c}}\tilde{\mu}_{\mathsf{c}})^{H\ell/2}$$

$$\leq (1 - \eta_{\mathsf{c}}\tilde{\mu}_{\mathsf{c}})^{H(k+1)/2} \left\|\hat{\mathsf{q}}_{-1} - \tilde{\mathsf{q}}_{\theta_0}^\lambda\right\|_2^2 + \frac{\tilde{C}_\lambda^2 \eta_{\mathsf{a}}^2 (16 + 8\lambda^2 + 24\lambda^2 \log(|\mathcal{A}|)^2)}{(1 - \gamma)^2} \cdot \frac{1 - (1 - \eta_{\mathsf{c}}\tilde{\mu}_{\mathsf{c}})^{H(k+1)/2}}{1 - (1 - \eta_{\mathsf{c}}\tilde{\mu}_{\mathsf{c}})^{H/2}}$$

$$+ \frac{\eta_{\mathsf{c}}\sigma_{\mathsf{c}}^2}{\tilde{\mu}_{\mathsf{c}}}(1 - (1 - \eta_{\mathsf{c}}\tilde{\mu}_{\mathsf{c}})^H) \cdot \frac{1 - (1 - \eta_{\mathsf{c}}\tilde{\mu}_{\mathsf{c}})^{H(k+1)/2}}{1 - (1 - \eta_{\mathsf{c}}\tilde{\mu}_{\mathsf{c}})^{H/2}} \quad .$$

Finally, using that $\frac{1 - (1 - \eta_{\mathsf{c}}\tilde{\mu}_{\mathsf{c}})^H}{1 - (1 - \eta_{\mathsf{c}}\tilde{\mu}_{\mathsf{c}})^{H/2}} \leq 2$ concludes the proof. $\qquad\square$

We define

$$B := 2\left\|\hat{\mathsf{q}}_{-1} - \tilde{\mathsf{q}}_{\theta_0}^\lambda\right\|_2^2 + \frac{2\tilde{C}_\lambda^2 \rho_{\min}^2 \tau_\lambda^2 (2 + \lambda^2 + 3\lambda^2 \log(|\mathcal{A}|)^2)}{L^2} + \frac{2(1 - \gamma)^2 \rho_{\min}\tau_\lambda\sigma_{\mathsf{c}}^2}{20\tilde{\mu}_{\mathsf{c}}} + \frac{2|\mathcal{S}||\mathcal{A}|(1 + \lambda^2 \log(|\mathcal{A}|)^2)}{(1 - \gamma)^2} \quad . \tag{31}$$

**Corollary 5.** *Assume* $A_\rho$,

$$\eta_{\mathsf{a}} \leq \frac{(1 - \gamma)\rho_{\min}\tau_\lambda}{8L} \quad ,$$

*and that* $\eta_{\mathsf{c}}$ *and* $H$ *satisfy the condition* (28). *Then, for any* $k \geq 1$, *it holds that*

$$\frac{1}{1 - (1 - \eta_{\mathsf{c}}\tilde{\mu}_{\mathsf{c}})^{H/2}} \leq 2, \qquad \mathbb{E}\left[\left\|\hat{\mathsf{q}}_{k-1} - \tilde{\mathsf{q}}_{\theta_k}^\lambda\right\|_2^2\right] \leq B.$$

*Proof.* We bound the two terms separately.

**Bound on the first term.** Using

$$H \geq \frac{2}{\eta_{\mathsf{c}}\tilde{\mu}_{\mathsf{c}}} \log\left(2 + 4\tilde{C}_\lambda^2 \eta_{\mathsf{a}}^2\right)$$

and the inequality $1 - x \leq e^{-x}$ for $x \geq 0$, we obtain

$$(1 - \eta_{\mathsf{c}}\tilde{\mu}_{\mathsf{c}})^{H/2} \leq \exp(-\eta_{\mathsf{c}}\tilde{\mu}_{\mathsf{c}}H/2) \leq \frac{1}{2 + 4\tilde{C}_\lambda^2 \eta_{\mathsf{a}}^2} \leq \frac{1}{2} \quad .$$

Therefore,

$$\frac{1}{1 - (1 - \eta_{\mathsf{c}}\tilde{\mu}_{\mathsf{c}})^{H/2}} \leq 2. \tag{32}$$

**Bound on the second term.** Applying Young's inequality, it holds that

$$\mathbb{E}\left[\left\|\hat{\mathsf{q}}_{k-1} - \tilde{\mathsf{q}}_{\theta_k}^\lambda\right\|_2^2\right] \leq 2\underbrace{\mathbb{E}\left[\left\|\hat{\mathsf{q}}_{k-1} - \tilde{\mathsf{q}}_{\theta_{k-1}}^\lambda\right\|_2^2\right]}_{(\mathbf{W})} + 2\underbrace{\mathbb{E}\left[\left\|\tilde{\mathsf{q}}_{\theta_{k-1}}^\lambda - \tilde{\mathsf{q}}_{\theta_k}^\lambda\right\|_2^2\right]}_{(\mathbf{Z})} \quad .$$

*Bounding* $(\mathbf{W})$. Applying Lemma 19, yields

$$\mathbb{E}\left[\left\|\hat{\mathsf{q}}_{k-1} - \tilde{\mathsf{q}}_{\theta_{k-1}}^\lambda\right\|_2^2\right] \leq (1 - \eta_{\mathsf{c}}\tilde{\mu}_{\mathsf{c}})^{Hk/2} \left\|\hat{\mathsf{q}}_{-1} - \tilde{\mathsf{q}}_{\theta_0}^\lambda\right\|_2^2 + \frac{\tilde{C}_\lambda^2 \eta_{\mathsf{a}}^2 (16 + 8\lambda^2 + 24\lambda^2 \log(|\mathcal{A}|)^2)}{(1 - \gamma)^2} \cdot \frac{1}{1 - (1 - \eta_{\mathsf{c}}\tilde{\mu}_{\mathsf{c}})^{H/2}}$$

$$+ \frac{2\eta_c \sigma_c^2}{\tilde{\mu}_c} \ .$$

Using Equation (32), combined with the fact that $1 - \eta_c \tilde{\mu}_c \leq 1$ yields

$$(\mathbf{W}) \leq \left\| \hat{q}_{-1} - \tilde{q}_{\theta_0}^\lambda \right\|_2^2 + \frac{\tilde{C}_\lambda^2 \eta_a^2 (32 + 16\lambda^2 + 48\lambda^2 \log(|\mathcal{A}|)^2)}{(1-\gamma)^2} + \frac{2\eta_c \sigma_c^2}{\tilde{\mu}_c} \ .$$

Plugging in the conditions on $\eta_a$ and $\eta_c$, we get

$$(\mathbf{W}) \leq \left\| \hat{q}_{-1} - \tilde{q}_{\theta_0}^\lambda \right\|_2^2 + \frac{\tilde{C}_\lambda^2 \rho_{\min}^2 \tau_\lambda^2 (2 + \lambda^2 + 3\lambda^2 \log(|\mathcal{A}|)^2)}{L^2} + \frac{(1-\gamma)^2 \rho_{\min} \tau_\lambda \sigma_c^2}{20\tilde{\mu}_c} \ .$$

*Bounding* $(\mathbf{Z})$. Using that $\left\| \tilde{q}_{\theta_{k-1}}^\lambda - \tilde{q}_{\theta_k}^\lambda \right\|_\infty \leq (1 + \lambda \log(|\mathcal{A}|))/(1-\gamma)$, combined with the fact that for any $x \in \mathbb{R}^{|\mathcal{S}||\mathcal{A}|}$, we have $\|x\|_\infty \leq |\mathcal{S}||\mathcal{A}| \|x\|_2$, gives

$$(\mathbf{Z}) \leq 2|\mathcal{S}||\mathcal{A}|(1 + \lambda^2 \log(|\mathcal{A}|)^2)/(1-\gamma)^2 \ .$$

Combining the two previous bounds concludes the proof $\hfill \square$

## E. General Analysis of `Ent-AC`: Combining the Two Recursions

Next, we derive the convergence rate of `Ent-AC`. In this section, we prove the final theorem that combines both the actor recursion and the critic recursion.

**Theorem 4.** *Assume $\mathbf{A}_\rho$ and assume that $\eta_c \leq (1-\gamma)^2 \rho_{\min} \tau_\lambda / 40$, $H \geq \frac{2}{\eta_c \tilde{\mu}_c} \log(2 + 4\tilde{C}_\lambda^2 \eta_a^2)$, and that $\eta_a \leq \frac{(1-\gamma)\rho_{\min}\tau_\lambda}{8L}$. For any $k \geq 0$, it holds that*

$$\mathbb{E}\left[ \tilde{J}_\lambda^\star - \tilde{J}_\lambda(\theta_K) \right] \leq \left( 1 - \frac{\eta_a \underline{\tilde{\mu}}_\lambda}{8} \right)^K \left[ \tilde{J}_\lambda^\star - \tilde{J}_\lambda(\theta_0) \right] + \frac{2L\eta_a^2}{(1-\gamma)^2} K \max\left( 1 - \frac{\eta_a \underline{\tilde{\mu}}_\lambda}{8}, (1 - \eta_c \tilde{\mu}_c)^{H/2} \right)^K \left\| \hat{q}_{-1} - \tilde{q}_{\theta_0}^\lambda \right\|_2^2$$

$$+ \frac{16(1 - \eta_c \tilde{\mu}_c)^H}{\underline{\tilde{\mu}}_\lambda (1-\gamma)^2} B + \eta_a^3 \frac{256\tilde{C}_\lambda^2 L (1 + \lambda^2 \log(|\mathcal{A}|)^2)}{\underline{\tilde{\mu}}_\lambda (1-\gamma)^4} + \frac{32L\eta_a \eta_c \sigma_c^2}{(1-\gamma)^2 \tilde{\mu}_c \underline{\tilde{\mu}}_\lambda} \ .$$

*Proof.* Using Lemma 13, it holds that

$$\mathbb{E}\left[ \tilde{J}_\lambda^\star - \tilde{J}_\lambda(\theta_{k+1}) \big| \mathcal{F}_k \right] \leq \tilde{J}_\lambda^\star - \tilde{J}_\lambda(\theta_k) - \left( \frac{\eta_a}{2} - \frac{2L\eta_a^2}{(1-\gamma)\rho_{\min}\tau_\lambda} \right) \left\| \nabla \tilde{J}_\lambda(\theta_k) \right\|_2^2$$

$$+ \frac{2\eta_a}{(1-\gamma)^2} \sum_{(s,a)\in\mathcal{S}\times\mathcal{A}} d_\rho^{\theta_k}(s)^2 \pi_{\theta_k}(a|s)^2 \left( \mathbb{E}[\hat{a}_k(s,a)|\mathcal{F}_k] - \tilde{a}_{\theta_k}^\lambda(s,a) \right)^2$$

$$+ \frac{2L\eta_a^2}{(1-\gamma)^2} \sum_{(s,a)\in\mathcal{S}\times\mathcal{A}} d_\rho^{\theta_k}(s) \pi_{\theta_k}(a|s) \mathbb{E}\left[ (\hat{a}_k(s,a) - \tilde{a}_{\theta_k}^\lambda(s,a))^2 | \mathcal{F}_k \right] \ .$$

Next, using that $d_\rho^{\theta_k}(s)\pi_{\theta_k}(a|s) \leq 1$, combined with the fact that for any $p \in \mathcal{P}(\mathcal{A})$, and $x_1, \dots, x_{|\mathcal{A}|} \in \mathbb{R}$, we have $\sum_{i=1}^{|\mathcal{A}|} p_i(x_i - \bar{x})^2 \leq \sum_{i=1}^{|\mathcal{A}|} p_i x_i^2$ where $\bar{x}$ is the $p$-average of the $(x_i)_{i=1}^{|\mathcal{A}|}$ gives

$$\mathbb{E}\left[ \tilde{J}_\lambda^\star - \tilde{J}_\lambda(\theta_{k+1}) \big| \mathcal{F}_k \right] \leq \tilde{J}_\lambda^\star - \tilde{J}_\lambda(\theta_k) - \left( \frac{\eta_a}{2} - \frac{2L\eta_a^2}{(1-\gamma)\rho_{\min}\tau_\lambda} \right) \left\| \nabla \tilde{J}_\lambda(\theta_k) \right\|_2^2$$

$$+ \frac{2\eta_a}{(1-\gamma)^2} \sum_{(s,a)\in\mathcal{S}\times\mathcal{A}} d_\rho^{\theta_k}(s) \pi_{\theta_k}(a|s) \left( \mathbb{E}[\hat{q}_k(s,a)|\mathcal{F}_k] - \tilde{q}_{\theta_k}^\lambda(s,a) \right)^2$$

$$+ \frac{2L\eta_a^2}{(1-\gamma)^2} \sum_{(s,a)\in\mathcal{S}\times\mathcal{A}} d_\rho^{\theta_k}(s) \pi_{\theta_k}(a|s) \mathbb{E}\left[ (\hat{q}_k(s,a) - \tilde{q}_{\theta_k}^\lambda(s,a))^2 | \mathcal{F}_k \right]$$

$$\leq \tilde{J}_\lambda^\star - \tilde{J}_\lambda(\theta_k) - \left( \frac{\eta_a}{2} - \frac{2L\eta_a^2}{(1-\gamma)\rho_{\min}\tau_\lambda} \right) \left\| \nabla \tilde{J}_\lambda(\theta_k) \right\|_2^2$$

$$+ \frac{2\eta_{\mathsf{a}}}{(1-\gamma)^2} \left\| \mathbb{E}[\hat{\mathsf{q}}_k | \mathcal{F}_k] - \tilde{\mathsf{q}}_{\theta_k}^{\lambda} \right\|_2^2 + \frac{2L\eta_{\mathsf{a}}^2}{(1-\gamma)^2} \mathbb{E}\left[ \|\hat{\mathsf{q}}_k - \tilde{\mathsf{q}}_{\theta_k}^{\lambda}\|_2^2 | \mathcal{F}_k \right] \quad .$$

Taking the expectation with respect to all the stochasticity, and using Lemma 18, combined with Lemma 19 gives

$$\mathbb{E}\left[ \tilde{J}_{\lambda}^{\star} - \tilde{J}_{\lambda}(\theta_{k+1}) \right] \leq \mathbb{E}\left[ \tilde{J}_{\lambda}^{\star} - \tilde{J}_{\lambda}(\theta_k) \right] - \left( \frac{\eta_{\mathsf{a}}}{2} - \frac{2L\eta_{\mathsf{a}}^2}{(1-\gamma)\rho_{\min}\tau_{\lambda}} \right) \mathbb{E}\left[ \left\| \nabla \tilde{J}_{\lambda}(\theta_k) \right\|_2^2 \right]$$

$$+ \frac{2\eta_{\mathsf{a}}}{(1-\gamma)^2} (1 - \eta_{\mathsf{c}}\tilde{\mu}_{\mathsf{c}})^H \mathbb{E}\left[ \|\hat{\mathsf{q}}_{k-1} - \tilde{\mathsf{q}}_{\theta_k}^{\lambda}\|_2^2 \right]$$

$$+ \frac{2L\eta_{\mathsf{a}}^2}{(1-\gamma)^2} \left[ (1 - \eta_{\mathsf{c}}\tilde{\mu}_{\mathsf{c}})^{H(k+1)/2} \|\hat{\mathsf{q}}_{-1} - \tilde{\mathsf{q}}_{\theta_0}^{\lambda}\|_2^2 + \frac{\tilde{C}_{\lambda}^2 \eta_{\mathsf{a}}^2 (16 + 8\lambda^2 + 24\lambda^2 \log(|\mathcal{A}|)^2)}{(1-\gamma)^2} \cdot \frac{1}{1 - (1 - \eta_{\mathsf{c}}\tilde{\mu}_{\mathsf{c}})^{H/2}} + \frac{2\eta_{\mathsf{c}}\sigma_{\mathsf{c}}^2}{\tilde{\mu}_{\mathsf{c}}} \right] \quad .$$

Next, applying Jensen's inequality, combined with Lemma 14, yields

$$\mathbb{E}\left[ \tilde{J}_{\lambda}^{\star} - \tilde{J}_{\lambda}(\theta_{k+1}) \right] \leq \mathbb{E}\left[ \tilde{J}_{\lambda}^{\star} - \tilde{J}_{\lambda}(\theta_k) \right] - \left( \frac{\eta_{\mathsf{a}}}{2} - \frac{2L\eta_{\mathsf{a}}^2}{(1-\gamma)\rho_{\min}\tau_{\lambda}} \right) \mathbb{E}\left[ \left\| \nabla \tilde{J}_{\lambda}(\theta_k) \right\|_2^2 \right] + \frac{2\eta_{\mathsf{a}}}{(1-\gamma)^2} (1 - \eta_{\mathsf{c}}\tilde{\mu}_{\mathsf{c}})^H B$$

$$+ \frac{2L\eta_{\mathsf{a}}^2}{(1-\gamma)^2} \left[ (1 - \eta_{\mathsf{c}}\tilde{\mu}_{\mathsf{c}})^{H(k+1)/2} \|\hat{\mathsf{q}}_{-1} - \tilde{\mathsf{q}}_{\theta_0}^{\lambda}\|_2^2 + \frac{\tilde{C}_{\lambda}^2 \eta_{\mathsf{a}}^2 (16 + 8\lambda^2 + 24\lambda^2 \log(|\mathcal{A}|)^2)}{(1-\gamma)^2} \cdot \frac{1}{1 - (1 - \eta_{\mathsf{c}}\tilde{\mu}_{\mathsf{c}})^{H/2}} + \frac{2\eta_{\mathsf{c}}\sigma_{\mathsf{c}}^2}{\tilde{\mu}_{\mathsf{c}}} \right] \quad .$$

Using the bound $1/(1 - (1 - \eta_{\mathsf{c}}\tilde{\mu}_{\mathsf{c}})^{H/2}) \leq 2$, established in Corollary 5, we obtain

$$\mathbb{E}\left[ \tilde{J}_{\lambda}^{\star} - \tilde{J}_{\lambda}(\theta_{k+1}) \right] \leq \mathbb{E}\left[ \tilde{J}_{\lambda}^{\star} - \tilde{J}_{\lambda}(\theta_k) \right] - \left( \frac{\eta_{\mathsf{a}}}{2} - \frac{2L\eta_{\mathsf{a}}^2}{(1-\gamma)\rho_{\min}\tau_{\lambda}} \right) \mathbb{E}\left[ \left\| \nabla \tilde{J}_{\lambda}(\theta_k) \right\|_2^2 \right] + \frac{2\eta_{\mathsf{a}}}{(1-\gamma)^2} (1 - \eta_{\mathsf{c}}\tilde{\mu}_{\mathsf{c}})^H B$$

$$+ \frac{2L\eta_{\mathsf{a}}^2}{(1-\gamma)^2} \left[ (1 - \eta_{\mathsf{c}}\tilde{\mu}_{\mathsf{c}})^{Hk/2} \|\hat{\mathsf{q}}_{-1} - \tilde{\mathsf{q}}_{\theta_0}^{\lambda}\|_2^2 + \frac{\tilde{C}_{\lambda}^2 \eta_{\mathsf{a}}^2 (32 + 16\lambda^2 + 48\lambda^2 \log(|\mathcal{A}|)^2)}{(1-\gamma)^2} + \frac{2\eta_{\mathsf{c}}\sigma_{\mathsf{c}}^2}{\tilde{\mu}_{\mathsf{c}}} \right] \quad .$$

Next, using Lemma 3 combined with (16), and the fact that $\eta_{\mathsf{a}} \leq (1-\gamma)\rho_{\min}\tau_{\lambda}/(8L)$ gives

$$\mathbb{E}\left[ \tilde{J}_{\lambda}^{\star} - \tilde{J}_{\lambda}(\theta_{k+1}) \right] \leq (1 - \frac{\eta_{\mathsf{a}}\underline{\tilde{\mu}}_{\lambda}}{8})\mathbb{E}\left[ \tilde{J}_{\lambda}^{\star} - \tilde{J}_{\lambda}(\theta_k) \right] + \frac{2\eta_{\mathsf{a}}}{(1-\gamma)^2} (1 - \eta_{\mathsf{c}}\tilde{\mu}_{\mathsf{c}})^H B$$

$$+ \frac{2L\eta_{\mathsf{a}}^2}{(1-\gamma)^2} \left[ (1 - \eta_{\mathsf{c}}\tilde{\mu}_{\mathsf{c}})^{H(k+1)/2} \|\hat{\mathsf{q}}_{-1} - \tilde{\mathsf{q}}_{\theta_0}^{\lambda}\|_2^2 + \frac{\tilde{C}_{\lambda}^2 \eta_{\mathsf{a}}^2 (32 + 16\lambda^2 + 48\lambda^2 \log(|\mathcal{A}|)^2)}{(1-\gamma)^2} + \frac{2\eta_{\mathsf{c}}\sigma_{\mathsf{c}}^2}{\tilde{\mu}_{\mathsf{c}}} \right] \quad .$$

Finally, unrolling the recursion concludes the proof $\qquad \square$

Finally, we derive the sample complexity of `Ent-AC` to find an $\epsilon$-precision of the entropy-regularized problem.

**Corollary 6.** *Assume* $A_{\rho}$. *Let* $\epsilon > 0$, *and set*

$$\eta_{\mathsf{c}} = \frac{(1-\gamma)^2 \rho_{\min}\tau_{\lambda}}{40}$$

*and*

$$\eta_{\mathsf{a}} = \min\left( \frac{(1-\gamma)\rho_{\min}\tau_{\lambda}}{8L}, \left[ \frac{\underline{\tilde{\mu}}_{\lambda}(1-\gamma)^4 \epsilon}{1280 \tilde{C}_{\lambda}^2 L (1 + \lambda^2 \log(|\mathcal{A}|)^2)} \right]^{1/3}, \frac{\tilde{\mu}_{\mathsf{c}}\underline{\tilde{\mu}}_{\lambda}\epsilon}{4L\sigma_{\mathsf{c}}^2 \rho_{\min}\tau_{\lambda}} \right) \quad .$$

*If, in addition,*

$$H \geq \frac{40}{(1-\gamma)^2 \rho_{\min}\tau_{\lambda} \tilde{\mu}_{\mathsf{c}}} \max\left\{ 2\log\left( 2 + 4\tilde{C}_{\lambda}^2 \left( \frac{(1-\gamma)\rho_{\min}\tau_{\lambda}}{8L} \right)^2 \right), \log\left( \frac{80B}{\underline{\tilde{\mu}}_{\lambda}(1-\gamma)^2 \epsilon} \right) \right\},$$

*and*

$$K \geq \frac{16}{\underline{\tilde{\mu}}_{\lambda}} \max\left( \frac{8L}{(1-\gamma)\rho_{\min}\tau_{\lambda}}, \left[ \frac{1280 \tilde{C}_{\lambda}^2 L (1 + \lambda^2 \log(|\mathcal{A}|)^2)}{\underline{\tilde{\mu}}_{\lambda}(1-\gamma)^4 \epsilon} \right]^{1/3}, \frac{4L\sigma_{\mathsf{c}}^2 \rho_{\min}\tau_{\lambda}}{\tilde{\mu}_{\mathsf{c}}\underline{\tilde{\mu}}_{\lambda}\epsilon} \right)$$

$$\times \max\left\{\log\left(\frac{5(\tilde{J}_\lambda^\star - \tilde{J}_\lambda(\theta_0))}{\epsilon}\right), \log\left(\frac{20\rho_{\min}\tau_\lambda \left\|\hat{\mathsf{q}}_{-1} - \tilde{\mathsf{q}}_{\theta_0}^\lambda\right\|_2^2}{e(1-\gamma)\tilde{\underline{\mu}}_\lambda \epsilon}\right)\right\},$$

*then* `Ent-AC` *satisfies*

$$\mathbb{E}\left[\tilde{J}_\lambda^\star - \tilde{J}_\lambda(\theta_K)\right] \leq \epsilon .$$

*Proof.* Let

$$a \triangleq 1 - \frac{\eta_{\mathsf{a}}\tilde{\underline{\mu}}_\lambda}{8}, \qquad b \triangleq (1 - \eta_{\mathsf{c}}\tilde{\mu}_{\mathsf{c}})^{H/2}, \qquad \Delta_0 \triangleq \tilde{J}_\lambda^\star - \tilde{J}_\lambda(\theta_0), \qquad E_0 \triangleq \left\|\hat{\mathsf{q}}_{-1} - \tilde{\mathsf{q}}_{\theta_0}^\lambda\right\|_2^2 .$$

By Theorem 4, combined with Corollary 5, it holds that

$$\mathbb{E}\left[\tilde{J}_\lambda^\star - \tilde{J}_\lambda(\theta_K)\right] \leq a^K \Delta_0 + \frac{2L\eta_{\mathsf{a}}^2}{(1-\gamma)^2} K \max(a,b)^K E_0$$

$$+ \frac{16(1 - \eta_{\mathsf{c}}\tilde{\mu}_{\mathsf{c}})^H}{\tilde{\underline{\mu}}_\lambda(1-\gamma)^2} B + \eta_{\mathsf{a}}^3 \frac{256\tilde{C}_\lambda^2 L\left(1 + \lambda^2 \log(|\mathcal{A}|)^2\right)}{\tilde{\underline{\mu}}_\lambda(1-\gamma)^4} + \frac{32L\eta_{\mathsf{a}}\eta_{\mathsf{c}}\sigma_{\mathsf{c}}^2}{(1-\gamma)^2 \tilde{\mu}_{\mathsf{c}}\tilde{\underline{\mu}}_\lambda} .$$

We now show that each of the five terms on the right-hand side is at most $\epsilon/5$.

**Step 1: control of the max-term.** We first prove that $b \leq a$. Recall from the previous lemmas that

$$\tilde{\mu}_{\mathsf{c}} = \frac{(1-\gamma)^2 \rho_{\min}\tau_\lambda}{2}, \qquad \tilde{\underline{\mu}}_\lambda = \frac{\lambda(1-\gamma)\rho_{\min}^2\tau_\lambda^2}{|\mathcal{S}|} .$$

Since

$$\eta_{\mathsf{a}} \leq \frac{(1-\gamma)\rho_{\min}\tau_\lambda}{8L},$$

we obtain

$$\frac{\eta_{\mathsf{a}}\tilde{\underline{\mu}}_\lambda}{8} \leq \frac{(1-\gamma)\rho_{\min}\tau_\lambda\tilde{\underline{\mu}}_\lambda}{64L}$$

$$= \frac{\lambda(1-\gamma)^2\rho_{\min}^3\tau_\lambda^3}{64L|\mathcal{S}|} .$$

Hence, by monotonicity of the map $x \mapsto \log(\frac{1}{1-x})$ on $(0,1)$,

$$\log\left(\frac{1}{1 - \eta_{\mathsf{a}}\tilde{\underline{\mu}}_\lambda/8}\right) \leq \log\left(\frac{1}{1 - \frac{(1-\gamma)\rho_{\min}\tau_\lambda\tilde{\underline{\mu}}_\lambda}{64L}}\right) .$$

Now, if

$$H \geq \frac{2}{\eta_{\mathsf{c}}\tilde{\mu}_{\mathsf{c}}} \log\left(\frac{1}{1 - \frac{(1-\gamma)\rho_{\min}\tau_\lambda\tilde{\underline{\mu}}_\lambda}{64L}}\right),$$

then

$$H \geq \frac{2}{\eta_{\mathsf{c}}\tilde{\mu}_{\mathsf{c}}} \log\left(\frac{1}{1 - \eta_{\mathsf{a}}\tilde{\underline{\mu}}_\lambda/8}\right) .$$

Since

$$\eta_{\mathsf{c}} = \frac{(1-\gamma)^2\rho_{\min}\tau_\lambda}{40}, \qquad \tilde{\mu}_{\mathsf{c}} = \frac{(1-\gamma)^2\rho_{\min}\tau_\lambda}{2},$$

we have

$$\frac{2}{\eta_{\mathsf{c}}\tilde{\mu}_{\mathsf{c}}} = \frac{80}{(1-\gamma)^4 \rho_{\min}^2 \tau_\lambda^2} = \frac{40}{(1-\gamma)^2 \rho_{\min} \tau_\lambda \, \tilde{\mu}_{\mathsf{c}}} \ .$$

Therefore, the lower bound

$$H \geq \frac{40}{(1-\gamma)^2 \rho_{\min} \tau_\lambda \, \tilde{\mu}_{\mathsf{c}}} \, 2\log\!\left(\frac{1}{1 - \frac{(1-\gamma)\rho_{\min}\tau_\lambda \underline{\tilde{\mu}}_\lambda}{64L}}\right)$$

implies

$$H \geq \frac{2}{\eta_{\mathsf{c}}\tilde{\mu}_{\mathsf{c}}} \log\!\left(\frac{1}{1 - \eta_{\mathsf{a}}\underline{\tilde{\mu}}_\lambda/8}\right) \ .$$

Consequently,

$$\frac{\eta_{\mathsf{c}}\tilde{\mu}_{\mathsf{c}} H}{2} \geq \log\!\left(\frac{1}{1 - \eta_{\mathsf{a}}\underline{\tilde{\mu}}_\lambda/8}\right) = -\log(a) \ .$$

Multiplying by $-1$ and exponentiating both sides yield

$$\exp\!\left(-\frac{\eta_{\mathsf{c}}\tilde{\mu}_{\mathsf{c}} H}{2}\right) \leq a \ .$$

Using $1 - x \leq e^{-x}$ for all $x \geq 0$, we obtain

$$b = (1 - \eta_{\mathsf{c}}\tilde{\mu}_{\mathsf{c}})^{H/2} \leq \exp\!\left(-\frac{\eta_{\mathsf{c}}\tilde{\mu}_{\mathsf{c}} H}{2}\right) \leq a \ .$$

Hence

$$\max(a, b) = a \ .$$

**Step 2: bound on the first term.** To ensure that $a^K \Delta_0 \leq \epsilon/5$, it is enough that

$$K \geq \frac{1}{\log(1/a)} \log\!\left(\frac{5\Delta_0}{\epsilon}\right) \ .$$

Since $-\log(1 - x) \geq x$ for all $x \in (0, 1)$, we have

$$\log(1/a) = -\log\!\left(1 - \frac{\eta_{\mathsf{a}}\underline{\tilde{\mu}}_\lambda}{8}\right) \geq \frac{\eta_{\mathsf{a}}\underline{\tilde{\mu}}_\lambda}{8} \ .$$

Thus it is sufficient that

$$K \geq \frac{8}{\eta_{\mathsf{a}}\underline{\tilde{\mu}}_\lambda} \log\!\left(\frac{5\Delta_0}{\epsilon}\right) \ .$$

**Step 3: bound on the second term.** We use the inequality

$$t x^t \leq \frac{2}{e \log(1/x)} x^{t/2}, \qquad x \in (0, 1), \ t \geq 0,$$

which follows by setting $u = t\log(1/x)$ and using $u e^{-u/2} \leq 2/e$ for all $u \geq 0$. Applying this with $x = a$ and $t = K$, we obtain

$$K a^K \leq \frac{2}{e \log(1/a)} a^{K/2} \leq \frac{16}{e\eta_{\mathsf{a}}\underline{\tilde{\mu}}_\lambda} a^{K/2},$$

where we used again $\log(1/a) \geq \eta_a \tilde{\underline{\mu}}_\lambda/8$. Hence

$$\frac{2L\eta_a^2}{(1-\gamma)^2} K a^K E_0 \leq \frac{32L\eta_a}{e(1-\gamma)^2\tilde{\underline{\mu}}_\lambda} a^{K/2} E_0 \ .$$

Therefore, it is sufficient to require

$$K \geq \frac{16}{\eta_a\tilde{\underline{\mu}}_\lambda} \log\left(\frac{160L\eta_a E_0}{e(1-\gamma)^2\tilde{\underline{\mu}}_\lambda\epsilon}\right)$$

to make the second term at most $\epsilon/5$. Now, since

$$\eta_a \leq \frac{(1-\gamma)\rho_{\min}\tau_\lambda}{8L},$$

we have

$$\frac{160L\eta_a E_0}{e(1-\gamma)^2\tilde{\underline{\mu}}_\lambda\epsilon} \leq \frac{20\rho_{\min}\tau_\lambda E_0}{e(1-\gamma)\tilde{\underline{\mu}}_\lambda\epsilon} \ .$$

Hence a sufficient condition is

$$K \geq \frac{16}{\eta_a\tilde{\underline{\mu}}_\lambda} \log\left(\frac{20\rho_{\min}\tau_\lambda E_0}{e(1-\gamma)\tilde{\underline{\mu}}_\lambda\epsilon}\right) \ .$$

**Step 4: bound on the critic-bias term.** To ensure that

$$\frac{16(1-\eta_c\tilde{\mu}_c)^H}{\tilde{\underline{\mu}}_\lambda(1-\gamma)^2} B \leq \frac{\epsilon}{5},$$

it is enough that

$$(1-\eta_c\tilde{\mu}_c)^H \leq \frac{\tilde{\underline{\mu}}_\lambda(1-\gamma)^2\epsilon}{80B} \ .$$

Using $(1-x)^m \leq e^{-xm}$, a sufficient condition is

$$H \geq \frac{1}{\eta_c\tilde{\mu}_c} \log\left(\frac{80B}{\tilde{\underline{\mu}}_\lambda(1-\gamma)^2\epsilon}\right) = \frac{40}{(1-\gamma)^2\rho_{\min}\tau_\lambda\tilde{\mu}_c} \log\left(\frac{80B}{\tilde{\underline{\mu}}_\lambda(1-\gamma)^2\epsilon}\right) \ .$$

**Step 5: bound on the policy-switch term.** To ensure that

$$\eta_a^3 \frac{256\tilde{C}_\lambda^2 L\left(1+\lambda^2\log(|\mathcal{A}|)^2\right)}{\tilde{\underline{\mu}}_\lambda(1-\gamma)^4} \leq \frac{\epsilon}{5},$$

it is sufficient that

$$\eta_a \leq \left[\frac{\tilde{\underline{\mu}}_\lambda(1-\gamma)^4\epsilon}{1280\tilde{C}_\lambda^2 L\left(1+\lambda^2\log(|\mathcal{A}|)^2\right)}\right]^{1/3} \ .$$

**Step 6: bound on the variance term.** To ensure that

$$\frac{32L\eta_a\eta_c\sigma_c^2}{(1-\gamma)^2\tilde{\mu}_c\tilde{\underline{\mu}}_\lambda} \leq \frac{\epsilon}{5},$$

it is sufficient that

$$\eta_a \leq \frac{(1-\gamma)^2\tilde{\mu}_c\tilde{\underline{\mu}}_\lambda\epsilon}{160L\sigma_c^2\eta_c} = \frac{\tilde{\mu}_c\tilde{\underline{\mu}}_\lambda\epsilon}{4L\sigma_c^2\rho_{\min}\tau_\lambda} \ .$$

**Conclusion.** Since

$$\eta_{\mathsf{a}} = \min\left( \frac{(1-\gamma)\rho_{\min}\tau_\lambda}{8L}, \left[ \frac{\tilde{\mu}_\lambda(1-\gamma)^4\epsilon}{1280\tilde{C}_\lambda^2 L\left(1+\lambda^2\log(|\mathcal{A}|)^2\right)} \right]^{1/3}, \frac{\tilde{\mu}_{\mathsf{c}}\tilde{\underline{\mu}}_\lambda\epsilon}{4L\sigma_{\mathsf{c}}^2\rho_{\min}\tau_\lambda} \right),$$

we have

$$\frac{1}{\eta_{\mathsf{a}}} = \max\left( \frac{8L}{(1-\gamma)\rho_{\min}\tau_\lambda}, \left[ \frac{1280\tilde{C}_\lambda^2 L\left(1+\lambda^2\log(|\mathcal{A}|)^2\right)}{\tilde{\mu}_\lambda(1-\gamma)^4\epsilon} \right]^{1/3}, \frac{4L\sigma_{\mathsf{c}}^2\rho_{\min}\tau_\lambda}{\tilde{\mu}_{\mathsf{c}}\tilde{\underline{\mu}}_\lambda\epsilon} \right) .$$

Therefore, the condition

$$K \geq \frac{16}{\tilde{\underline{\mu}}_\lambda}\max\left( \frac{8L}{(1-\gamma)\rho_{\min}\tau_\lambda}, \left[ \frac{1280\tilde{C}_\lambda^2 L\left(1+\lambda^2\log(|\mathcal{A}|)^2\right)}{\tilde{\mu}_\lambda(1-\gamma)^4\epsilon} \right]^{1/3}, \frac{4L\sigma_{\mathsf{c}}^2\rho_{\min}\tau_\lambda}{\tilde{\mu}_{\mathsf{c}}\tilde{\underline{\mu}}_\lambda\epsilon} \right)\max\left\{ \log\left( \frac{5\Delta_0}{\epsilon} \right), \log\left( \frac{20\rho_{\min}\tau_\lambda E_0}{e(1-\gamma)\tilde{\underline{\mu}}_\lambda\epsilon} \right) \right\}$$

implies both bounds obtained in Steps 2 and 3.

Similarly, since

$$\eta_{\mathsf{a}} \leq \frac{(1-\gamma)\rho_{\min}\tau_\lambda}{8L},$$

we have

$$2\log\left( 2 + 4\tilde{C}_\lambda^2\eta_{\mathsf{a}}^2 \right) \leq 2\log\left( 2 + 4\tilde{C}_\lambda^2\left( \frac{(1-\gamma)\rho_{\min}\tau_\lambda}{8L} \right)^2 \right),$$

$$2\log\left( \frac{1}{1 - \eta_{\mathsf{a}}\tilde{\underline{\mu}}_\lambda/8} \right) \leq 2\log\left( \frac{1}{1 - \frac{(1-\gamma)\rho_{\min}\tau_\lambda\tilde{\underline{\mu}}_\lambda}{64L}} \right).$$

Therefore, the lower bound

$$H \geq \frac{40}{(1-\gamma)^2\rho_{\min}\tau_\lambda\,\tilde{\mu}_{\mathsf{c}}}\max\left\{ 2\log\left( 2 + 4\tilde{C}_\lambda^2\left( \frac{(1-\gamma)\rho_{\min}\tau_\lambda}{8L} \right)^2 \right), 2\log\left( \frac{1}{1 - \frac{(1-\gamma)\rho_{\min}\tau_\lambda\tilde{\underline{\mu}}_\lambda}{64L}} \right), \log\left( \frac{80B}{\tilde{\mu}_\lambda(1-\gamma)^2\epsilon} \right) \right\}$$

simultaneously guarantees the condition of Corollary 5, the domination $b \leq a$, and the bound on the critic-bias term. Observing that the first term of the maximum dominates the second allows to simplify the bound and to obtain the desired condition on $H$. Under all these conditions, each of the five terms above is at most $\epsilon/5$, and therefore

$$\mathbb{E}\left[ \tilde{J}_\lambda^\star - \tilde{J}_\lambda(\theta_K) \right] \leq \epsilon .$$

This concludes the proof. □

## F. Montone Improvement operator

The goal of this section is therefore to show the existence of an operator $\mathcal{U}_\tau\colon \pi \to \mathcal{U}_\tau(\pi)$ with three crucial properties: (i) for any policy, applying this operator produces a new policy with a higher regularized value than the one achieved by $\pi$; (ii) every policy generated by this operator assigns at least a fixed minimum probability to every action; (iii) th. The main idea is to build the improvement operator such that it slightly augments the smallest probability weights, such that for any state action pair $(s, a) \in \mathcal{S} \times \mathcal{A}$ the probability $\pi(a|s)$ stays above a certain threshold. We will show below that this procedure improves the global objective while keeping the probabilities uniformly bounded away from $0$ when the threshold is properly chosen. For any policy $\pi$, state $s \in \mathcal{S}$, $\tau < 1/(2|\mathcal{A}|^2)$, we respectively define $\mathcal{A}_\tau^\pi(s)$, and $a_{\max}^\pi(s)$ as

$$\mathcal{A}_\tau^\pi(s) \stackrel{\Delta}{=} \{a \in \mathcal{A}, \pi(a|s) \leq \tau\} , \quad a_{\max}^\pi(s) = \arg\max_{a \in \mathcal{A}}\{\pi(a|s)\} ,$$

where the $\arg\max$ is chosen at random in the case of ties. Note that the definition of $\tau_\lambda$ ensures that $a_{\max}^\pi(s)$ does not belong to the set $\mathcal{A}_\tau^\pi(s)$ as

$$\max_{a \in \mathcal{A}} \pi(a|s) \geq 1/|\mathcal{A}| \ .$$

Finally, we define the improvement operator as follows:

$$\begin{aligned} \mathcal{U}_\tau : \mathcal{P}(\mathcal{A})^{\mathcal{S}} &\longrightarrow \mathcal{P}(\mathcal{A})^{\mathcal{S}}, \\ \pi &\longmapsto \mathcal{U}_\tau(\pi), \end{aligned} \tag{33}$$

where for every $(s, a) \in \mathcal{S} \times \mathcal{A}$,

$$\mathcal{U}_\tau(\pi)(a|s) = \begin{cases} \tau, & \text{if } \pi(a|s) \leq \tau, \\ \pi(a|s) - \displaystyle\sum_{b \in \mathcal{A}_\tau^\pi(s)} \left(\tau - \pi(b|s)\right), & \text{if } a = a_{\max}^\pi(s), \\ \pi(a|s), & \text{otherwise.} \end{cases}$$

The operator $\mathcal{U}_\tau$ builds $\mathcal{U}_\tau(\pi)(a|s)$ by (statewise) raising each $a \in \mathcal{A}_\tau^\pi(s)$ to $\tau$, substracting the total added mass from the single action $a_{\max}^\pi(s)$, and leaving other actions unchanged. If $\mathcal{A}_\tau^\pi(s) = \emptyset$, for all $s \in \mathcal{S}$, then $\mathcal{U}_\tau(\pi) = \pi$. Note that mass conservation is immediate from the definition and the fact that $\tau < 1/(2|\mathcal{A}|^2)$. Non-negativity of $\mathcal{U}_\tau(\pi)(a_{\max}^\pi(s)|s)$ follows because the removed mass is

$$\sum_{a \in \mathcal{A}_\tau^\pi(s)} \{\tau - \pi(a|s)\} \leq \tau \times |\mathcal{A}| \leq \frac{1}{2|\mathcal{A}|}$$

Since $\pi(a_{\max}^\pi(s)|s) \geq 1/|\mathcal{A}|$, we get that $\mathcal{U}(\pi)(a_{\max}^\pi(s)|s) \geq 1/(2|\mathcal{A}|)$. This in particular shows that $\mathcal{U}_\tau(\pi)$ is a policy. Next, define

$$\tau_\lambda \overset{\Delta}{=} \min\left( \frac{1}{3} \exp\left( -\frac{16 + 8\gamma\lambda\log(|\mathcal{A}|)}{\lambda(1-\gamma)^2 \rho_{\min}} \right), \frac{1}{3^8 |\mathcal{A}|^4} \right) \ . \tag{34}$$

The following lemma establishes the crucial improvement property when $\tau = \tau_\lambda$.

**Lemma 20.** *Assume that the initial distribution $\rho$ satisfies $A_\rho$. For any policy $\pi$, it holds that*

$$\tilde{v}_{\mathcal{U}_{\tau_\lambda}(\pi)}^\lambda(\rho) \geq \tilde{v}_\pi^\lambda(\rho) \ .$$

*Additionally, for any policy $\pi$, we have that*

$$\mathcal{U}_{\tau_\lambda}(\pi)(a|s) \geq \tau_\lambda \ .$$

*Proof.* Set an arbitrary policy $\pi$. For avoiding heavy notations, we will, through this proof, denote by $\mathcal{A}_\tau^\pi = \mathcal{A}_{\tau_\lambda}^\pi$. We consider the case where there is $s \in \mathcal{S}$ such that $\mathcal{A}_\tau^\pi(s) \neq \emptyset$ (alternatively $\mathcal{U}_{\tau_\lambda}(\pi) = \pi$, which makes the previous inequality immediately valid). Define $\tilde{\pi} = \mathcal{U}_{\tau_\lambda}(\pi)$. The following applies

$$\begin{aligned} \tilde{v}_{\tilde{\pi}}^\lambda(\rho) - \tilde{v}_\pi^\lambda(\rho) &= \sum_{s \in \mathcal{S}} d_\rho^{\tilde{\pi}}(s) \sum_{a \in \mathcal{A}} \left[\tilde{\pi}(a|s)\mathsf{r}(s,a) - \lambda\tilde{\pi}(a|s)\log\left(\tilde{\pi}(a|s)\right)\right] \\ &\quad - \sum_{s \in \mathcal{S}} d_\rho^\pi(s) \sum_{a \in \mathcal{A}} \left[\pi(a|s)\mathsf{r}(s,a) - \lambda\pi(a|s)\log\left(\pi(a|s)\right)\right] \\ &= \underbrace{\sum_{s \in \mathcal{S}} \left(d_\rho^{\tilde{\pi}}(s) - d_\rho^\pi(s)\right) \sum_{a \in \mathcal{A}} \left[\tilde{\pi}(a|s)\mathsf{r}(s,a) - \lambda\tilde{\pi}(a|s)\log\left(\tilde{\pi}(a|s)\right)\right]}_{\textbf{(I)}} \\ &\quad + \underbrace{\sum_{s \in \mathcal{S}} d_\rho^\pi(s) \sum_{a \in \mathcal{A}} (\tilde{\pi}(a|s) - \pi(a|s))\mathsf{r}(s,a)}_{\textbf{(II)}} \end{aligned}$$

$$+ \lambda \underbrace{\sum_{s \in \mathcal{S}} d_\rho^\pi(s) \sum_{a \in \mathcal{A}} \left[ \pi(a|s) \log \left( \pi(a|s) \right) - \tilde{\pi}(a|s) \log \left( \tilde{\pi}(a|s) \right) \right]}_{\text{(III)}} \quad .$$

We now lower-bound each of the three terms separately.

**Bounding (I).** Using Lemma 25, we have

$$\textbf{(I)} \geq - \| d_\rho^{\tilde{\pi}} - d_\rho^\pi \|_1 \max_{s \in \mathcal{S}} \left| \sum_{a \in \mathcal{A}} \left[ \tilde{\pi}(a|s) \mathsf{r}(s,a) - \lambda \tilde{\pi}(a|s) \log \left( \tilde{\pi}(a|s) \right) \right] \right|$$

$$\geq -\frac{\gamma}{1-\gamma} \sup_{s \in \mathcal{S}} \| \tilde{\pi}(\cdot|s) - \pi(\cdot|s) \|_1 \left( 1 + \lambda \log(|\mathcal{A}|) \right) \quad .$$

**Bounding (II).** Using the triangle inequality yields

$$\textbf{(II)} \geq - \sup_{s \in \mathcal{S}} \| \tilde{\pi}(\cdot|s) - \pi(\cdot|s) \|_1 \quad .$$

**Bounding (III).** All the state-action pairs on which the original $\pi$ allocates the same probability then the policy $\tilde{\pi}$ are equal to 0 in **(III)** allowing us to simplify this term

$$\textbf{(III)} = \lambda \sum_{s \in \mathcal{S}} d_\rho^\pi(s) \sum_{a \in \mathcal{A}} \left[ \pi(a|s) \log \left( \pi(a|s) \right) - \tilde{\pi}(a|s) \log \left( \tilde{\pi}(a|s) \right) \right]$$

$$= \lambda \sum_{s \in \mathcal{S}} d_\rho^\pi(s) \sum_{a \in \mathcal{A}_\tau^\pi(s)} \left[ \pi(a|s) \log \left( \pi(a|s) \right) - \tilde{\pi}(a|s) \log(\tilde{\pi}(a|s)) \right]$$

$$+ \lambda \sum_{s \in \mathcal{S}} \mathbf{1}(\mathcal{A}_\tau^\pi(s) \neq \emptyset) d_\rho^\pi(s) \left[ \pi(a_{\max}^\pi(s)|s) \log \left( \pi(a_{\max}^\pi(s)|s) \right) - \tilde{\pi}(a_{\max}^\pi(s)|s) \log \left( \tilde{\pi}(a_{\max}^\pi(s)|s) \right) \right] \quad .$$

Since $x \mapsto x \log(x)$ is convex, for all $u, v \in [0; 1]$, $u \log(u) - v \log(v) \geq \left[ \log(v) + 1 \right] (u - v)$, we have

$$\textbf{(III)} \geq \lambda \sum_{s \in \mathcal{S}} d_\rho^\pi(s) \sum_{a \in \mathcal{A}_\tau^\pi(s)} \left( \pi(a|s) - \tilde{\pi}(a|s) \right) \left[ \log(\tau_\lambda) + 1 \right] \qquad (\text{since } \tilde{\pi}(a|s) = \tau_\lambda)$$

$$+ \lambda \sum_{s \in \mathcal{S}} \mathbf{1}(\mathcal{A}_\tau^\pi(s) \neq \emptyset) d_\rho^\pi(s) \left[ \pi(a_{\max}^\pi(s)|s) - \tilde{\pi}(a_{\max}^\pi(s)|s) \right] \left[ \log \left( \tilde{\pi}(a_{\max}^\pi(s)|s) \right) + 1 \right] \quad ,$$

Next, using that

$$\tilde{\pi}(a_{\max}^\pi(s)|s) \geq \pi(a_{\max}^\pi(s)|s) - |\mathcal{A}|\tau_\lambda \geq \frac{1}{|\mathcal{A}|} - \frac{1}{2|\mathcal{A}|} = \frac{1}{2|\mathcal{A}|} \quad ,$$

combined with the monotonicity of $x\colon \log(x) + 1$ and the fact that $\pi(a_{\max}^\pi(s)|s) - \tilde{\pi}(a_{\max}^\pi(s)|s) \geq 0$ yields

$$\textbf{(III)} \geq \lambda \sum_{s \in \mathcal{S}} d_{\rho,\tilde{\pi}}(s) \sum_{a \in \mathcal{A}_\tau^\pi(s)} \left( \pi(a|s) - \tilde{\pi}(a|s) \right) \left[ \log(\tau_\lambda) + 1 \right]$$

$$+ \lambda \sum_{s \in \mathcal{S}} \mathbf{1}(\mathcal{A}_\tau^\pi(s) \neq \emptyset) d_\rho^{\tilde{\pi}}(s) \left[ \pi(a_{\max}^\pi(s)|s) - \tilde{\pi}(a_{\max}^\pi(s)|s) \right] \left[ \log(\frac{1}{2|\mathcal{A}|}) + 1 \right] \quad ,$$

Additionally, since

$$0 \leq \pi(a_{\max}^\pi(s)|s) - \tilde{\pi}(a_{\max}^\pi(s)|s) = \sum_{a \in \mathcal{A}_\tau^\pi(s)} \left( \pi(a|s) - \tilde{\pi}(a|s) \right) = \frac{1}{2} \| \pi(\cdot|s) - \tilde{\pi}(\cdot|s) \|_1 \quad ,$$

implies

$$\textbf{(III)} \geq -\frac{\lambda}{2} \sum_{s \in \mathcal{S}} d_\rho^{\tilde{\pi}}(s) \mathbf{1}(\mathcal{A}_\tau^\pi(s) \neq \emptyset) \| \pi(\cdot|s) - \tilde{\pi}(\cdot|s) \|_1 \left[ \log(\tau_\lambda) + 1 \right]$$

$$-\frac{\lambda}{2}\sum_{s\in\mathcal{S}}d_\rho^{\tilde{\pi}}(s)\mathbf{1}(\mathcal{A}_\tau^\pi(s)\neq\emptyset)\left\|\pi(\cdot|s)-\tilde{\pi}(\cdot|s)\right\|_1\left[\log(2|\mathcal{A}|)+1\right]\ ,$$

$$\geq-\frac{\lambda}{4}\sum_{s\in\mathcal{S}}d_\rho^{\tilde{\pi}}(s)\mathbf{1}(\mathcal{A}_\tau^\pi(s)\neq\emptyset)\left\|\pi(\cdot|s)-\tilde{\pi}(\cdot|s)\right\|_1\left[\log(\tau_\lambda)+1\right]\ ,$$

where in the last inequality, we used that $\tau_\lambda\leq\frac{1}{3^8|\mathcal{A}|^4}\leq\exp(-4\log(2|\mathcal{A}|)-5)$. Hence, by using $\mathbf{A}_\rho$, we can lower bound this term as follows

$$(\mathbf{III})\geq-\frac{\lambda}{4}(1-\gamma)\rho_{\min}\max_{s\in\mathcal{S}}\left\|\pi(\cdot|s)-\tilde{\pi}(\cdot|s)\right\|_1\left[\log(\tau_\lambda)+1\right]\ .$$

Collecting these lower bounds and using that

$$[\log(\tau_\lambda)+1]\leq-\frac{16+8\gamma\lambda\log(|\mathcal{A}|)}{\lambda(1-\gamma)^2\rho_{\min}}$$

concludes the proof. $\qquad\square$

Finally, we define the operator that maps each policy to one corresponding parameter

$$\mathcal{L}:\Pi\ \to\ \mathbb{R}^{|\mathcal{S}||\mathcal{A}|}$$

by

$$\mathcal{L}(\pi)(s,a)\ \triangleq\ \log(\pi(a|s)),\quad\text{for all}\,(s,a)\in\mathcal{S}\times\mathcal{A}\,.\tag{35}$$

Finally, we define the improvement operator on the logitspace as

$$\mathcal{T}_\tau\triangleq\mathcal{L}\circ\mathcal{U}_\tau\ .$$

The following lemma shows that $\mathcal{L}_\tau$ successfully recovers a parameter that gives the policy and that $\mathcal{T}_\tau$ improves the value of the objective when $\tau=\tau_\lambda$.

**Lemma 21.** *Assume that the initial distribution $\rho$ satisfies $\mathbf{A}_\rho$. For any policy $\pi$, it holds that*

$$\pi_{\mathcal{L}(\pi)}=\pi\ ,$$

*Additionally, for any $\theta\in\mathbb{R}^{|\mathcal{S}||\mathcal{A}|}$ and $(s,a)\in\mathcal{S}\times\mathcal{A}$, we have that*

$$\tilde{v}_{\mathcal{T}_{\tau_\lambda}(\theta)}^\lambda(\rho)\geq\tilde{v}_\theta^\lambda(\rho)\ ,\quad\pi_{\mathcal{T}_{\tau_\lambda}(\theta)}\geq\tau_\lambda\ .$$

*Proof.* The proof follows immediately from the definition of the softmax policy, from (35), and Lemma 20. $\qquad\square$

Define the set

$$\Pi_\tau\triangleq\left\{\pi\in\mathcal{P}(\mathcal{A})^\mathcal{S},\text{ such that for all }(s,a)\in\mathcal{S}\times\mathcal{A},\pi(a|s)\geq\tau\right\}\ .$$

The next lemma shows that $\mathcal{U}_\tau$ can be seen as a projection on this set.

**Lemma 22.** *Fix any policy $\pi_1$ and any policy $\pi_2\in\Pi_\tau$. Then*

$$\left\|\pi_1-\mathcal{U}_\tau(\pi_1)\right\|_1\leq\left\|\pi_1-\pi_2\right\|_1.$$

*More precisely, for every $s\in\mathcal{S}$,*

$$\left\|\pi_1(\cdot\mid s)-\mathcal{U}_\tau(\pi_1)(\cdot\mid s)\right\|_1\leq\left\|\pi_1(\cdot\mid s)-\pi_2(\cdot\mid s)\right\|_1.$$

*Proof.* Since the $\ell_1$-norm decomposes over the states, it is enough to prove the statewise inequality. Fix $s \in \mathcal{S}$ and set

$$A_s := \mathcal{A}_\tau^{\pi_1}(s), \qquad D_s := \sum_{a \in A_s} \big(\tau - \pi_1(a \mid s)\big).$$

If $A_s = \emptyset$, then $\mathcal{U}_\tau(\pi_1)(\cdot \mid s) = \pi_1(\cdot \mid s)$, so the claim is immediate. Assume now that $A_s \neq \emptyset$. By definition of $\mathcal{U}_\tau$, each action in $A_s$ is increased to $\tau$, the action $a_{\max}^{\pi_1}(s)$ is decreased by exactly the total transferred mass $D_s$, and all other actions are unchanged. Therefore,

$$\|\pi_1(\cdot \mid s) - \mathcal{U}_\tau(\pi_1)(\cdot \mid s)\|_1 = \sum_{a \in A_s} \big(\tau - \pi_1(a \mid s)\big) + D_s = 2D_s.$$

Now decompose

$$\|\pi_1(\cdot \mid s) - \pi_2(\cdot \mid s)\|_1 = \sum_{a \in A_s} |\pi_1(a \mid s) - \pi_2(a \mid s)| + \sum_{a \notin A_s} |\pi_1(a \mid s) - \pi_2(a \mid s)|.$$

Since $\pi_2 \in \Pi_\tau$, for every $a \in A_s$ we have $\pi_2(a \mid s) \geq \tau \geq \pi_1(a \mid s)$. Hence

$$\sum_{a \in A_s} |\pi_1(a \mid s) - \pi_2(a \mid s)| = \sum_{a \in A_s} \big(\pi_2(a \mid s) - \pi_1(a \mid s)\big) \geq \sum_{a \in A_s} \big(\tau - \pi_1(a \mid s)\big) = D_s.$$

Also, because both $\pi_1(\cdot \mid s)$ and $\pi_2(\cdot \mid s)$ are probability distributions,

$$\sum_{a \in A_s} \big(\pi_2(a \mid s) - \pi_1(a \mid s)\big) = \sum_{a \notin A_s} \big(\pi_1(a \mid s) - \pi_2(a \mid s)\big).$$

Therefore, by the triangle inequality,

$$\sum_{a \notin A_s} |\pi_1(a \mid s) - \pi_2(a \mid s)| \geq \left| \sum_{a \notin A_s} \big(\pi_1(a \mid s) - \pi_2(a \mid s)\big) \right| = \sum_{a \in A_s} \big(\pi_2(a \mid s) - \pi_1(a \mid s)\big) \geq D_s.$$

Combining the two bounds gives

$$\|\pi_1(\cdot \mid s) - \pi_2(\cdot \mid s)\|_1 \geq D_s + D_s = 2D_s = \|\pi_1(\cdot \mid s) - \mathcal{U}_\tau(\pi_1)(\cdot \mid s)\|_1.$$

This proves the statewise inequality. Summing over $s \in \mathcal{S}$ concludes the proof. $\qquad\square$

# G. Technical Lemmas

**Lemma 23** (Lemma 1.2.3 in (Nesterov, 2013))**.** *Let $f : \mathbb{R}^d \to \mathbb{R}$ be twice continuously differentiable. Suppose there exists $L \geq 0$ such that for all $x \in \mathbb{R}^d$ and $v \in \mathbb{R}^d$,*

$$|v^\top \nabla^2 f(x)\, v| \leq L \|v\|^2.$$

*Then $f$ has an $L$-Lipschitz continuous gradient (i.e., $f$ is $L$-smooth); in particular,*

$$\|\nabla f(y) - \nabla f(x)\| \leq L \|y - x\|,$$

*and*

$$f(y) \geq f(x) + \langle \nabla f(x), y - x \rangle - \tfrac{L}{2}\|y - x\|^2$$

*for all $x, y \in \mathbb{R}^d$.*

**Lemma 24** (Flow conservation constraints (Puterman, 1994))**.** *For any $\pi \in \Pi$, and $s \in \mathcal{S}$, it holds that*

$$d_\rho^\pi(s) = (1 - \gamma)\rho(s) + \gamma \sum_{(s',a')} \mathsf{P}(s|s', a')\pi(a'|s')d_\rho^\pi(s') \ .$$

**Lemma 25.** *Consider any two policies $\pi_i$, $i = 1, 2$. It holds that*

$$\|d_\rho^{\pi_1} - d_\rho^{\pi_2}\|_1 \leq \frac{\gamma}{1-\gamma} \sup_{s \in \mathcal{S}} \|\pi_1(\cdot|s) - \pi_2(\cdot|s)\|_1 \ .$$

*Proof.* Let us start from the definition of flow conservation constraints for the discounted state occupancy Lemma 24, for $i \in \{1, 2\}$, we have

$$d_\rho^{\pi_i}(s) = (1 - \gamma)\rho(s) + \gamma \sum_{s'} \mathsf{P}_{\pi_i}(s|s') d_\rho^{\pi_i}(s') \ .$$

Then, we have

$$
\begin{aligned}
\sum_{s \in \mathcal{S}} |d_\rho^{\pi_2}(s) - d_\rho^{\pi_1}(s)| &\leq \gamma \sum_{(s', s')} \sum_s \left| \mathsf{P}(s|s', s')\pi_2(s'|s')d_\rho^{\pi_2}(s') - \mathsf{P}(s|s', s')\pi_1(s'|s')d_\rho^{\pi_1}(s') \right| \\
&\leq \gamma \sum_{s', s'} \sum_s \mathsf{P}(s|s', s') \left| \pi_2(a'|s') - \pi_1(a'|s') \right| d_\rho^{\pi_2}(s') \\
&\quad + \gamma \sum_{s', a'} \sum_s \mathsf{P}(s|s', a')\pi_1(a'|s') \left| d_\rho^{\pi_1}(s') - d_\rho^{\pi_2}(s') \right| \\
&\leq \gamma \sup_{s \in \mathcal{S}} \|\pi_1(\cdot|s) - \pi_2(\cdot|s)\|_1 + \gamma \sum_{s'} |d_\rho^{\pi_1}(s') - d_\rho^{\pi_2}(s')| \ ,
\end{aligned}
$$

which concludes the proof. $\qquad\square$

**Lemma 26** (Soft-Performance Difference Lemma). *Consider any two policies $\pi_i$, $i = 1, 2$. For any $s \in \mathcal{S}$, it holds that*

$$\tilde{\mathsf{v}}_{\pi_2}^\lambda(s) - \tilde{\mathsf{v}}_{\pi_1}^\lambda(s) = \frac{1}{1-\gamma} \sum_{s \in \mathcal{S}} d_s^{\pi_1}(s') \left[ \sum_{a \in \mathcal{A}} (\pi_2(a|s') - \pi_1(a|s')) \left[ \tilde{\mathsf{q}}_{\pi_2}^\lambda(s', a) - \lambda \log(\pi_2(a|s')) + \lambda \mathrm{KL}(\pi_1(\cdot|s')\|\pi_2(\cdot|s')) \right] \right] .$$

**Lemma 27** (Bound on the entropy regularized Q-value: Equation 86 of (Mei et al., 2020b)). *For any policy $\pi \in \Pi$, $\lambda \geq 0$, and $0 \leq \gamma < 1$, it holds that*

$$\|\tilde{\mathsf{q}}_\pi^\lambda\|_\infty \leq \frac{1 + \lambda \log(|\mathcal{A}|)}{1 - \gamma} \ .$$

**Lemma 28** (Pinsker's inequality (discrete version)). *Let $p = (p_i)_{i=1}^n$ and $q = (q_i)_{i=1}^n$ be probability distributions on a finite set $\{1, \ldots, n\}$. Then*

$$\frac{1}{2} \sum_{i=1}^n |p_i - q_i| \ \leq \ \sqrt{\tfrac{1}{2} \mathrm{KL}(p\|q)}.$$

**Lemma 29** (KL upper bound). *Let $p = (p_i)_{i=1}^n$ and $q = (q_i)_{i=1}^n$ be probability distributions on a finite set $\{1, \ldots, n\}$, and assume*

$$q_{\min} := \min_{i \in [n]} q_i \ > \ 0 \ .$$

*Then*

$$\mathrm{KL}(p\|q) \ \leq \ \frac{1}{q_{\min}} \|p - q\|_1 \ .$$

*Proof.* Let $A := \{i \in [n] : p_i \geq q_i\}$. Since $\log(\cdot)$ is increasing, for $i \notin A$ we have $\log(p_i/q_i) \leq 0$, hence

$$\mathrm{KL}(p\|q) = \sum_{i=1}^n p_i \log \frac{p_i}{q_i} \ \leq \ \sum_{i \in A} p_i \log \frac{p_i}{q_i}.$$

For $i \in A$, set $u_i := p_i/q_i \geq 1$. Moreover, $p_i \leq 1$ and $q_i \geq q_{\min}$ imply $u_i \leq 1/q_{\min}$. We claim that for any $m \in (0, 1]$ and any $u \in [1, 1/m]$,

$$u \log u \ \leq \ \frac{u - 1}{m}. \tag{36}$$

To see this, define $h(u) := \frac{u-1}{m} - u \log u$. Then $h''(u) = -1/u < 0$, so $h$ is concave. Also $h(1) = 0$. At $u = 1/m$,

$$h(1/m) = \frac{1/m - 1}{m} - \frac{1}{m} \log \frac{1}{m} = \frac{1}{m} \left( \frac{1 - m}{m} - \log \frac{1}{m} \right) \geq 0,$$

where the last inequality follows from $\log x \leq x - 1$ with $x = 1/m$. By concavity, $h(u) \geq 0$ for all $u \in [1, 1/m]$, proving (36).

Applying (36) with $m = q_{\min}$ and $u = u_i$ yields

$$p_i \log \frac{p_i}{q_i} = q_i \, u_i \log u_i \ \leq \ q_i \cdot \frac{u_i - 1}{q_{\min}} = \frac{p_i - q_i}{q_{\min}}.$$

Summing over $i \in A$ gives

$$D_{\mathrm{KL}}(p \| q) \ \leq \ \frac{1}{q_{\min}} \sum_{i \in A} (p_i - q_i).$$

which completes the proof. $\qquad\qquad\qquad\qquad\qquad\qquad\qquad\qquad\qquad\qquad\qquad\qquad\qquad\square$

**Lemma 30** (KL-logit inequality (Lemma 27 of (Mei et al., 2020b))). *For $\theta \in \mathbb{R}^{|\mathcal{A}|}$, define $\mathrm{softmax}_\theta \in \mathcal{P}(\mathcal{A})$ such that*

$$\mathrm{softmax}_\theta(a) = \frac{\exp(\theta(a))}{\sum_{a' \in \mathcal{A}} \exp(\theta(a'))} \ .$$

*Fix $\theta, \theta' \in \mathbb{R}^{|\mathcal{A}|}$. Then for any constant $c \in \mathbb{R}$, we have*

$$\mathrm{KL}(\mathrm{softmax}_\theta \, \| \, \mathrm{softmax}_{\theta'}) \leq \frac{1}{2} \left\| \theta - \theta' - c \mathbf{1}_{|\mathcal{A}|} \right\|_\infty^2 \ .$$

**Lemma 31.** *Let $p = (p_i)_{i=1}^n \in \Delta_n$, where*

$$\Delta_n := \left\{ (p_1, \ldots, p_n) \in [0, 1]^n : \sum_{i=1}^n p_i = 1 \right\}.$$

*Then*

$$\sum_{i=1}^n p_i \left( \log p_i \right)^2 \leq 1 + (\log n)^2.$$

*Here and below, $\log$ denotes the natural logarithm, and we use the convention $0(\log 0)^2 := 0$, which is justified by*

$$\lim_{x \downarrow 0} x (\log x)^2 = 0.$$

*Proof.* Define

$$F_n(p) := \sum_{i=1}^n p_i (\log p_i)^2, \qquad p \in \Delta_n.$$

We prove the claim by induction on $n$.

**Step 1: the cases $n = 1$ and $n = 2$.**

If $n = 1$, then $p_1 = 1$, so

$$F_1(p) = 1 \cdot (\log 1)^2 = 0 \leq 1.$$

Now let $n = 2$. Consider the function

$$h(x) := x (\log x)^2, \qquad x \in [0, 1].$$

A direct computation gives

$$h'(x) = \log x \, (\log x + 2).$$

Hence the only critical point in $(0, 1)$ is $x = e^{-2}$, and since $h(0) = h(1) = 0$, the maximum of $h$ on $[0, 1]$ is

$$\max_{x \in [0,1]} h(x) = h(e^{-2}) = \frac{4}{e^2}.$$

Therefore, for every $(p_1, p_2) \in \Delta_2$,

$$F_2(p) = h(p_1) + h(p_2) \leq \frac{8}{e^2}.$$

Also,

$$e > 1 + 1 + \frac{1}{2} + \frac{1}{6} = \frac{8}{3},$$

so

$$\frac{8}{e^2} < \frac{8}{(8/3)^2} = \frac{9}{8} < \frac{5}{4}.$$

On the other hand, since $e < 4$, we have $\sqrt{e} < 2$, hence

$$\log 2 > \frac{1}{2},$$

and therefore

$$1 + (\log 2)^2 > 1 + \frac{1}{4} = \frac{5}{4}.$$

Combining the last two inequalities,

$$F_2(p) \leq \frac{8}{e^2} < \frac{5}{4} < 1 + (\log 2)^2.$$

So the result holds for $n = 2$ as well.

**Step 2: induction hypothesis.**

Assume now that $n \geq 3$, and that the statement has already been proved for all dimensions $1, 2, \ldots, n - 1$.

Let $p^\star \in \Delta_n$ be a maximizer of $F_n$ on $\Delta_n$. Such a maximizer exists because $\Delta_n$ is compact and $F_n$ is continuous.

We distinguish two cases.

**Case 1: $p^\star$ lies on the boundary of $\Delta_n$.**

Then at least one coordinate of $p^\star$ is zero. Let $m$ be the number of positive coordinates of $p^\star$. Then $1 \leq m < n$. After removing the zero coordinates, we obtain a vector $q \in \Delta_m$ such that

$$F_n(p^\star) = F_m(q),$$

because the zero coordinates contribute nothing to the sum.

By the induction hypothesis,
$$F_n(p^\star) = F_m(q) \leq 1 + (\log m)^2 \leq 1 + (\log n)^2.$$

So the claim holds in this case.

**Case 2: $p^\star$ lies in the interior of $\Delta_n$.**

Then every coordinate of $p^\star$ is strictly positive, so we may apply the method of Lagrange multipliers to maximize $F_n$ under the constraint $\sum_{i=1}^n p_i = 1$. Thus there exists $\lambda \in \mathbb{R}$ such that for every $i = 1, \ldots, n$,

$$\frac{\partial}{\partial p_i} F_n(p^\star) = \lambda.$$

Since

$$\frac{d}{dx}\left[x(\log x)^2\right] = (\log x)^2 + 2\log x,$$

we get

$$(\log p_i^\star)^2 + 2\log p_i^\star = \lambda \qquad \text{for all } i = 1, \ldots, n.$$

Now suppose two positive numbers $a, b$ satisfy

$$(\log a)^2 + 2\log a = (\log b)^2 + 2\log b.$$

Subtracting the two sides gives

$$(\log a - \log b)(\log a + \log b + 2) = 0.$$

Hence either

$$a = b,$$

or

$$\log a + \log b + 2 = 0, \qquad \text{i.e.} \qquad ab = e^{-2}.$$

It follows that the coordinates of $p^\star$ can take at most two distinct values. So there are two possibilities:

*(i) All coordinates are equal.* Then

$$p_i^\star = \frac{1}{n} \qquad \text{for all } i,$$

and therefore

$$F_n(p^\star) = n \cdot \frac{1}{n} \big(\log(1/n)\big)^2 = (\log n)^2 \le 1 + (\log n)^2.$$

*(ii) There are exactly two distinct values.* Say $a$ occurs $k$ times and $b$ occurs $n - k$ times, where $1 \le k \le n - 1$, $a \ne b$, and necessarily

$$ab = e^{-2}.$$

Since the coordinates sum to 1,

$$ka + (n - k)b = 1.$$

Substituting $b = e^{-2}/a$ into this equation yields

$$ka + (n - k)\frac{e^{-2}}{a} = 1,$$

or equivalently

$$ka^2 - a + (n - k)e^{-2} = 0.$$

This is a quadratic equation in $a$, whose discriminant is

$$\Delta = 1 - 4k(n - k)e^{-2}.$$

But for $1 \le k \le n - 1$ we have

$$k(n - k) \ge n - 1,$$

so, since $n \ge 3$,

$$\Delta \le 1 - 4(n - 1)e^{-2} \le 1 - 8e^{-2} < 0.$$

This is impossible. Hence case (ii) cannot occur.

Therefore the only interior maximizer is the uniform distribution, and in that case

$$F_n(p^\star) = (\log n)^2 \le 1 + (\log n)^2.$$

Since every maximizer satisfies the desired bound, we conclude that for every $p \in \Delta_n$,

$$F_n(p) = \sum_{i=1}^n p_i (\log p_i)^2 \le 1 + (\log n)^2.$$

This completes the proof. □

