# OpenReview forum: "Refined Analysis of Entropy-Regularized Actor-Critic"
_ICML.cc/2026/Conference — ICML 2026 regular_

### Official Review · Reviewer_49KT · 2026-03-12

**Soundness:** 4
**Presentation:** 3
**Significance:** 3
**Originality:** 3
**Overall Recommendation:** 5
**Confidence:** 3

**Summary:**

The authors revisit the importance of the critic in entropy-regularized actor-critic reinforcement learning settings. Under the assumption of an (exactly) known critic, they establish a logarithmic convergence guarantee, matching that of deterministic policy gradients. In the general unknown case, the critic has to be updated alongside the actor, for which the authors derive a novel $1/\epsilon$ convergence guarantee to achieve an $\epsilon$-optimal objective. They derive an actor recursion formula which reveals that the improvement in each actor step is controlled by the bias and the variance of the critic, highlighting the importance of a good critic estimation.

**Compliance With Llm Reviewing Policy:**

Affirmed.

**Key Questions For Authors:**

Row one of Figure 1 looks slightly concerning. As the size of the grid-world slightly increases, the performance almost completely degenerates in all non-exact-critic settings. Can the authors elaborate further on this? Additionally experiments with larger scale grid-worlds e.g. 10x10 would also be highly appreciated.

**Limitations:**

Yes

**Strengths And Weaknesses:**

**Strengths:**

-- The paper is well-written and provides strong theoretical results. The proofs are detailed, and the sketches in the main part provide helpful intuition.

-- The idea of revisiting the role of the critic is well-motivated and practically relevant. Comparison to prior work is sound, and results are positioned well with respect to concurrent literature.

-- Results in the exact critic setting are very interesting, and the subsequent analysis of the generalization in Theorem 2 is very strong.

**Weaknesses:**

-- The methodology for the theoretical analysis seems to be heavily tied to the entropy-regularized objective, cf. Lemma 4 connecting the lower bound on $\pi_\min$ with higher (regularized) objective values. It is not clear to me how this could be translated to the case without regularization. The literature review on policy gradient methods is also missing works addressing the case without entropy regularization.

-- Beyond the tabular setting, e.g., continuous state-action spaces, controlling $\pi_\min$ using the operator $U_\tau$ seems quite challenging. It is not quite clear how this could potentially be generalized.

-- The experiments seem to be limited to very small state-action spaces (see key questions below). It would be very interesting to see if the approach is scalable (beyond the theoretically analyzable setting) to continuous state-action space examples.

-- Figure 1 (e) shows a significant drop in performance after the first iteration. Could the authors elaborate on this behavior?
Some statements in the text are not precise, e.g., the contributions claim “We propose a novel theoretical analysis of actor-critic.” -

-- However, Table 1 shows there were several prior work analysing actor-critic algorithms. Also, in Table 1 the references and column titles are not consistent. For example, “algorithm” column does not include algorithm names for prior work.

**Minor comments:**

-- Eq (2) typo $\rho$ has no argument.

-- It could be helpful to include explicit references to the proofs at each theorem/lemma/… to the appendix so jumping between the main part and the appendix is easier.

---

> ### Author Rebuttal · Authors · 2026-03-31
>
> We thank the reviewer for their thoughtful comments. We are glad that the reviewer found that our **"results in the exact critic setting are very interesting"**,  that **"Theorem 2 is very strong"**, and that our paper is **"well-written"**. Below we address each of the reviewer's remarks.
>
> > **"The methodology for the analysis seems to be tied to the entropy-regularized objective[...] It is not clear how this could be translated to the case without regularization"**
>
> We agree that our current analysis is tailored to the entropy-regularized actor-critic. A key motivation for focusing on the regularized setting is that, in practice, actor-critic methods are almost always implemented with some form of regularization.
>
> That said, our results can still be related to the unregularized objective. Specifically, fix any $\epsilon>0$. By Theorem 2 in [1], if $$\lambda \leq \frac{(1-\gamma)\epsilon}{\log(|A|)},$$ then an optimal policy for the regularized problem is $\epsilon$-optimal for the unregularized objective. Thus, by running regularized AC with a sufficiently small temperature, our guarantees for the regularized problem directly translate to the unregularized one.
>
> > **"The literature review on policy gradient methods is missing works addressing the case without entropy regularization"**
>
> Since our focus is on the regularized setting, we originally excluded the work in unregularized setting.
> Yet, we agree that these works are significant prior work and should be acknowledged. In the revision, we will expand the discussion and add the following references:
>
> - Lu et al, Towards Principled, Practical Policy Gradient for Bandits and Tabular MDPs
>
> - Mei et al, Escaping the Gravitational Pull of Softmax
>
> - Mei et al, Ordering-based Conditions for Global Convergence of Policy Gradient Methods
>
> If the reviewer had additional references in mind, we would be happy to include them as well.
>
> > **"Beyond the tabular setting, controlling $\pi_{min}$ using  $U_{\tau}$ seems quite challenging"**
>
> The focus of our work is to understand the variance-reduction properties of actor-critic methods. To our knowledge, this result was not known in the tabular setting, which justifies the focus on such problems.
>
> While this is beyond the scope of our paper, extending the analysis to the continous setting would require first establishing a non-uniform Łojasiewicz inequality for the regularized objective. If the corresponding coefficient depends on a lower bound on the policy density, one could then hope to design a projection mechanism analogous to $U_{\tau}$. We believe this is a highly non-trivial and promising direction for future work.
>
> >**"The experiments seem to be limited to very small state-action spaces [...] Experiments with larger grid-worlds e.g. 10x10 would also be highly appreciated"**
>
> In the following anonymous link, we provide the requested experiments:
>
> https://anonymous.4open.science/r/Plots-E306
>
> These results corroborate the trends that we already observed: better critic estimation leads to better performance. We will include these additional experiments in the revised manuscript.
>
> >**" It would be interesting to see if the approach is scalable to continuous space"**
>
> While our theory is limited to the tabular setting, the main message extends more broadly: accurate critic estimation can substantially improve AC performance. This is exactly the phenomenon studied empirically in [1], where Figure 4 shows that increasing the number of critic updates significantly improves performance on continuous-control MuJoCo tasks.
>
> > **"Figure 1 (e) shows a significant drop in performance after the first iteration"**
>
> This initial drop is caused by the critic being highly inaccurate at the start of training. In the early iterations, the bias in the critic estimate can lead to harmful actor updates and hence to a temporary decrease in performance. Similar behavior can also be observed in prior AC papers, for instance on the Ant-v2 environment in Figure 1 of [2]. As the critic estimate improves, the actor updates become more reliable, leading to the subsequent recovery.
>
> > **"Some statements in the text are not precise, e.g. “We propose a novel theoretical analysis of actor-critic"**
>
>  In the revision, we will replace this statement with:
> “We propose a new theoretical analysis of actor-critic that clarifies the role of the critic, leading to an $O(\log(1/\epsilon))$ rate in the exact-critic setting and an $O(1/\epsilon)$ rate in the general setting.”
>
> >**"Typo Eq (2), issue in the column name of Table 1,  and explicit references to the proof"**
>
> We thank the reviewer for pointing out this typo and the inconsistency in the column name of Table 1, which will be corrected in the revised manuscript.  We will also include explicit references to the proofs at each theorem/lemma.
>
> [1] Wang et al, Improving Value Estimation Critically Enhances Vanilla Policy Gradient
>
> [2] Haarnoja et al, Soft Actor-Critic Algorithms and Applications

---

> > ### Author Rebuttal · Reviewer_49KT · 2026-04-05
> >
> > I thank the authors for addressing the remaining questions. Regarding additional related work related to the unregularized setting, I can suggest including:
> >
> > $\bullet$ Fatkhullin et al. Stochastic policy gradient methods: Improved sample complexity for fisher-non-degenerate policies.
> >
> > $\bullet$ Barakat et al. Reinforcement Learning with General Utilities: Simpler Variance Reduction and Large State-Action Space
> >
> > $\bullet$ Mondal and Aggarwal. Improved Sample Complexity Analysis of Natural Policy Gradient Algorithm with General Parameterization for Infinite Horizon Discounted Reward Markov Decision Processes
> >
> > The first work works with an unregularized setting in a continuous state action space, and works with a different (Hessian based) variance reduction mechanism using uniform Lojasiewicz-type inequality. The second one also uses a different (momentum and importance samplibe-based) variance reduction mechanism and uses hidden convexity properties of unregularized objectives. The third one also tackles an unregularized objective, but improves the sample complexity of PG methods using another approach -- the one based on inverting a Fisher information matrix.
> >
> > These works are complementary to the current submission and can help to better contextualize the new results in the broader literature on unregularized MDPs.

---

> > > ### Author Response · Authors · 2026-04-05
> > >
> > > We are very grateful to the reviewer for acknowledging that we addressed all of their remaining questions and for suggesting these helpful references. We agree that these works are complementary to our paper and help place our results in the broader literature on unregularized MDPs. We will include and discuss them in the related-work section of the revised manuscript.

---

### Official Review · Reviewer_mNBJ · 2026-03-13

**Soundness:** 3
**Presentation:** 3
**Significance:** 3
**Originality:** 3
**Overall Recommendation:** 5
**Confidence:** 3

**Summary:**

The paper analyzes entropy-regularized, tabular, discounted actor–critic. The paper shows that when the critic is exact, the stochastic policy-gradient estimator’s variance can be bounded by a constant times the squared norm of the true gradient, so the noise vanishes near optimality and the method achieves linear (geometric) convergence with only $\widetilde{O}(\log(1/\epsilon))$ sample complexity. With a learned critic, the paper quantifies how critic estimation error affects the convergence and derives an $\widetilde{O}(1/\epsilon)$ sample complexity guarantee when the critic is trained sufficiently. The experiment on synthetic tabular MDP confirms the theoretical findings of the paper.

**Compliance With Llm Reviewing Policy:**

Affirmed.

**Final Justification:**

The paper is technically solid and well written. The authors have also addressed my concerns in the response. Therefore, I increased my score to 5.

**Key Questions For Authors:**

see weaknesses

**Limitations:**

yes

**Strengths And Weaknesses:**

Strength:

1. The paper establishes $\widetilde{O}(\log(1/\epsilon))$ sample complexity with an exact critic and $\widetilde{O}(1/\epsilon)$ sample complexity with a learned critic. These are compelling rates; in particular, the learned-critic result appears to be the first such $\widetilde{O}(1/\epsilon)$ guarantee in the entropy-regularized tabular actor-critic setting (as positioned by the paper).

2. The paper is generally well written and goes beyond stating convergence rates. For the learned-critic case, it offers an intuitive decomposition of error sources and explains why critic error/variance can dominate the actor's progress. This framing makes the proof strategy easier to follow and helps connect the theory to practical design choices (e.g., spending more updates on the critic).


Weaknesses and questions

1. The analysis requires a uniform lower bound on action probabilities, enforced via an additional projection that restricts the policy to a smaller feasible set. This introduces additional tuning. It would be helpful if the authors clarified whether this projection is essential in practice or primarily a technical device for the analysis or it is mostly for proof convenience.

2. While the theoretical development is solid and the authors do attempt to provide intuition, parts of the presentation are quite technical for the main body. I would encourage moving more of the heavier technical details to the appendix and expanding the high-level explanation in the main text.

3. The results are limited to the tabular case. Do the authors see a path to extending the analysis to broader settings, such as linear function approximation (or other restricted function classes)?

Overall, this is a theoretically solid and well-motivated paper with strong sample-complexity results and a helpful conceptual explanation of why the critic can substantially accelerate actor updates. I would be happy to see it accepted, and I think addressing the points above would further improve clarity and impact.

---

> ### Author Rebuttal · Authors · 2026-03-31
>
> We thank the reviewer for carefully reviewing our paper; we are glad that the reviewer found our paper **"well-written"** the derived rates **"compelling"** and that **the reviewer "would be happy to see the paper accepted"**. Below, we address each of the reviewer's concerns point by point.
>
> > **"1)The analysis requires a uniform lower bound on action probabilities, enforced via an additional projection that restricts the policy to a smaller feasible set. This introduces additional tuning. It would be helpful if the authors clarified whether this projection is essential in practice or primarily a technical device for the analysis or it is mostly for proof convenience."**
>
> We thank the reviewer for this important question. The projection operator is primarily a technical device for the analysis, rather than a component that is critical in practice. More precisely, its role is to ensure that the non-uniform Łojasiewicz coefficient remains uniformly bounded away from zero along the iterates.
>
> By contrast, theoretical works that do not use such a projection typically assume directly that this non-uniform Łojasiewicz coefficient remains almost surely uniformly bounded away from zero; see, e.g., [1] and [2]. This is a strong and unverifiable assumption, since the coefficient depends on the full trajectory of the iterates and cannot be checked in advance. Our projection operator is introduced precisely to avoid making this assumption and instead guarantee the required lower bound by construction.
>
> An interesting open direction is therefore to determine whether guarantees comparable to ours can be established without introducing such an operator. We will make this point more explicit in the revised manuscript.
>
> [1] Lu et al, Towards Principled, Practical Policy Gradient for Bandits and Tabular MDPs, RLC 2024
>
> [2] Kumar et al, On the Convergence of Single-Timescale Actor-Critic, NeurIPS 2025
>
> >**"2)While the theoretical development is solid and the authors do attempt to provide intuition, parts of the presentation are quite technical for the main body. I would encourage moving more of the heavier technical details to the appendix and expanding the high-level explanation in the main text"**
>
> We thank the reviewer for this very useful feedback. We agree that some parts of the presentation can be made lighter in the main body. In the revised version, we will move the technical Lemma 8, which is mainly an auxiliary ingredient for Lemma 10, to the appendix, and add in the main text a higher-level proof sketch of Lemma 10. We believe this change will make the presentation more readable and accessible. If there are other passages that the reviewer also finds too technical for the main body, we would be very happy to move them to the appendix as well and replace them with higher-level explanations.
>
> >**"3) The results are limited to the tabular case. Do the authors see a path to extending the analysis to broader settings, such as linear function approximation (or other restricted function classes)?
> "**
>
> We thank the reviewer for this excellent question. Yes, we do see a possible path, although it is likely to be technically quite involved. At present, the strongest global-convergence results for policy-based methods with linear function approximation are for vanilla policy gradient in the stochastic bandit setting [3]. A natural roadmap would therefore be to first extend these bandit results to the entropy-regularized RL setting, and then investigate whether an analogous gradient-domination property can be established there as well. If such a property can indeed be proved, we believe that the rest of our analysis could be extended using decompositions similar to those developed in the present paper.
> That being said, this is outside the scope of this manuscript, which focuses on the tabular setting, but this constitutes a very promising direction for future research.
>
> [3] Lin et al, Rethinking the Global Convergence of Softmax Policy Gradient with Linear Function Approximation, NeurIPS 2025
>
> ---
> We thank the reviewer again and remain available for any further questions.

---

> > ### Author Rebuttal · Reviewer_mNBJ · 2026-04-03
> >
> > I thank the authors for their response! The response addressed my concerns, and I have adjusted my rating accordingly.

---

### Official Review · Reviewer_tAsz · 2026-03-13

**Soundness:** 3
**Presentation:** 3
**Significance:** 3
**Originality:** 3
**Overall Recommendation:** 5
**Confidence:** 3

**Summary:**

The paper takes a discrete and finite state and action space, infinite-time-horizon discounted and entropy regularized MDP and studies convergence of an actor critic algorithm.

The algorithm is "double loop".
Before each policy update the the critic is updated using TD (Temporal difference) learning $H$ times (to ensure good critic values).
The policy $\pi_\theta$ is then updated using a mirror descent step with penalty $1/\eta_a$.
The update is written in the logit space i.e in $\theta$ followed by a projection to ensure no action degenerates below threshold $\tau_\lambda$ for any state.
Note that the algorithm isn't based on rollouts but rather uses a strong sampling assumption, see discussion below.

In this setting the authors then prove sample sample complexity  $\tilde{\mathcal O}(1/\varepsilon)$.

The key tools in some sense are: L-smoothness and the non-uniform Lojasiewicz inequality of (Mei et. al, 2020b) and then the clipping operator $\mathcal T_{\tau_\lambda}$, introduced in (Zhang et al.
2021a), (Labbi et al. 2026) which has the crucial improvement propery Lemma 4.

**Compliance With Llm Reviewing Policy:**

Affirmed.

**Final Justification:**

The questions this reviewer left and the weaknesses pointed out have been addressed.

To some extent this reviewer agrees with `bTJt` that the work perhaps isn't all that significant. However some of the techniques used are interesting and the paper is well written and most likely correct.

So this reviewer is happy to keep their assessment as is.

**Key Questions For Authors:**

Q1 Would it be better to write $\mathcal T_{\tau_\lambda}$ on line 10 of Algorithm 1 to tie the projection directly to the theory?

Q2 Should Lemma 5 we phrased as "Let $\theta \in \mathbb R^{|S||A|}$ be such that $\pi_\theta(a|s) \geq \pi_{\min} > 0$. Then it holds that $\ldots$".

Q3 Should the filtration (defined on p. 13) be written as $\mathcal F_K = \ldots$ since little $k$ is just a set index on the right?

Q4 Can you comment on whether the clipping operator would have an analogue for general action spaces $A$? I.e is there any hope to adapt for density functions?

**Limitations:**

1. It should be noted that the algorithm assumes that one can first sample any state $s$ from the occupation measure under policy $\pi_\theta$ denoted $d^\theta_\rho$, then sample $a$ from $\pi_\theta(\cdot|\tilde s)$ and then obtain $\tilde s \sim P(\cdot|s,a)$ and finally $\tilde a \sim \pi_\theta(\cdot|\tilde s)$.
This is an extremely strong assumption about how we can interact with the environemnt which (beyond simle grid-world MDPs) is unlikely to hold in applications to true black box systems.

2. As the authors point out it's not clear what would carry over to continous state-action spaces.

**Strengths And Weaknesses:**

### Strengths

S1 The topic is interesting and the paper is well written.

S2 The result is strong (within the limitations) and seems novel to this reviewer.

### Weaknesses

W1 Please clarify which of the papers in Table 1 also use this strong sampling assumption (see limitations below for details). It seems to be present in Cayci et al. 2024 but not clear if it's common across all the works you're comparing to.

W2 In the statement of Lemma you write "The regularized value $\tilde v^\lambda_{\pi_{\theta}}(\rho)$ is L-smooth." But L-smoothness is a property of functions and what you wrote is a number. Of course in many situations this would be interpretable but faced with $\tilde v^\lambda_{\pi_{\theta}}(\rho)$ a reader can choose this to be a function of: $\lambda$, $\rho$ and $\theta$.
Of course you mean that as a function of $\theta$ the function is L-smooth.
You could write $\mathbb R^d \ni \theta \mapsto \tilde v^\lambda_{\pi_{\theta}}(\rho) \in \mathbb R$ is L-smooth.
It would also help to spell out what it means to avoid confusion: L-smoothness says that first order Taylor expansion is everywhere accurate up to $\frac{L}{2}\|\theta - \theta'\|^2$ term which is: for all $\theta, \theta'\in \mathbb R^d$ we have
$$
|\tilde v^\lambda\_{\pi\_{\theta'}}(\rho) - \tilde v^\lambda_{\pi_{\theta}}(\rho) - \langle \nabla_\theta \tilde v^\lambda_{\pi_{\theta}}(\rho), \theta'-\theta\rangle|\leq \frac{L}2\|\theta'-\theta\|_2^2\,.
$$


W3 Proof of Theorem 3 doesn't look correct as written. First of all you write: "Applying Lemma 2, combined with Lemma 24 yields ..." If you really need Lemma 24 you need to verify the Hessian condition needed for Lemma 24.
On the other hands, it seems, at first sight that Lemma 2 gives you L-smoothness which, taking $\theta=\theta_k$ and $\theta'=\theta_{k+1}$ would yield

$$ \-\frac{L}{2}\|\theta_{k+1}-\theta_k\|\_2^2 + \langle \nabla_\theta \tilde v^\lambda_{\pi_{\theta_k}}(\rho), \theta_{k+1}-\theta_k\rangle + \tilde v^\lambda_{\pi_{\theta_k}}(\rho)\leq \tilde v^\lambda_{\pi_{\theta_{k+1}}}(\rho). $$
That's almost the statement you have, in fact it would be if we could write
$$
\theta_{k+1} = \theta_k + \eta_a g_a^{Y_{k+1}}(\tilde a_{\theta_k}^\lambda).
$$
But this isn't the actor update as you have (line 10 of Algorithm 1) the truncation $\mathcal T$.
Because of Lemma 4 this looks fixable: apply L-smoothness to an intermediate $\theta' = \theta_k + \eta_a g_a^{Y_{k+1}}(\tilde a_{\theta_k}^\lambda)$ and then use that $\theta_{k+1} = \mathcal T_{\tau_\lambda}[\theta']$ and so because of Lemma 4 $v^\lambda_{\pi_{\theta_{k+1}}}(\rho) \geq v^\lambda_{\pi_{\theta'}}(\rho)$.
Okay so after all this: you just need the correct reference to Lemma 4 not Lemma 24.

W4 Perhaps discussing some recent work on continuous action spaces would be interesting. There the main tool is Bregman proximal inequality (3 point lemma)
and performance difference leading L-smoothness and "almost convexity", see

- Lan, G. (2023). Policy mirror descent for reinforcement learning: Linear convergence, new sampling complexity, and generalized problem classes. Mathematical programming, 198(1), 1059-1106.
- Kerimkulov, B., Leahy, J. M., Siska, D., Szpruch, L., & Zhang, Y. (2025). A Fisher–Rao Gradient Flow for Entropy-Regularised Markov Decision Processes in Polish Spaces: B. Kerimkulov et al. Foundations of Computational Mathematics, 1-75.

---

> ### Author Rebuttal · Authors · 2026-03-31
>
> We sincerely thank the reviewer for the very careful reading of our manuscript and for the constructive comments. We are  glad that the reviewer found the **"paper well-written"** and the result both **"strong and novel'**. We address each of the reviewer’s remarks below.
>
> > **"W1-Please clarify which of the papers in Table 1 also use this strong sampling assumption"**
>
> We thank the reviewer for allowing us to clarify. The related actor-critic analyses in Table 1 rely on comparable sampling assumptions. In particular, in [1], [2], and [4], sampling is performed through the discounted occupancy measure, which is the same as in our manuscript. In [3], the authors instead sample directly at each round from the initial distribution $\rho$. However, if one is allowed to sample from $\rho$, then samples from $d_{\rho}^{\theta}$ can be generated via the standard geometric sampling procedure described in Algorithm 1 of [5]. We will make this comparison explicit in the revised manuscript.
>
> > **"W2 - In the statement of Lemma you write ”The regularized value $\tilde{v}_{\theta}^\lambda$ is L-smooth.” But L-smoothness is a property of functions and what you wrote is a number [...]"**
>
> Thank you for pointing out this typo. We will revise Lemma 2 by using the exact wording suggested by the reviewer: ``The regularized value  $R^{|S||A|} \ni \theta \mapsto v_{\pi_{\theta}}^{\lambda}(\rho) \in R$ is L-smooth, that is, for all  [...]".
>
> > **"W3-Proof of Theorem 3 doesn't look correct as written. [...] Okay so after all this: you just need the correct reference to Lemma 4 not Lemma 24."**
>
> We are very grateful to the reviewer for carefully checking the appendix. There is indeed a small missing argument, **which does not affect the validity of our result**: as written, the proof omits an intermediate step, and the reference is not the right one.  We should first apply $L$-smoothness to the intermediate point $\theta'=\theta_k+\eta_a\, g_a^{Y_{k+1}}(\hat a_{\theta_k}^\lambda)$
> which gives a descent inequality at $\theta'$, and then invoke Lemma 4, rather than Lemma 24, to get  $\tilde v_{\pi_{\theta_{k+1}}}^\lambda(\rho)\ge \tilde v_{\pi_{\theta'}}^\lambda(\rho)$.
> We will make this intermediate argument explicit in the revised manuscript.
>
> > **"W4-Perhaps discussing some recent work on continuous action spaces would be interesting [...]"**
>
> We will make sure to cite these works by adding the following discussion at the end of the paragraph ``Policy Gradient Methods'' in the related work section, replacing lines 84--90 by:
>
> ``With stochastic gradients, Zhang et al. (2021b); Yuan et al. (2022) proved convergence to first-order stationary points under Monte-Carlo gradient estimates, and subsequent work showed that, **in the tabular setting**, global guarantees can be recovered under additional structure, notably through regularization (Zhang et al.,2021a; Ding et al.,2025; Labbi  et al.,2026). **In contrast, in continuous action settings, even regularized gradient-based methods~[6,7] are only known to enjoy global convergence guarantees in deterministic regimes**.''
>
> > **"Q1, Q2, and Q3 --- Notational corrections."**
>
> These modifications would indeed make our paper more rigorous, we sincerely thank the reviewer for pointing them out. We will make the corrections in the revised manuscript.
>
> > **"Q4 Can you comment on whether the clipping operator would have an analogue for general action spaces?"**
>
> The role of the clipping operator in our analysis is to keep the non-uniform Łojasiewicz coefficient uniformly bounded away from zero. In a general action-space setting, one would need two ingredients:
>
> - a counterpart of the non-uniform Łojasiewicz inequality for a suitable policy parametrization over densities;
>
> - an operator on policies or densities that is value-improving while also enforcing the Lojasiewicz coefficient to stay bounded away from zero.
>
> At this stage, we prefer not to overstate what is currently possible: without a precise analogue of the non-uniform Łojasiewicz inequality in the continuous setting, it is difficult to propose a definitive operator. This is an exciting direction for future work.
>
> ---
>
> We thank the reviewer again for the thoughtful comments.
>
> [1] Olshevsky et al, A small gain analysis of single timescale actor critic
>
> [2] Kumar et al. On the convergence of single-timescale actor-critic
>
> [3] Gaur et al, Closing the gap: Achieving global convergence (last iterate) of actor-criti cunder markovian sampling with neural network parametrization
>
> [4] Cayci et al. Finite-time analysis of entropy-regularized neural natural actor-critic algorithm
>
> [5] Agarwal et al, On the Theory of Policy Gradient Methods: Optimality, Approximation, and Distribution Shift
>
> [6] Kerimkulov et al, A Fisher-Rao gradient flow for entropy-regularised Markov decision processes in Polish spaces
>
> [7] Zorba et al, Convergence of an Actor-Critic gradient flow for entropy regularised MDPS in general spaces

---

> > ### Author Rebuttal · Reviewer_tAsz · 2026-04-03
> >
> > The minor comments, questions left by this reviewer have been addressed. Just as an aside, the "strong sampling assumption" is commonly referred to as solving the "planning problem"; that is probably the terminology the reviewer should have used but better mention it now than never.

---

### Official Review · Reviewer_bTJt · 2026-03-15

**Soundness:** 3
**Presentation:** 3
**Significance:** 2
**Originality:** 2
**Overall Recommendation:** 3
**Confidence:** 5

**Summary:**

This paper studies entropy-regularized Actor-Critic. First, under deterministic critic (where we assume the exact action value function is known) and stochastic actor, they show that the entropy-regularized Actor-Critic converges geometrically fast. To show this, the authors particularly take advantage of Lemma 5 which shows that the variance of the stochastic actor can be bounded by the expectation of the actor gradient. Next, the authors extend their result to stochastic critic, and show that it has the typical convergence of the stochastic approximation as the summation of an exponentially decaying bias and a constant variance proportional to the step size.

**Compliance With Llm Reviewing Policy:**

Affirmed.

**Key Questions For Authors:**

- What is the purpose of Theorem 1? Eventually, what matters is the analysis under stochastic critic (which is what you can practically implement), which is done in Theorem 2. What do you mean by variance reduction in Theorem 1? In this theorem you are not studying any variance reduction. Rather, it is just a property of the regularized actor-critic with deterministic critic that can achieve linear rate of convergence even with stochastic update of the actor.
- How does Theorems 1 and 2 look like under time-varying step sizes?

**Limitations:**

- In lines 3 and 9 of algorithm 1 the authors assume sampling according to \nu^c and \nu^a. However, sampling according d_\rho^\theta is hard in practice, and it is a limitation of the problem setup. However, this limitation can be resolved as done by geometric sampling in Algorithm 1 in Agarwal et al. 2021.

**Strengths And Weaknesses:**

Soundness: The results seem to be correct. I did not check all the proofs, but they sound valid.

Presentation:
- Lemma 4 and the discussion coming before it is out of place. Why are you mentioning them there? You can directly go Theorem 1 and introduce the u_\tau operator only before the theorem.
- In general, you should have the main theorems first and then create a proof sketch section where you mention all the lemmas.

Significance: Given the prior work on the analysis of actor-critic, this work is not significant.

Originality: The authors need to compare their sample complexity with the sample complexity in Proposition 1 of "Policy Mirror Descent for Reinforcement Learning: Linear Convergence, New Sampling Complexity, and Generalized Problem Classes". Despite the authors' claim that they show a "variance reduction" property of entropy regularized actor-critic, it is not clear if their sample complexity is an improvement compared to the prior work.

---

> ### Author Rebuttal · Authors · 2026-03-31
>
> We thank the reviewer for the careful reading of our manuscript and for the thoughtful comments. We address the reviewer’s concerns point by point below.
>
> > **"Lemma 4 [...] is out of place. You can directly [...]introduce the $U_{\tau}$ operator before  theorem 1."**
>
> We introduce $U_{\tau}$ before Lemma 4 because it is already needed at that stage. Specifically, Lemma 4 uses $U_{\tau}$ to enforce a lower bound on $\pi_{\min}$, which is then used both to obtain a uniform Łojasiewicz constant along the iterates and to establish the variance bound in Lemma 5. Theorem 1 then follows directly from these two ingredients.
>
> Our intention was therefore to introduce $U_{\tau}$ where it first becomes necessary, while also providing intuition for its role. That being said, we are happy to move this discussion closer to Theorem 1 if the reviewer feels this would improve the presentation.
>
> > **"Given the prior work on the analysis of actor-critic, this work is not significant"**
>
> We strongly disagree with that point, and believe that our contribution is significant both quantitatively and conceptually. On the quantitative side, our analysis is the **first analysis of regularized actor-critic (AC) to achieve an $O(1/\epsilon)$ sample complexity**, vastly improving over the prior $O(1/\epsilon^5)$ guarantee.
>
> This sharper rate stems from a variance-reduction property of AC that had not been formally established before. With an exact critic, the variance of the stochastic actor update is controlled by the gradient norm and therefore vanishes near the optimum, yielding linear convergence. In the general setting, we show that the overall complexity is governed by the critic's error, which clarifies the role of the critic in AC methods.
>
> > **"Compare [...] with ”Policy Mirror Descent”**
>
> We thank the reviewer for this reference, which we will discuss in the related work. We emphasize that the goal of this paper is to develop a sharper theoretical understanding of the most widely used method in practice, that is actor critic. We prove, for the first time, that AC achieves the best known sample complexity for solving entropy regularized RL, matching the rates of policy mirror descent (PMD).
>
> This result is significant because our analysis works directly with a parameterized policy, in line with how AC methods are implemented in practice. In contrast, for methods such as PMD, it is less clear how the algorithmic counterpart extends beyond the tabular setting, since the updates are formulated directly in policy space.
>
> > **"What is the purpose of Theorem 1? [...] What do you mean by variance reduction?"**
>
> The purpose of Theorem 1 is to isolate the exact-critic regime in order to identify the mechanism through which the critic benefits the actor. In this setting, we show that actor-critic enjoys **variance-reduction** in the **strong sense**: the variance of the stochastic actor gradient estimator vanishes as the iterates approach the optimum; see, e.g., the discussion on ``modern variance reduction methods'' in [1].  This property is what enables the linear convergence result in Theorem 1.
>
> Theorem 1 aims to highlight this *strong* variance reduction property in the idealized setting where the critic is known, serving as a conceptual stepping stone for the general analysis in Theorem 2. Indeed, Theorem 2 shows that this phenomenon subsists when the critic is learned online, up to the critic variance. We will revise the discussion around Theorem 1 to make this point clearer.
>
> [1] Gower et al, Variance-reduced methods for machine learning
>
> > **"How does Theorems 1 and 2 look like under time-varying step sizes?"**
>
> The goal of this work is to establish that AC is a variance reduction method. Thus, it does not require the actor step size to decrease, which is crucial to obtain our fast rates.
>
> **In Theorem 1, there is no noise term, and thus no need for diminishing step size at all**: in fact, using diminishing step sizes for the actor would make the algorithm slower, replacing our $O(\log(1/\epsilon))$ to a much slower rate of $O(1/\epsilon)$.
>
> **In Theorem 2, however, one could use diminishing step size for the critic**, which would replace Lemma 10 by a result of the form of [2]'s Theorem 1. This would allow to take larger actor step size, but would not affect the overall $O(1/\epsilon)$ rate since the critic's diminishing step size would require more critic updates to make the critic error small. If the reviewer thinks this result is essential for the presentation of our paper, we are happy to include it in the revised manuscript.
>
> [2] Bach et al, Non-Asymptotic Analysis of Stochastic Approximation Algorithms for Machine Learning
>
> >**"The authors assume sampling according to [...] This limitation can be resolved as done by geometric sampling in Agarwal et al. 2021"**
>
> Indeed, geometric sampling overcomes this limitation, and is common practice in policy gradient methods. We will explicit this point in the revised manuscript.

---

> > ### Author Rebuttal · Reviewer_bTJt · 2026-04-04
> >
> > - If U_{\tau} is only used in the proof of Lemma 4, it does not need to necessarily be introduced before it.
> > - I still do not believe that this work is significant compared to prior work. I believe the PMD paper is significantly better in terms of sample complexity. Also, the authors are claiming that the PMD cannot be implemented in practice. But I disagree. PMD can be seen as the tabular form of PPO algorithm.
> > Hence, I keep my score.

---

> > > ### Author Response · Authors · 2026-04-05
> > >
> > > We thank the reviewer for this feedback. Below, we address the two remaining remarks in detail.
> > >
> > > > **"If $U_{\tau}$ is only used in the proof of Lemma 4, it does not need to necessarily be introduced before it."**
> > >
> > > We thank the reviewer for this helpful suggestion. In the revised version, we will move the definition of $U_{\tau}$ closer to its first essential use and recall it immediately before Theorem 1. We agree that this change will improve the flow of the presentation.
> > >
> > > > **"I still do not believe that this work is significant compared to prior work. I believe the PMD paper is significantly better in terms of sample complexity. Also, the authors are claiming that the PMD cannot be implemented in practice. But I disagree. PMD can be seen as the tabular form of PPO algorithm. Hence, I keep my score."**
> > >
> > > We respectfully disagree with the reviewer’s assessment. We would first like to clarify that we never claimed that Policy Mirror Descent cannot be implemented in the tabular setting. Rather, our point was about how our paper should be positioned with respect to the existing literature in general, and to Policy Mirror Descent in particular. We therefore restate this positioning more clearly below.
> > >
> > > Regularized actor-critic methods are among the cornerstones of modern reinforcement learning and remain one of the most widely used algorithmic frameworks in practice. **Despite this central role, the theoretical understanding of regularized Actor-Critic has remained limited, and in particular, the role of the critic in improving performance was not well understood.** Prior to our work, the best available sample complexity bound for regularized actor-critic methods was $\mathcal{O}(1/\epsilon^5)$, which is highly suboptimal. **Our paper substantially improves this understanding by establishing an $\mathcal{O}(1/\epsilon)$ sample complexity guarantee and by formally identifying the variance-reduction role played by the critic.**
> > >
> > > We agree with the reviewer that the same $\mathcal{O}(1/\epsilon)$ sample complexity for solving the entropy-regularized objective had previously been obtained for Policy Mirror Descent, which is an important related line of work. However, we do not believe that this diminishes the significance of our contribution. **Our goal is to analyze regularized actor-critic itself, as this is the method that is most central in practice and empirically successful, and to explain theoretically why this method works well.** From this perspective, our contribution is complementary rather than redundant.
> > >
> > > Additionally, while PMD is appealing in the tabular setting, it remains unclear how to extend it naturally beyond this regime, as it operates directly in the policy space. In contrast, actor-critic methods are designed to operate directly in parameter space and provide a more natural foundation for scalable extensions. **For this reason, we strongly believe that developing a sharp theoretical understanding of regularized actor-critic is both meaningful and timely.**
> > >
> > > ---
> > > We thank the reviewer again for their remarks.

---

### Decision · Program_Chairs · 2026-04-30

**Decision:**

Accept (regular)

**Comment:**

Most reviewers championed this paper because it provides strong theoretical results; in particular, it significantly tightens the sample complexity rate.

A concern was raised that the same rate has been achieved in PMD which applies to the same settings.

I found the mentioned concern is, to some extent, reasonable. However, this concern sounds more for the whole line of research on actor-critic methods. It is thus not fully fair to punish this specific paper.